# ARITHMETIC TRANSFORMERS CAN LENGTH-GENERALIZE IN BOTH OPERAND LENGTH AND COUNT

**Hanseul Cho**,[*] **Jaeyoung Cha**[*]
Kim Jaechul Graduate School of AI, KAIST
{jhs4015, chajaeyoung}@kaist.ac.kr

**Srinadh Bhojanapalli**
Google Research
bsrinadh@google.com

**Chulhee Yun**
Kim Jaechul Graduate School of AI, KAIST
chulhee.yun@kaist.ac.kr

## ABSTRACT

Transformers often struggle with *length generalization*, meaning they fail to generalize to sequences longer than those encountered during training. While arithmetic tasks are commonly used to study length generalization, certain tasks are considered notoriously difficult, e.g., multi-operand addition (requiring generalization over both the number of operands and their lengths) and multiplication (requiring generalization over both operand lengths). In this work, we achieve approximately 2–3× length generalization on both tasks, which is the first such achievement in arithmetic Transformers. We design task-specific scratchpads enabling the model to focus on a fixed number of tokens per each next-token prediction step, and apply multi-level versions of *Position Coupling* (Cho et al., 2024; McLeish et al., 2024) to let Transformers know the right position to attend to. On the theory side, we prove that a 1-layer Transformer using our method can solve multi-operand addition, up to operand length and operand count that are exponential in embedding dimension.[1]

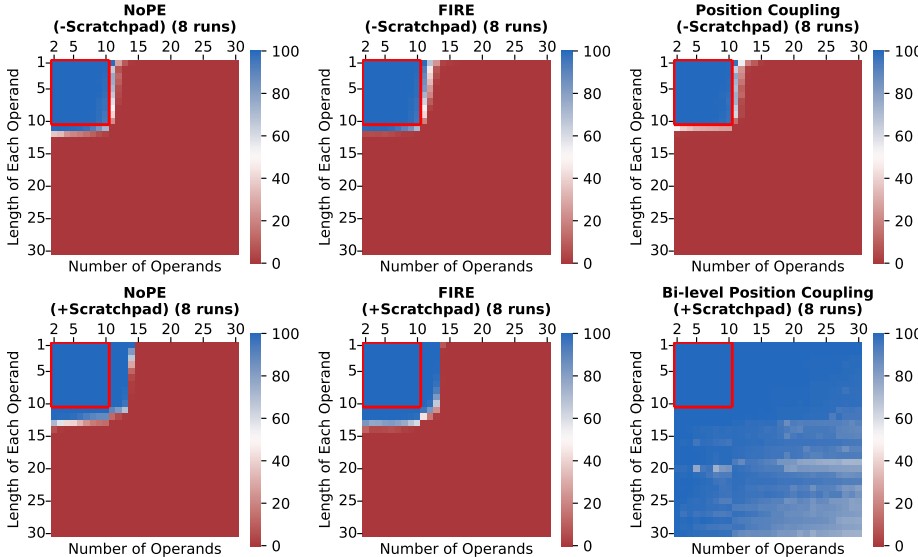

Figure 1: **Unlocking Length Generalization on Multi-Operand Addition Task.** We present median exact-match accuracies for 6-layer 8-head decoder-only Transformers trained on multi-operand additions of 2–10 operands, each having 1–10 digits (red boxes represent the scope of trained lengths). We compare three state-of-the-art position embedding (PE) methods for length generalization: NoPE (Kazemnejad et al., 2023), FIRE (Li et al., 2024), and Position Coupling (Cho et al., 2024; McLeish et al., 2024). With a proper *scratchpad* enabling Transformers to do extrinsic multi-step reasoning (described in Section 4), all three PE methods can extend their generalization scope (blue area of heatmaps). Remarkably, with our proposed **multi-level Position Coupling with scratchpad**, we achieve a significant length generalization superior to all other methods.

---

[*]Authors contributed equally to this paper.
[1]All our experiments are run with code in github.com/HanseulJo/position-coupling.

# 1 INTRODUCTION

Transformer-based language models (Vaswani et al., 2017) have become a cornerstone of modern deep learning in recent years (Chowdhery et al., 2023; Gemini et al., 2023; OpenAI, 2023; Thoppilan et al., 2022). Despite their seemingly limitless capabilities, they often struggle with a critical limitation known as the *length generalization* problem, meaning that the model does not perform well on input sequences longer than those encountered during training (Anil et al., 2022; Deletang et al., 2023; Press et al., 2022; Wu et al., 2023; Zhang et al., 2023). Length generalization has recently dragged the attention of many researchers because of the following two aspects: (1) the failure in length generalization corroborates the models' fundamental limitation that they may not genuinely understand the task-solving algorithm but may rely on short-cut learning that is only applicable to sequences of trained lengths; (2) improving length generalization can automatically extend the applicability of the models in both memory-efficient and computation-efficient way.

As manageable and intriguing test beds, arithmetic and algorithmic tasks are commonly used to study the capabilities (including length generalization) of Transformers (Cho et al., 2024; Fan et al., 2024; Kazemnejad et al., 2023; Kim et al., 2021; Lee et al., 2024; McLeish et al., 2024; Nogueira et al., 2021; Qian et al., 2023; Sabbaghi et al., 2024; Zhou et al., 2024a;b). In this paper, we mainly focus on arithmetic tasks, specifically integer addition and multiplication. While humans can easily generalize to longer examples of these tasks, recent works have shown that Transformers often struggle in length generalization, and various approaches (Cho et al., 2024; McLeish et al., 2024) have been proposed to help Transformers learn the true underlying mechanisms that solve addition and multiplication.

While recent work has made significant progress on the addition task, the scope has largely been limited to cases of two operands. Similarly, studies on multiplication have achieved length generalization for just one operand, while the other is kept fixed at a small length. Notably, as far as we know, no research has demonstrated significant generalization in terms of the *number of operands* for the addition task; e.g., training on problems with up to four operands and successfully extending to problems with more than four. Likewise, for multiplication, none has achieved length generalization for *both operands* simultaneously (e.g., see Figure 5 of McLeish et al., 2024).

In this paper, we address these challenges by proposing a nontrivial combination of two techniques: scratchpad (Nye et al., 2021) and Position Coupling (Cho et al., 2024; McLeish et al., 2024). By equipping decoder-only Transformers with a scratchpad (an appended sequence that contains multi-step reasoning) and integrating a bi-level extension of Position Coupling, we demonstrate that Transformers can learn to solve multi-operand addition, generalizing in terms of both the number of operands and their lengths. Similarly, for multiplication, we employ a scratchpad and a tri-level Position Coupling to train Transformers that length-generalize in terms of both operand lengths.

Admittedly, the two key components—scratchpads and Position Coupling—are not entirely new, and they have been adopted in existing approaches to improve length generalization in arithmetic tasks. Here, our key contribution is to combine them in a complementary way that empowers Transformers to solve the tasks that were considered very challenging. We propose novel form of scratchpads to eliminate the need to attend to an increasing number of positions as the number of operands increases. By spelling out the intermediate outcomes on scratchpads, it now suffices for the model to attend to only a constant number of tokens at each inference step. Position Coupling then offers the model information about the "correct" position(s) to attend to, thereby assisting the model to quickly learn the correct inference mechanism from data.

## 1.1 SUMMARY OF CONTRIBUTIONS

- We first tackle the multi-operand addition task (Section 4). By devising and employing a scratchpad with bi-level Position Coupling, we achieve a significant length generalization not only in the length of each operand but also in greater numbers of operands. Our model substantially improves the generalization performance (with median exact-match accuracy $\geq 90.0\%$) for the integer addition task involving up to 30 operands of lengths up to 30, even though it was trained on samples with a maximum of 10 digits and 10 operands. In contrast, models trained with either NoPE (Kazemnejad et al., 2023) or FIRE (Li et al., 2024) completely fail to solve for 13 operands of length 13, even with the help of the same scratchpad (Figure 1).

- By refining the scratchpad and employing tri-level Position Coupling, we achieve a significant length generalization for multiplication tasks where both operands can vary in length (Section 5).

Our models are capable of multiplying (up to) 20-digit integers times (up to) 15-digit integers with median exact-match accuracy $\geq 78.55\%$, even though it was trained on samples with a maximum of 10 digits for each operand (Figure 9).

- We develop a theoretical construction of a small (1-layer, 4-head) Transformer model, equipped with a scratchpad and bi-level Position Coupling, capable of solving multi-operand addition (Theorem 4.1). Our construction can handle problems involving both exponentially long operands and exponentially many operands. The scratchpad is crucial, as it enables this shallow architecture to accurately predict the next tokens by ensuring that the model only needs to attend to constant number of tokens at each inference step, which we also verify in trained Transformers (Figure 7).

## 2 PRELIMINARIES

We use next-token prediction (NTP) with decoder-only Transformers to solve every task. Each task can be represented as a set of sequences of the form "`[query]=[response]`", where the goal is to correctly infer the *response* sequence from a given *query* sequence via NTP, starting from a sequence "`[query]=`". Since we are mostly studying length generalization on arithmetic tasks, we treat a single digit (between 0–9) as a single token, but there are other non-digit tokens such as '+', '×' (or interchangeably '⋆'), '=', '→' (or interchangeably '>'), and special tokens like beginning-/end-of-sequence (BOS/EOS) and padding (PAD) tokens.[2] Moreover, because of the deterministic nature of arithmetic/algorithmic tasks, we only use greedy decoding for every NTP step. In the following subsections, we provide an explanation of the background underlying our approach. For additional related works on length generalization, we refer the readers to Appendix A.

### 2.1 RELATED WORKS

**Position Embedding Methods for Length Generalization.** Various position embedding (PE) methods have been explored to enhance the Transformers' length generalization capability. Kazemnejad et al. (2023) claim that, in some downstream tasks, a decoder-only model without PE (NoPE) can achieve a length generalization performance comparable to those of widely-used PE techniques including ALiBi (Press et al., 2022), Rotary (Su et al., 2024), and T5's Relative PE (Raffel et al., 2020). However, it is still a promising research direction to enhance length generalizability of Transformers with a more appropriate choice of PEs (Jelassi et al., 2023; Ruoss et al., 2023; Shen et al., 2023; Zhou et al., 2024b).

**Position Coupling.** Independent works by Cho et al. (2024) and McLeish et al. (2024) propose *Position Coupling* (also called *Abacus*), a structured position ID assignment rule—established on a learned absolute PE (Gehring et al., 2017)—that captures the inherent positional symmetry of the target task. In this approach, tokens in an input sequence are grouped and each group of tokens is assigned a sequence of consecutive integers as position IDs. For example, in the integer addition task, the identical position IDs are assigned to the digits at the same significance across numbers in both the query (i.e., operands) and the response (or, answer). During training, the starting position ID is randomized to mitigate the problem of encountering unseen position IDs for longer sequences. At test time, the starting position ID is arbitrarily fixed (e.g., 1). Position Coupling demonstrates a remarkable length generalization performance on several arithmetic and algorithmic tasks such as two-operand addition and $N$-digit $\times$ 2-digit multiplication.

**Scratchpad.** To enhance the reasoning capabilities of Transformer models, several heuristic-driven methods for data formatting have been introduced. One such technique is *scratchpad* (Nye et al., 2021), an auxiliary intermediate sequence of tokens before arriving at the final answer. Training the model with a scratchpad allows the model to store and refer to intermediate task-solving states when predicting subsequent tokens. This approach has been shown to enhance both in-distribution and out-of-distribution performance in tasks such as integer addition and code execution (Anil et al., 2022; Nye et al., 2021). Zhou et al. (2024a) further validate these findings in the parity task, explaining that the effectiveness of the scratchpad lies in its ability to simplify next-token prediction.

## 3 WARM-UP: LENGTH GENERALIZATION ON PARITY TASK

Before moving on to our findings about addition and multiplication tasks, we begin with a warm-up example: the *parity* task. Given a binary sequence as a query, the goal of the parity task is to output 1

---

[2]If possible, we ignore the details about the BOS/EOS/PAD tokens for a simple presentation.

if the query contains an odd number of 1's, and 0 otherwise. Despite its simple description, it is well known that Transformers struggle with achieving length generalization for the parity task (Anil et al., 2022; Deletang et al., 2023; Hahn & Rofin, 2024; Kazemnejad et al., 2023; Zhou et al., 2024a). In this section, we will demonstrate an enhanced length generalization performance on the parity task by applying Position Coupling on top of the input sequence with a properly designed scratchpad.

## 3.1 METHOD: SCRATCHPAD & POSITION COUPLING

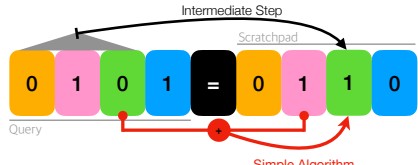

Figure 2: An illustration of a scratchpad for a parity problem (with a query `0101`).

We follow the scratchpad format proposed by Anil et al. (2022). Figure 2 illustrates an example of an input sequence consisting of a 4-bit parity problem (with query `0101`) and its scratchpad (`0110`). The idea of this scratchpad is to record every intermediate parity state as a given binary query is processed starting from the leftmost token. That is, the $k$-th bit in the scratchpad is the parity of the subsequence containing the first $k$ bits of the query. Then, the final rightmost token of the scratchpad is the desired answer for the parity task. Recording the process of solving the parity task is beneficial in the following two points:

1. *It makes the task-solving algorithm simpler.* The first token in the scratchpad is just a copy of the first token in the query sequence. Also, we do not need to directly solve the intermediate parity task at once in order to infer the $k$-th ($k > 1$) intermediate token in the scratchpad; instead, it is enough to focus on the $k$-th token of the query and the $(k-1)$-th token in the scratchpad, and then sum them up modulo 2 (see Figure 2).

2. *It is straightforward to apply Position Coupling onto the input sequence with the scratchpad.* The scratchpad generates a natural positional correspondence between the query and the response. Thus, we can assign the same position ID to the $k$-th query token and $k$-th scratchpad token, for example, `234512345` in the example depicted in Figure 2.

To sum up, such a scratchpad simplifies the task by allowing the model to perform step-by-step reasoning without need of attending to a *linearly increasing* number of query tokens until it generates the answer, while Position Coupling (applied on top of the scratchpad) can explicitly let the model know "where it should focus on" to perform every step of the reasoning. We thus can expect a synergetic effect of such combination, and our experiments do align with this expectation.

## 3.2 EXPERIMENTS & DISCUSSION

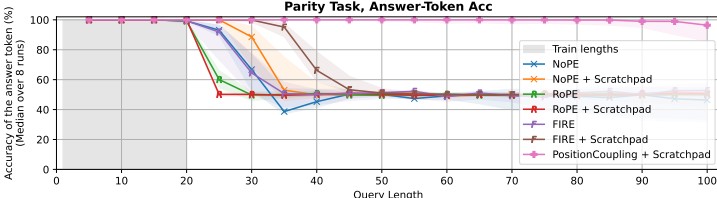

Figure 3: **Parity task.** We report the accuracies only for the answer token (i.e., the token before EOS) (light area: 95% confidence intervals). The gray region indicates the query lengths in our training data. A complete failure is indicated by the accuracy $\simeq 50\%$: a random guess between 0 and 1.

To test the efficacy of the combination of scratchpad and Position Coupling, we compare its performance against six other configurations: models trained with NoPE, RoPE, and FIRE, each tested with and without the scratchpad. The result is presented in Figure 3. Please refer to Table 1 for detailed experimental details.

We observe that, without the scratchpad, all three PE methods generalize well to in-distribution samples but struggle with out-of-distribution queries. Their performance sharply drops to 50%

accuracy for query lengths slightly exceeding the training samples, indicating no better than random guessing. When the scratchpad is employed, NoPE and FIRE demonstrate improved performance, achieving strong generalization up to 35-bit queries. However, when a scratchpad is combined with Position Coupling, the model demonstrates outstanding length extrapolation, achieving near-perfect generalization on queries up to 100 bits. From these observations, we conclude that for the parity task, (1) the scratchpad helps length generalization, albeit not significantly, and (2) the combination of the scratchpad with Position Coupling substantially boosts the model's length extrapolation capability.

# 4 LENGTH GENERALIZATION ON MULTI-OPERAND ADDITION TASK

In the previous section, we demonstrated the potential of integrating the scratchpad with Position Coupling for better length generalization. We now aim to evaluate the effectiveness of this approach on a more challenging task. Specifically, we address the problem of achieving length generalization in the integer addition task, where *operand length and count can both get longer at test time*. Indeed, there has already been a length extrapolation result on additions with more than two operands: e.g., the "triple addition" task presented by Cho et al. (2024). However, their approach still requires the number of operands to be fixed (e.g., 3), leaving the challenge of generalizing the problem setting to varying operand counts as an open question. Here, we demonstrate that it can be overcome by employing a scratchpad in conjunction with a carefully designed multi-level Position Coupling.

## 4.1 METHOD: SCRATCHPAD & BI-LEVEL POSITION COUPLING

```
Token : 057+048+096=000>750>501>102
PosID1: 4321432143212341234123412341234
PosID2: 1111222233311112222333344444
```

Figure 4: An example input sequence equipped with scratchpad and bi-level Position Coupling. The original example was "57+48+96=201": the query is inside a blue box and the response is inside a red box. All numbers in the scratchpad (i.e., intermediate steps) are reversed; all numbers in the whole sequence are minimally zero-padded to match their length.

**Scratchpad for Multi-operand Addition.** Similar to the parity task, we store the intermediate cumulative sums in the scratchpad. It makes the algorithm of solving the multi-operand addition task easier. This is because a model can obtain the $k$-th intermediate cumulative sum by adding exactly *two* numbers: the most recently generated $((k-1)$-th) intermediate sum and the $k$-th number/operand in the query. As a minor detail, we start the scratchpad with zeros in order to make the task-solving rule more clear and consistent.[3]

**Bi-Level Position coupling.** Now we motivate the usage of the bi-level Position Coupling: each token has two levels of position IDs, whose couplings happen only in each level.[4] As observed in prior works, adding two numbers can be successfully done by coupling the digits of the same significance in every number by assigning the identical position ID. The resulting position IDs (level 1) are as in `PosID1` in Figure 4. However, this is not enough because we want to know which specific numbers we should add together, while the numbers are not very distinguishable solely with level-1 position IDs. Our mitigation is to add another level of position IDs[5] that can distinguish properly between numbers. Combining the idea of Position Coupling again, we naturally couple two numbers of the same *order* in query and response: the resulting position IDs (level 2) is as in `PosID2` in Figure 4. In short, the model will choose which numbers it should add based on level-2 position IDs, and perform two-operand addition with the help of level-1 position IDs. Lastly, when we implement multi-level position IDs for each token, we use separate PE modules for different levels of position IDs to map them to PE vectors, and then add them all to the token embedding vector.

---

[3]This is not a must. We empirically observed that directly starting the scratchpad with the (zero-padded, reversed) first operand does not hurt the performance of moderate-sized models. We choose to start the scratchpads with all zeros for the sake of simplicity in our theoretical construction, later presented in Section 4.4. In fact, for multiplication (Section 5) we adopt a scratchpad that does not start with all zeros.

[4]It means that even if a token has a position ID $p$ at level 1 and another token has the same position ID $p$ at level 2, these two tokens are not necessarily coupled by these position IDs.

[5]It is worth mentioning that multi-level position ID has already been studied (He et al., 2024; Zhang et al., 2024). The implementation detail is very different because their approaches involve relative PEs.

**Input Formats.** There are additional design choices regarding the input sequence format: *zero-padding* and *number reversing*. We apply zero-padding to every number in both query and response to ensure that the length of every number is identical to the maximum possible length of the final answer, depending on the operand count. For example, if we add 11 operands of length at most $n \geq 2$, the final answer can have a length $n + 2$ (because $99 \times 11 = 1089$), so we match the length of the numbers to be $n + 2$ with zero-padding. In addition, we reverse all numbers in the scratchpad (i.e., intermediate cumulative sums). This is a quite natural choice since even humans do additions starting from the least significant digit (Lee et al., 2024; Nogueira et al., 2021).

## 4.2 EXPERIMENTAL SETUP

Given two integers $a \leq b$, we denote by $[a : b] := \{a, a + 1, \cdots, b\}$ a set of consecutive integers.

**Data Sampling.** For the sake of simplicity of notation, we denote a training set by $\mathcal{S}_A(n, m)$, consisting of addition problems where each operand can have at most $n$ digits and the operand count can range from 2 to $m$. The dataset consists of two equally sized chunks. In the first chunk, for each sample, the number of operands is uniformly sampled from $[2 : m]$, and the length of each operand is independently sampled from $[1 : n]$; this means that the operands within a single sample can differ in length. Then, based on the chosen lengths (e.g., 4), each of the operands are uniformly randomly generated (e.g., from $[1000 : 9999]$). In the second chunk, for each sample, the number of operands is still uniformly sampled from $[2 : m]$, but this time, the lengths of all operands are identical. The operand length is sampled from $[1 : n]$, and all operands are randomly chosen to be of the chosen length. We use $\mathcal{S}_A(10, 10)$ with size 500,000 as the baseline training set.

We also denote a test set by $\mathcal{T}_A(n, m)$, consisting of a single component. In each sample, both the number of operands and the length of each operand are *fixed* specifically at $m$ and $n$, respectively. For model evaluation, we draw a $30 \times 29$ grid heatmap and assess the performance of the model on the test set $\mathcal{T}_A(i, j)$ of size 1,000 for every entry $(i, j) \in [1 : 30] \times [2 : 30]$.

**Model and Training.** Our baseline model is a 6-layer 8-head decoder-only Transformer with embedding dimension 1024 (with approximately 63M parameters), trained from scratch. We do not incorporate weight decay or dropout. Further details can be found in Table 2.

**Random Offset of Position IDs During Training.** An important detail about the training procedure is that we randomly choose offsets (for each level of position IDs) and add them to every position ID in each training sample. This is to promote learning all the position embedding vectors as evenly as possible, and this training-time random assignment of position IDs is already used in prior works for similar reasons (Cho et al., 2024; McLeish et al., 2024; Ruoss et al., 2023). As Cho et al. (2024) and McLeish et al. (2024) did, we pre-define the maximum position IDs (for each level) as hyperparameters, which naturally determine maximum testable ranges of operand lengths and count. See Table 2 for our choice of the maximum position IDs.

## 4.3 EXPERIMENTAL RESULTS

**Position Coupling and Scratchpad Together Enable Powerful Length Generalization.** We trained models using 3 different PE methods—Position Coupling, NoPE, and FIRE—both with and without scratchpad. The implementation details of Position Coupling differ by the presence/absence of scratchpad: if we use scratchpads, we apply the bi-level Position Coupling explained in Section 4.1; if not, we use a simple single-level Position Coupling that matches the position IDs for digits at the same significance in all numbers, which is also used for "triple addition" task in Cho et al. (2024). The results are showcased in Figure 1. We measure the exact-match accuracy for correct inference of the whole response, including scratchpad if it is used. The top 3 heatmaps in the figure, which are the results without scratchpads, indicate that these models can only generalize to in-distribution samples and exhibit near-zero exact-match accuracy on out-of-distribution samples. Next, when either NoPE or FIRE is combined with the scratchpad, the models show a slight improvement in terms of generalizable operand lengths and counts. They barely solve problems involving 13 numbers, each 12 digits long. In contrast, the combination of the scratchpad and Position Coupling enables much stronger length generalization, achieving non-trivial accuracy even on test samples involving 30 numbers, each with 30 digits. We emphasize that such problems are extremely difficult, as the model must accurately predict a total of 1023 consecutive tokens to solve the problem.

**Effect of Trained Length.** We compare the models trained on different lengths: $\mathcal{S}_A(7, 7)$, $\mathcal{S}_A(10, 10)$, and $\mathcal{S}_A(13, 13)$. The results are presented in Figure 5. As expected, the model's ability to generalize to longer sequences improves as the training data covers a wider range of lengths.

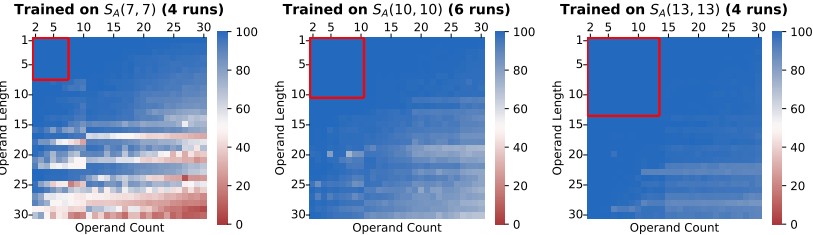

Figure 5: **Comparison of training lengths in the integer addition task.** We report exact-match accuracies (median over at least 4 runs) for the addition task. The red box indicates the training distribution: from the left plot, we trained on $\mathcal{S}_A(7, 7)$, $\mathcal{S}_A(10, 10)$, and $\mathcal{S}_A(13, 13)$.

**Effect of Zero-padding.** Figure 6 exhibits the exact-match accuracies for models trained on input sequences with scratchpad but *without zero-padding* in both the query and the response. Although there is a moderate degradation in overall performance, the models still generalize well with respect to an increased number of operands. Also, they maintain reasonable accuracy for operand lengths below 25 digits. It implies that, although zero-padding aids in enhancing length extrapolation capability, it is not an absolute necessity.

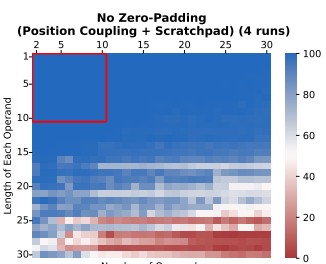

Figure 6: Training without zero-padding in input sequences.

**Effect of Architecture.** To study whether our approach can be applied across various depth/width configurations, we explore the performance as we vary the number of layers and heads. Specifically, we tested configurations with 1, 2, 4, and 6 layers, and 2, 4, and 8 heads, resulting in 12 distinct configurations. The results are illustrated in Figure 10 (see Appendix C).

It turns out that 1-layer 2-head models perform the worst. While they could generalize across operand counts, they immediately fail for longer digits. The remaining 11 configurations exhibit significantly better performance than the 1-layer 2-head models. It is noteworthy that networks with only four heads in total (1-layer 4-head or 2-layer 2-head) show surprisingly high accuracy, even outperforming our baseline (6-layer 8-head). In particular, the 2-layer 2-head models achieve at least 90.0% exact-match accuracy (median over 6 trials) across every combination $(n, m) \in [1:30] \times [2:30]$, whereas the 6-layer 8-head models achieve 67.8% or above. However, we do not observe an overall trend between the accuracy and the number of layers/heads, as the optimal number of heads varied depending on the number of layers, and vice versa. We suspect that this is due to the stochasticity of the training process.

For broader ablation results including head/embedding dimensions, training set sizes, and input formats, we refer the reader to Appendix C.

### 4.4 THEORETICAL ANALYSIS ON 1-LAYER TRANSFORMER

In this section, we explain the success of our approach by providing a theoretical analysis in Theorem 4.1. Specifically, we construct a Transformer model (whose normalization layer is omitted for simplicity) that is capable of solving the addition task involving both exponentially long operands and exponentially large number of operands when our approach is applied. Furthermore, the constructed model is a 1-layer 4-head Transformer, which supports the experimental results presented in Figure 10: the 1-layer 4-head model succeeds, but the 1-layer 2-head model fails.

**Theorem 4.1.** *With a proper input format, scratchpad, and Position Coupling, there exists a 1-layer 4-head decoder-only Transformer that solves the multi-operand integer addition task involving up to $m$ operands each with up to $n$ digits. Here, a sufficient choice of the embedding dimension is $d = \mathcal{O}(\log_2(m + 1) + \log_2(n + 1))$.*

Theorem 4.1 says that a 1-layer 4-head model is *enough* to solve integer additions involving exponentially many and exponentially long operands (in embedding dimension) when a proper scratchpad and Position Coupling are applied. As being a sufficiency result, it implies that larger architectures (with more layers and attention heads) can solve the same task as well. We put its detailed and constructive proof in Appendix D. We also remark that our theorem extends Theorem 5.1 of Cho et al. (2024), which can only handle addition problems with a fixed number of operands.

To illustrate our key idea, consider an example problem $057+048+096 = 000 \rightarrow 750 \rightarrow 501 \rightarrow 102$ (with the output reversed). First, consider the case without a scratchpad: $057 + 048 + 096 = 102$. To predict the least significant digit of the final answer, $1$, the model must attend to the least significant digits of all the operands, which are $7$, $8$, and $6$, in the query sequence. In a scenario with $m$ operands, the model would need to attend to $m$ digits. This property—the number of tokens the model has to attend to increases with the number of operands—makes the construction difficult.

With a scratchpad, the process becomes a lot simpler. Instead of attending to every least significant digit of all operands, the model only needs to attend to two tokens: e.g., $6$ from $096$ and $5$ from $501$. Importantly, by utilizing the intermediate states stored in the scratchpad, the number of tokens the model needs to attend to remains fixed, even if the number of operands gets larger.

**Scrutinizing the Attention Patterns of Trained Transformer.** Surprisingly, our insight into the advantage of our scratchpad and its synergy with Position Coupling can be visually verified, supporting the significance of our method and theoretical finding. We probe the attention matrices of transformer models trained with a practical Adam optimizer. We visualize the lower-triangular row-stochastic matrix $\texttt{softmax}(\boldsymbol{Q}\boldsymbol{K}^{\top})$ as a heatmap in Figure 7. Thus, if you want to know the distribution of softmax logits over key tokens for an NTP at the $q$-th query token, you should look at $q$-th *row* of the heatmap; the brighter the point, the more the model attends to that key token position.

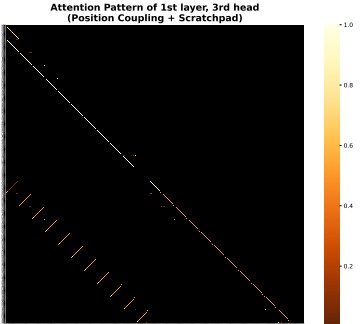 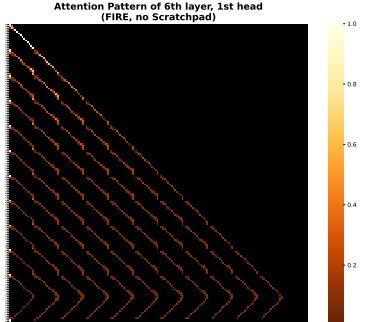

(a) 1-layer 4-head model trained with bi-level Position Coupling and Scratchpad.

(b) 6-layer 8-head model trained with FIRE but no scratchpad.

Figure 7: Extracted attention matrices from certain attention heads of trained Transformers on an addition dataset $\mathcal{S}_A(10, 10)$. We ran forward passes on 1,000 test samples from $\mathcal{T}_A(11, 11)$ and obtained average attention matrices. The softmax values below 0.01 are clipped out (black cells). We compared with the model without Position Coupling nor scratchpad to demonstrate that, without scratchpad, a model often try to "look at" every relevant positions at once, even without help of Position Coupling. We put more examples of attention patterns in Appendix E.

Now first look at the attention pattern extracted from a model trained with our scratchpad and bi-level Position Coupling (Figure 7a). Since the scratchpad takes up about half of the total input sequence length, we may focus on the bottom half of the heatmap. The attention pattern tells us that, for most of the NTP step, each attention head focuses on at most a fixed number of previous tokens: one on the short anti-diagonal line (which corresponds to the token in the query sequence) and one on the long diagonal line (which corresponds to the token in scratchpad). This observation is strikingly similar to our theoretical construction of the attention pattern.

On the other hand, let us move on to the attention pattern extracted from a model trained with FIRE but without scratchpad (Figure 7b). In this case, the length of the response is the same scale as a single operand, so you may focus on the last few rows of the heatmap. Unlike the previous case, and as we expected, the eleven anti-diagonal lines in the attention pattern reveal that the model actually focuses on as many previous token positions as the number of operands every time it performs NTP!

# 5 LENGTH GENERALIZATION ON INTEGER MULTIPLICATION TASK

Achieving length generalization for multiplication when both multiplier and multiplicand can vary in length has long been recognized as a challenging problem. To our knowledge, prior works on the multiplication task (Duan & Shi, 2023; Fan et al., 2024; Jelassi et al., 2023; McLeish et al., 2024) have never successfully addressed this issue. In this section, we demonstrate the combination of Position Coupling and the scratchpad can serve as a powerful solution to this obstacle.

## 5.1 METHOD: TWO-STAGE SCRATCHPAD & TRI-LEVEL POSITION COUPLING

```
        37  |
  *    925  |
       185  |  00185      Token :  37*925=581+470+333=58100>52900>52243
       074  |  00925      PosID1:  32000012341234123400000000000000000000
     +333   |  34225      PosID2:  00032111112222333311111122222333333
     34225  |             PosID3:  00000012342345345612345612345612345 6
         (a)                                    (b)
```

Figure 8: **(a)** A usual computation of integer multiplication, displaying the decomposition of multiplication task into two stages. **(b)** An example input sequence with a two-stage scratchpad and tri-level Position Coupling. The original example was "$37 \times 925 = 34225$": the query is inside a blue box and the response is inside a red box. All numbers in the scratchpad are reversed; all numbers in the whole sequence are minimally zero-padded to match the lengths of numbers in the same stage.

Different from the addition tasks, every digit of the first operand interacts with every digit of the second operand. This makes it difficult to come up with a single-stage cumulative scratchpad. To achieve strong length generalization in multiplication, we take a step beyond the simple cumulative scratchpads. We propose a *two-stage scratchpad*, motivated by an observation that the usual (human) computation of integer multiplication can be decomposed into two stages: (i) a series of $M$-digit $\times$ 1-digit multiplications and (ii) a (linearly shifted variant of) multi-operand addition. Figure 8b illustrates a concrete example of our two-stage scratchpad and tri-level position ID assignment rule, and Figure 8a explains the intuitive motivation of the scratchpad. Note that we concatenate two stages of scratchpad with a '=' token, which the model is required to infer as well as other digit/symbol tokens. For simplicity, let us write two operands as $A$ (with $M$ digits) and $B$ (with $N$ digits).

**Stage 1: $M$-digit$\times$1-digit Multiplications.** The first stage of the scratchpad consists of $N$ numbers: the first number is the product between $A$ and the least significant digit (LSD) of $B$, the second number is the product between $A$ and the second LSD of $B$, and so on. We reverse all the $N$ numbers, zero-pad them to match the length, and concatenate them in order with '+' tokens in between. Regarding the position ID assignment, observe that it is natural to (i) couple a $k$-th LSD of $A$ with $k$-th LSDs in every number of the scratchpad stage 1 and to (ii) couple the $k$-th LSD in $B$ with every digit in the $k$-th number of the scratchpad stage 1. This is reflected to `PosID1` and `PosID2` in Figure 8b, until the second '=' token.

**Stage 2: (Modified) Multi-Operand Addition.** The second stage is basically the same as a familiar multi-operand addition, with a slight difference in that we shift the operands to the left one by one as we add them sequentially. It can be done by viewing the LSD of the $k$-th number in stage 1 (i.e., the leftmost digit, as it is already reversed) as the $k$-th LSD when solving stage 2. This is semantically equivalent to converting "`581+470+333`" in the stage 1 into "`58100+04700+00333`". Because of this, we introduce a totally new level of position ID (level-3) that reflects this change of viewpoint on digit significance. Luckily, we can reuse the level-2 position IDs since they only distinguish the numbers. As a result, we expect the model to utilize level-3 and 2 of position IDs to solve the second stage. In this spirit, the position ID assignment rule is similar to that introduced in Section 4.1, reflected to `PosID2` and `PosID3` in Figure 8b. Lastly, we fill in unnecessary slots with 0's.

## 5.2 EXPERIMENTAL SETUP

**Data Sampling.** We denote the training set by $\mathcal{S}_M(n, m)$, consisting of multiplication problems where the first operand can have up to $n$ digits and the second operand can have up to $m$ digits. Specifically, the length of the first operand is uniformly selected from $[1 : n]$. The first operand is then randomly generated based on the chosen length. The second operand is chosen in a similar way, but its length is selected from $[1 : m]$. We use $\mathcal{S}_M(10, 10)$ as the baseline training set.

The test set $\mathcal{T}_M(n, m)$ is constructed in a similar manner, with the primary difference being that the lengths of both operands are strictly fixed at $n$ and $m$. For evaluation, we create a $30 \times 30$ grid heatmap and assess the performance of the model on the test set $\mathcal{T}_M(i, j)$ for every entry $(i, j) \in [1 : 30] \times [1 : 30]$.

**Model and Training.** We employ the same baseline architecture as in Section 4 (Refer to Table 3).

### 5.3 EXPERIMENTAL RESULTS

We evaluate models trained with 3 different position embedding methods, both with and without the scratchpad, and present the results in Figure 9. For models using Position Coupling without the scratchpad, we adopted a similar position ID assignment scheme proposed for solving $N$-digit $\times$ 2-digit multiplication in Cho et al. (2024). When the scratchpad is not applied, which corresponds to the top 3 plots, none of the models manage to generalize, even on in-distribution samples. When NoPE or FIRE is employed in conjunction with the scratchpad, the models show limited length generalization, achieving non-trivial accuracy on multiplication between two 12-digit integers. However, the combination of Position Coupling and the scratchpad again dominates others.

Interestingly, Position Coupling without the scratchpad shows weak length generalization when one of the operands is short. This should not come as a surprise, as Cho et al. (2024) already demonstrate that a model trained with Position Coupling alone can generalize to $N$-digit $\times$ 2-digit multiplication task: models trained on $N \leq 40$ can generalize to $N \geq 100$.

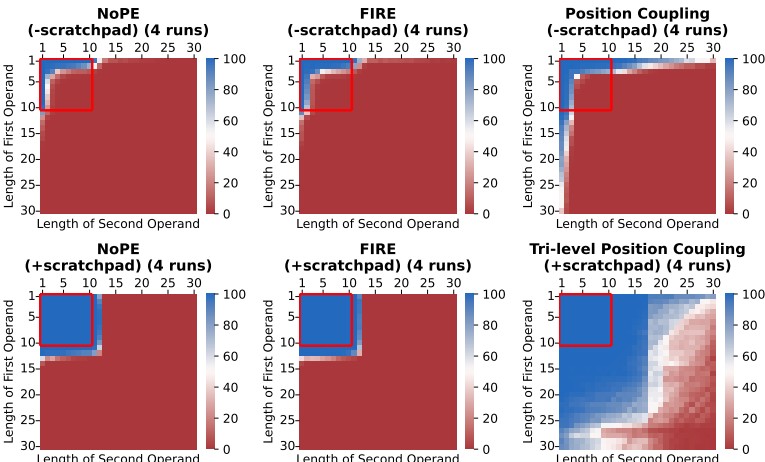

Figure 9: **Comparison of methods in the multiplication task.** We report exact-match accuracies for the multiplication task, with results taken as the median over 4 seeds for each method. Each axis represents the length of each operand. The red box indicates the training distribution, $\mathcal{S}_M(10, 10)$.

## 6 CONCLUSION

While length generalization in arithmetic Transformers has drawn a lot of attention, especially on operand length for the addition, the ability to generalize on operand counts is considered a difficult challenge and has not been explored yet. To address this challenge, we propose a combination of two techniques: scratchpad and Position Coupling. We show that a Transformer trained on problems involving 1–10 digit integers with 1–10 operands can solve addition tasks with up to 30 operands, each being as long as 30 digits. We also theoretically construct a 1-layer Transformer model capable of adding exponentially many operands with exponentially long integers when our approach is applied. Finally, we demonstrate the effectiveness of our approach for length generalization in the multiplication task, where both operand lengths can vary.

**Limitation.** One limitation of our approach is that it is only applicable to tasks whose structure is well-defined and can be effectively encoded by scratchpad and Position Coupling. This leaves us with two directions for future work. The first direction is to establish a clear principle for employing scratchpad and Position Coupling when the task structure is known, as the current design choice heavily relies on intuition. The second is to extend our method to the cases where the task structure is implicit or even entirely unknown.

ACKNOWLEDGMENTS

The authors sincerely thank Pranjal Awasthi and Anupam Gupta for their insightful discussions. This work was partly supported by a National Research Foundation of Korea (NRF) grant (No. RS-2024-00421203) and an Institute for Information & communications Technology Planning & Evaluation (IITP) grant (No. RS-2019-II190075, Artificial Intelligence Graduate School Program (KAIST)) funded by the Korean government (MSIT). HC, JC, and CY acknowledge support from a Google Gift on the research related to Long Context Transformers. The experiments contained in this work were supported in part through a Google Cloud Platform Credit Award.

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

## CONTENTS

# A   ADDITIONAL RELATED WORKS

## A.1   LENGTH GENERALIZATION IN ARITHMETIC/ALGORITHMIC TRANSFORMERS

Several works (Anil et al., 2022; Deletang et al., 2023; Nogueira et al., 2021) have scrutinized the Transformer's lack of ability to length generalize across algorithmic reasoning tasks. Here, we list the studies investigating and enhancing the length extrapolation capability of Transformers for arithmetic/algorithmic tasks.

**Addition Tasks.**   We first begin by exploring studies focused on encoder-only Transformers. Ruoss et al. (2023) introduce Randomized Position Encodings to address the problem of the appearance of unseen position indices in longer sequences. Additionally, Jelassi et al. (2023) demonstrate that the RPE method enables generalization to 15-digit addition problems when the model is trained on problems up to 5 digits.

We next move on to the studies considering decoder-only Transformers. Kazemnejad et al. (2023) investigate the effect of PE methods on length generalization performance, arguing that NoPE outperforms several popular PE methods such as ALiBi (Press et al., 2022), RoPE (Su et al., 2024), APE (Vaswani et al., 2017), and T5's Relative PE (Raffel et al., 2020). Meanwhile, Shen et al. (2023) propose the use of a scratchpad and random spacing, which facilitate generalization to 12-digit problems when trained on up to 10-digit problems. Zhou et al. (2024a) introduce the technique of "index-hinting", which inserts appropriate position markers in the sequence. Zhou et al. (2024b) integrate several existing techniques—FIRE (Li et al., 2024), index-hinting (Zhou et al., 2024a), and Randomized PE (Ruoss et al., 2023)—achieving generalization to 100-digit problems while training exclusively on samples with fewer than 40 digits. Furthermore, Cho et al. (2024) and McLeish et al. (2024) independently introduce Position Coupling (also called "Abacus"), demonstrating state-of-the-art performance in the literature by generalizing to 100-digit problems after training on samples with up to 20 digits. We lastly remark there is a recent attempt to solve addition by utilizing a looped transformer (Fan et al., 2024).

**Multiplication Tasks.**   Most studies on multiplication focus on problems where one operand has a fixed digit length. Jelassi et al. (2023) investigate $N$-digit$\times$3-digit multiplication but observe length generalization only when *train set priming* is applied, which involves adding a few long samples in the train set. Cho et al. (2024) present the effectiveness of Position Coupling in $N$-digit$\times$2-digit multiplication, achieving generalization to $N \geq 100$ after training on samples with $N \leq 40$. Fan et al. (2024) showcase the length generalization capability of looped Transformers to 16-digit problems from training up to 11 digits, where the numbers are encoded in binary format and the length of the first operand is up to 2.

McLeish et al. (2024) also train their models on multiplication tasks where both operand lengths can vary. They employ the "Abacus" embedding without scratchpad, and report near-perfect in-distribution accuracy but near-zero accuracy on out-of-distribution multiplication problems. A notable point is the success on in-distribution samples, whereas our results under a similar setting, shown in Figure 9 (upper rightmost plot), exhibit a completely different trend. We suspect that this discrepancy arises from differences in their architectural implementation details[6] compared to ours. One such major difference which we suspect as the main cause of the performance discrepancy is the implementation of attention (sub)layers. They used `torch.nn.MultiheadAttention` module predefined inside PyTorch (Paszke et al., 2019), whereas our code includes an explicit implementation of transformer attention layer with bunch of tensor multiplications (specifically based on a part of HuggingFace's T5 model implementation). We list a few other notable implementational details that are exclusively included in their codebase: the recurrent layers (i.e., looped Transformers), input injection technique, noise injection into the input embedding, and additional feed-forward layer between the last transformer block and the linear read-out layer.

There is also a body of literature focused on length generalization in algorithmic tasks. We highlight a few key contributions. Zhou et al. (2024a) introduce the concept of RASP-L, suggesting the conjecture of the Transformer's ability to length-generalize may depend on whether the task can be expressed in the RASP-L language. Additionally, Awasthi & Gupta (2023) create an auxiliary task

---

[6]`github.com/mcleish7/arithmetic`

associated with sorting, resulting in substantial length generalization improvements through multitask learning.

## A.2 CHAIN-OF-THOUGHTS PROMPTING

While the main focus of this work is on *training* the model with a scratchpad, *chain-of-thoughts* (CoT) prompting—showing a series of intermediate natural language reasoning steps before reaching the answer—is also extensively studied to enhance the reasoning ability of Transformers (Kojima et al., 2022; Suzgun et al., 2022; Wei et al., 2022). Similar to the spirit of scratchpad, CoT allows the model to decompose complex problems into several intermediate steps, which has demonstrated its importance in tasks that require arithmetic and reasoning.

In an attempt to understand the success of CoT, Feng et al. (2023) investigate the expressivity of CoT. They prove that the autoregressive Transformers of constant size can solve basic arithmetic/equation tasks, while finite-depth Transformer models cannot directly produce correct answers to these tasks unless their size grows super-polynomially with input length. In arithmetic tasks, their experiments reveal that models trained with CoT-formatted data can generalize to different numbers of operands, but the generalization leap is limited to 3 (from 15 to 18).

# B  EXPERIMENTAL DETAILS

We modify and customize the codebase from Kazemnejad et al. (2023) for all our experiments.[7] This codebase includes a custom implementation of a decoder-only T5 model (Raffel et al., 2020) built upon PyTorch (Paszke et al., 2019) and HuggingFace (Wolf et al., 2019), which incorporates several positional encoding methods.

We implemented a custom RMSNorm module (Zhang & Sennrich, 2019) and various normalization layer positioning schemes (e.g., PreNorm (Xiong et al., 2020), PostNorm (Vaswani et al., 2017)) to follow the implementation details outlined by Cho et al. (2024); Zhou et al. (2024b).

Below, we provide detailed settings of experiments.

Table 1: Hyperparameter summary for the parity task (Figure 3).

| Hyperparameter | Value |
| --- | --- |
| Architecture | Decoder-only Transformer |
| Number of Layers | 6 |
| Number of Attention Heads | 8 |
| Embedding Dimension | 512 |
| Dimension per Head | 64 |
| Hidden Width of Feed-forward Layer | 2048 |
| Activation Function of Feed-forward Layer | GEGLU (Shazeer, 2020) |
| Normalization Layer | RMSNorm (Zhang & Sennrich, 2019) |
| Normalization Layer Position | PreNorm and PostNorm |
| Trainable Parameter Count | 25M |
| Training Steps | 50,000 |
| Batch Size | 20 |
| Optimizer | Adam (Kingma & Ba, 2015) |
| Learning Rate (LR) | 0.00003 |
| LR Warm-up | Linear (From 0 to LR), 1% of total steps |
| LR Cool-down | Cosine Decay (From LR to 0.1LR) |
| Maximum Position ID (`max_pos`) | 101 |
| Training Dataset Size | 10,000 |
| Evaluation Dataset Size | 10,000 per query length |
| Device | NVIDIA RTX A6000 48GB |
| Training Time | $\leq$ 3 hours |

---

[7] github.com/McGill-NLP/length-generalization

Table 2: Hyperparameter summary for the addition task (Figure 1).

| Hyperparameter | Value |
|---|---|
| Architecture | Decoder-only Transformer |
| Number of Layers | 6 |
| Number of Attention Heads | 8 |
| Embedding Dimension | 1024 |
| Dimension per Head | 128 |
| Hidden Width of Feed-forward Layer | 2048 |
| Activation Function of Feed-forward Layer | GEGLU (Shazeer, 2020) |
| Normalization Layer | RMSNorm (Zhang & Sennrich, 2019) |
| Normalization Layer Position | PreNorm and PostNorm |
| Trainable Parameter Count | 63M |
| Training Steps | 50,000 |
| Batch Size | 400 |
| Optimizer | Adam (Kingma & Ba, 2015) |
| Learning Rate (LR) | 0.00003 |
| LR Warm-up | Linear (From 0 to LR), 1% of total steps |
| LR Cool-down | Cosine Decay (From LR to 0.1LR) |
| Maximum Position ID 1 (`max_pos_1`) | 40 |
| Maximum Position ID 2 (`max_pos_2`) | 40 |
| Training Dataset Size | 500,000 |
| Evaluation Dataset Size | 1,000 per operand length/count |
| Device | NVIDIA RTX A6000 48GB |
| Training Time | $\leq$ 12 hours |

Table 3: Hyperparameter summary for the multiplication task (Figure 9).

| Hyperparameter | Value |
|---|---|
| Architecture | Decoder-only Transformer |
| Number of Layers | 6 |
| Number of Attention Heads | 8 |
| Embedding Dimension | 1024 |
| Dimension per Head | 128 |
| Hidden Width of Feed-forward Layer | 2048 |
| Activation Function of Feed-forward Layer | GEGLU (Shazeer, 2020) |
| Normalization Layer | RMSNorm (Zhang & Sennrich, 2019) |
| Normalization Layer Position | PreNorm and PostNorm |
| Trainable Parameter Count | 63M |
| Training Steps | 50,000 |
| Batch Size | 200 |
| Optimizer | Adam (Kingma & Ba, 2015) |
| Learning Rate (LR) | 0.00005 |
| LR Warm-up | Linear (From 0 to LR), 1% of total steps |
| LR Cool-down | Cosine Decay (From LR to 0.1LR) |
| Maximum Position ID 1 (`max_pos_1`) | 64 |
| Maximum Position ID 2 (`max_pos_2`) | 32 |
| Maximum Position ID 3 (`max_pos_3`) | 64 |
| Training Dataset Size | 500000 |
| Evaluation Dataset Size | 1000 per length combination of first/second operands |
| Device | NVIDIA RTX A6000 48GB |
| Training Time | $\leq$ 12 hours |

## C  ADDITIONAL EXPERIMENTAL RESULTS

In this section, we provide additional experimental results not discussed in the main section.

We first present the results of experiments investigating how changes in model architecture (the number of layers and attention heads) affect task performance. We explore 12 different configurations. The heatmaps below represent the performance across different operand numbers and lengths, with blue regions indicating higher accuracy and red regions indicating lower accuracy.

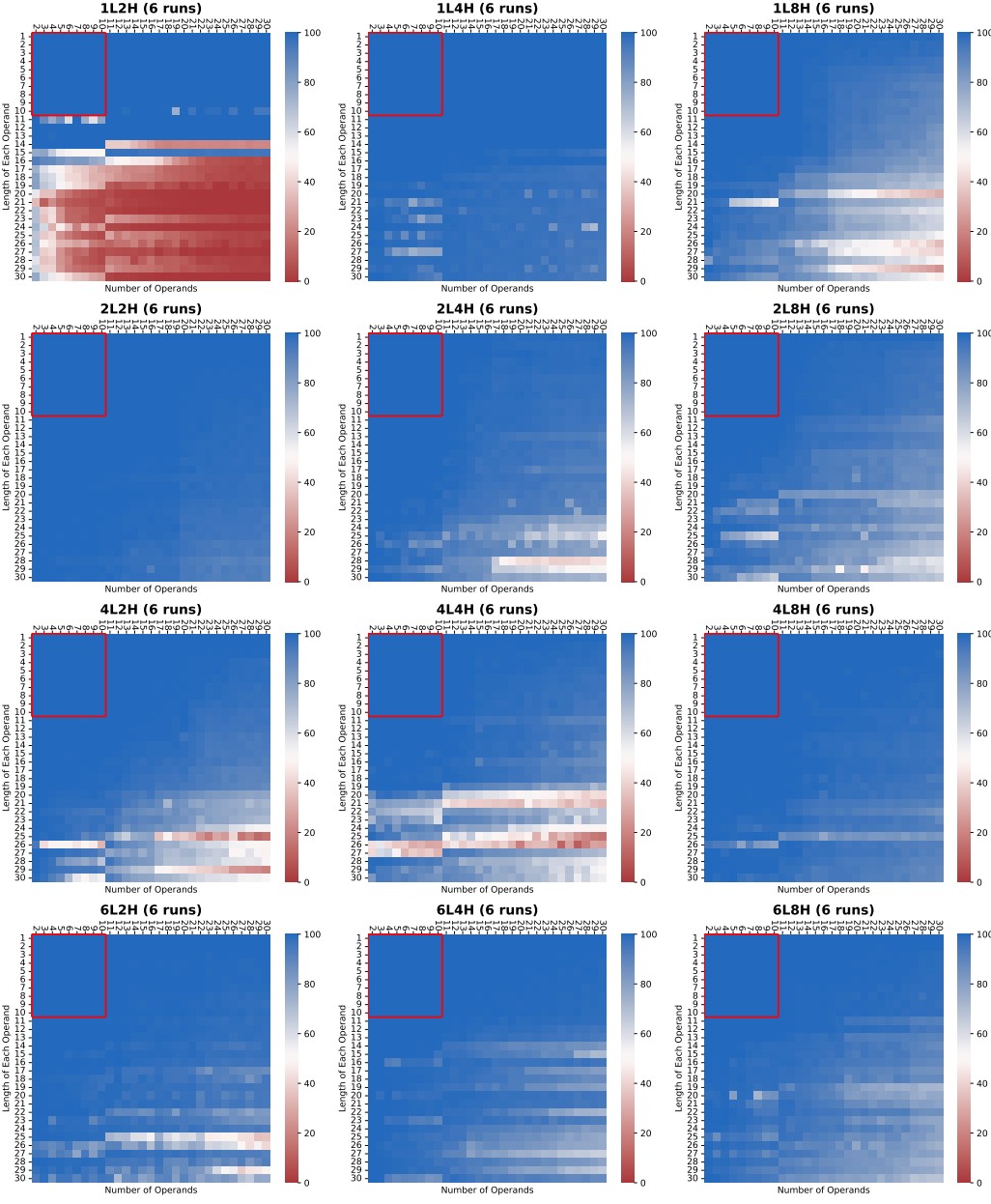

Figure 10: **Comparison of architectures in the integer addition task.** We report exact-match accuracies for the addition task, with results taken as the median over 6 seeds for each method. The x-axis corresponds to the number of operands, and the y-axis indicates the length of operands. The red box indicates the trained scope $\mathcal{S}_A(10, 10)$.

One might be concerned why 1L8H models perform worse than 1L4H models. To address this concern, we conduct experiments by controlling the total number of head dimensions. In Figure 10, since we fix the total number of head dimensions within a layer by 1024, each head in 1L4H models has 256 dimensions while each head in 1L8H models has only 128 dimensions. To make a fair comparison, we decoupled the embedding dimension from the head dimension, fixing the embedding dimension by 1024 for this setup only. The results are presented in Figure 11. We observe that the performance of 1L8H models improves when the dimension per head is increased to 256, while the performance of 1L4H models degrades when the dimension per head is decreased to 128. We conclude that the inferior performance of 1L8H models in Figure 10 is due to the reduced dimension per head. These findings suggest that a larger dimension per head is crucial for achieving strong performance in shallow models.

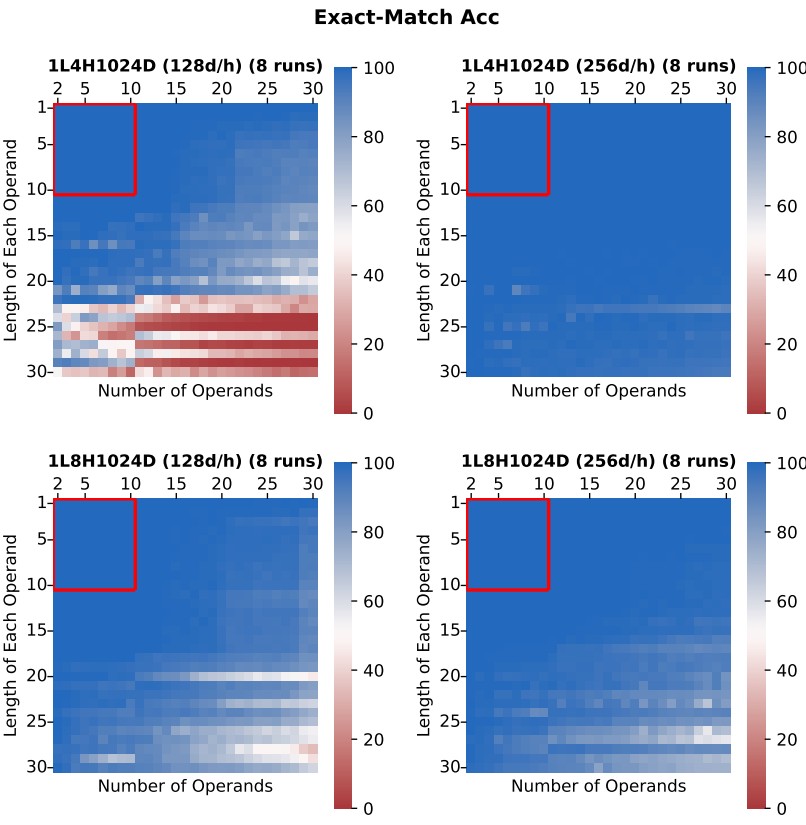

Figure 11: **Comparison of head dimensions in the integer addition task.** We report exact-match accuracies for the addition task, with results taken as the median over 8 seeds for each configuration. The x-axis corresponds to the number of operands, and the y-axis indicates the length of operands. The red box indicates the trained scope $\mathcal{S}_A(10, 10)$.

Next, we provide ablation results on varying embedding dimensions. We trained 1L4H models, each with embedding dimensions of 64, 128, 256, and 512 (with dimensions per head of 16, 32, 64, and 128, respectively). The results are illustrated in Figure 12. We observe that there is a significant performance gap between these configurations. While models with small embedding dimensions have sufficient expressive capacity for solving the task (Theorem 4.1), we believe larger embedding dimensions are crucial for enabling more effective optimization.

We now present the results of an ablation study on the training set size. Our baseline training data size is 500K, and we vary the size to 20K, 100K, 2M, and 10M. We fix the architecture as a 6-layer 8-head model with an embedding dimension of 1024. The results are illustrated in Figure 13. We observe that the training set size has no significant impact on the model's performance.

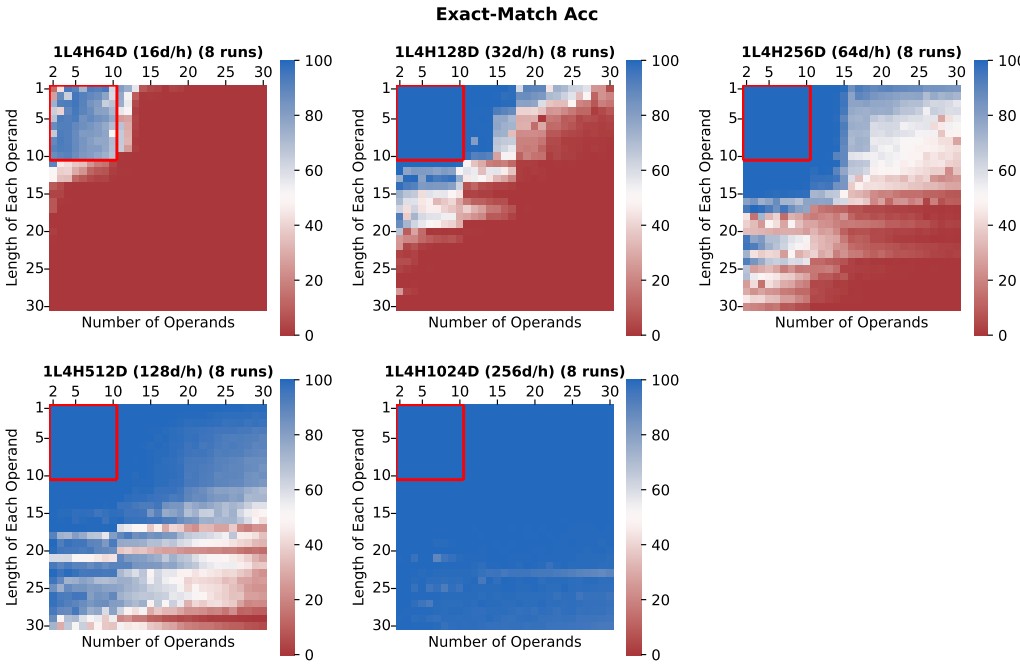

Figure 12: **Comparison of embedding dimensions in the integer addition task.** We report exact-match accuracies for the addition task, with results taken as the median over 8 seeds for each configuration. The x-axis corresponds to the number of operands, and the y-axis indicates the length of operands. The red box indicates the trained scope $\mathcal{S}_A(10, 10)$.

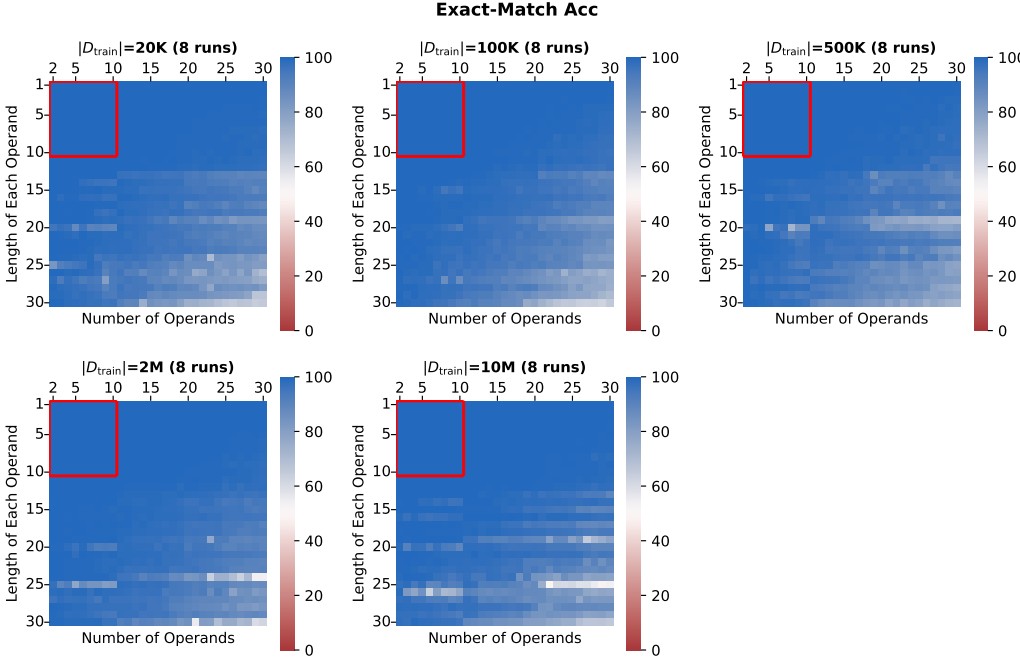

Figure 13: **Comparison of training set size in the integer addition task.** We report exact-match accuracies for the addition task, with results taken as the median over 8 seeds for each configuration. The x-axis corresponds to the number of operands, and the y-axis indicates the length of operands. The red box indicates the trained scope $\mathcal{S}_A(10, 10)$.

We lastly investigate the impact of input sequence formatting. We considered three factors: (1) reversed or plain query, (2) reversed or plain response, and (3) zero-padding or no zero-padding. We exclude the case where a reversed query and a plain response are used together, resulting in a total of 6 configurations. Note that our baseline format consists of a plain query, reversed response, and zero-padding. We fix the architecture as a 6-layer 8-head model with an embedding dimension of 1024. The results are illustrated in Figure 14. To clarify, the top 3 figures correspond to the zero-padding setting, while the bottom 3 figures correspond to the no zero-padding setting. From left to right, the figures represent: plain query with reversed response, reversed query with reversed response, and plain query with plain response.

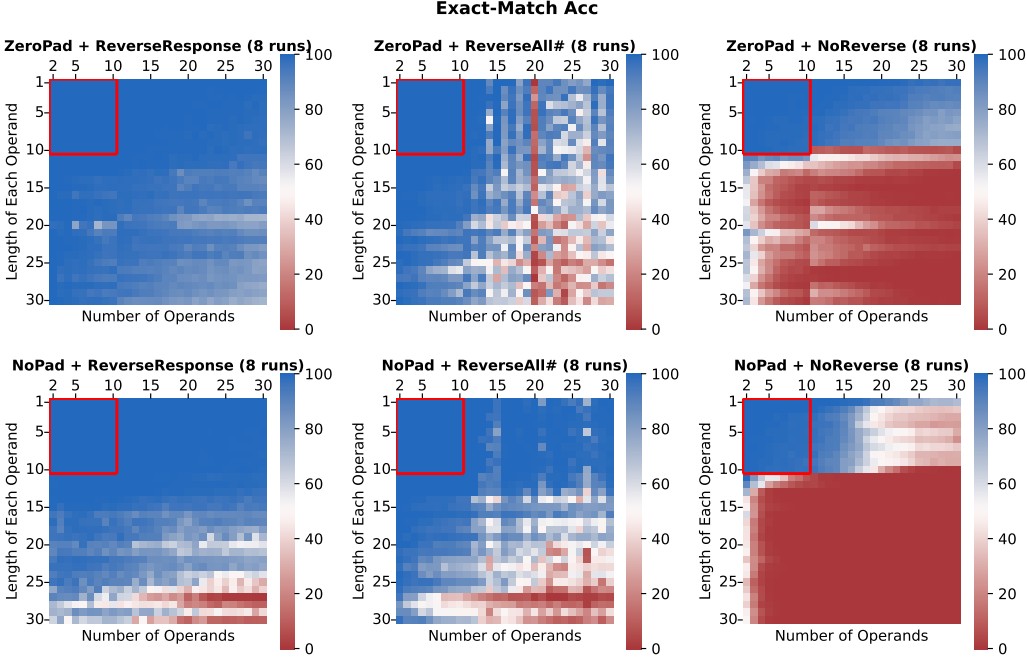

Figure 14: **Comparison of the input formats in the integer addition task.** We report exact-match accuracies for the addition task, with results taken as the median over 8 seeds for each configuration. The x-axis corresponds to the number of operands, and the y-axis indicates the length of operands. The red box indicates the trained scope $\mathcal{S}_A(10, 10)$.

# D  FORMAL CONSTRUCTION OF MULTI-OPERAND ADDITION TRANSFORMER

In this section, we prove Theorem 4.1 by formally constructing a 1-layer 4-head Transformer model capable of solving multi-operand addition problems. The proof framework mostly follows the proof idea of Theorem 5.1 in Cho et al. (2024). For the sake of readability, we restate our theorem statement.

**Theorem 4.1.** *With a proper input format, scratchpad, and Position Coupling, there exists a 1-layer 4-head decoder-only Transformer that solves the multi-operand integer addition task involving up to $m$ operands each with up to $n$ digits. Here, a sufficient choice of the embedding dimension is $d = \mathcal{O}(\log_2(m+1) + \log_2(n+1))$.*

## D.1  NOTATION

Let $e_i^d$ represent the $i$-th standard basis vector of $\mathbb{R}^d$. Define $\boldsymbol{I}_m$ as the identity matrix of size $m \times m$. The vectors $\mathbf{0}_p$ and $\mathbf{1}_p$ are $p$-dimensional vectors filled with zeros and ones, respectively. Let $\mathbf{0}_{m \times n}$ denote the $m \times n$ zero matrix. For $n \in \mathbb{N}$, we use $[n]$ to denote the set $\{1, ..., n\}$. For any matrix $\boldsymbol{A}$, we use $\boldsymbol{A}_{i\bullet}$ and $\boldsymbol{A}_{\bullet j}$ to refer to the $i$-th row and $j$-th column of $\boldsymbol{A}$, respectively.

Define an ordered vocabulary $\mathcal{V} = (0, 1, 2, 3, 4, 5, 6, 7, 8, 9, +, =, \rightarrow, \$)$. The special token '\$' represents both the beginning-of-sequence (BOS) token and the end-of-sequence (EOS) token. While BOS and EOS tokens do not have to be identical, we use a single symbol for simplicity. Let $\mathcal{V}_k$ represent the $k$-th element of $\mathcal{V}$.

## D.2  ARCHITECTURE

We adopt the same architecture as explained in the appendix of Cho et al. (2024). We direct the reader to Appendix D of their work for architectural specifications. In summary, we use a decoder-only Transformer with softmax operation, but omit the normalization layer for simplicity. Note that, since our construction involves a single-layer model, we omit the superscripts $(l)$ in the parameter matrices/vectors and the size of dimensions $d_{QK,h}^{(l)}$ and $d_{V,h}^{(l)}$ for simplicity.

## D.3  INPUT SEQUENCE

We aim to construct a decoder-only Transformer model capable of solving the addition $a_1 + a_2 + \cdots + a_m = b$ of $m$ operands whose lengths are at most $n$, where we regard every single digit as a single token. We employ the same input format illustrated in Figure 4. Now, we describe how to transform this addition problem into the input sequence $\mathcal{I}$, which will be fed to the Transformer.

We begin by introducing the scratchpad. Let $b_j := \sum_{i=1}^{j} a_i$ (for $\forall j \in [m]$) and $b_0 = 0$, representing the cumulative sum up to the $j$-th operand; thus, $b_m = b$. As in the experimental setup, the scratchpad contains every intermediate result ($\{b_i\}_{i=0}^{m}$) that arises during the addition process, with each result separated by an arrow ($\rightarrow$).

Next, we apply zero-padding to every number in $\{a_i\}_{i=1}^{m}$ and $\{b_i\}_{i=0}^{m}$ so that they have the equal length, $\ell$, which is determined by the maximum (possible) length among them. We set $\ell = n + 1 + \lfloor \log_{10} m \rfloor$ since the result of adding $m$ numbers each with $n$ digits can have a length at most $1 + \lfloor \log_{10}((10^n - 1)m) \rfloor \leq 1 + n + \lfloor \log_{10} m \rfloor$.

Also, we reverse the digits within each number in the response sequence ($\{b_i\}_{i=0}^{m}$), while keeping the numbers in the query ($\{a_i\}_{i=1}^{m}$) in their original order, which is consistent with the experimental setup.

To recap, the input sequence $\mathcal{I}$ can be formally written as $\overline{\sigma_1 \sigma_2 \cdots \sigma_N} \in \mathcal{V}^N$ of length $N = (2m+1)(\ell+1)$ consisting of the following parts:

1. BOS token $\sigma_1 = $ '\$';

2. $i$-th operand $A_i = \overline{\sigma_{(i-1)(\ell+1)+2} \cdots \sigma_{i(\ell+1)}}$ where $\sigma_j \in \{0, \ldots, 9\}$ (for $i \in [m]$, zero-padded $a_i$);

3. $i$-th addition symbol $\sigma_{i(\ell+1)+1} = $ '+' (for $i \in [m-1]$);

4. equality symbol $\sigma_{m(\ell+1)+1} = $ '=';

5. placeholder zeros $B_0 = \overline{\sigma_{m(\ell+1)+2} \cdots \sigma_{(m+1)(\ell+1)}} = \overline{00 \cdots 0}$ (zero-padded $b_0$);

6. $i$-th arrow symbol $\sigma_{(m+i)(\ell+1)+1} = $ '$\rightarrow$' (for $i \in [m]$);

7. $i$-th (reversed) intermediate step $B_i = \overline{\sigma_{(m+i)(\ell+1)+2} \cdots \sigma_{(m+i+1)(\ell+1)}}$ where $\sigma_j \in \{0, \ldots, 9\}$ (for $i \in [m]$, reversed & zero-padded $b_i$).

We call $\overline{\sigma_1 \cdots \sigma_{m(\ell+1)}}$ by the query sequence, and $\overline{\sigma_{m(\ell+1)+2} \cdots}$ by the response sequence. We note that during the inference process, the response sequence might be incomplete (i.e., $N < (2m+1)(\ell+1)$), as each digit in these components will be inferred one by one using the next-token prediction mechanism. However, in this section on formal construction, we focus on the training setup, where we infer all the digits of the response sequence simultaneously in a single forward pass via next-token prediction. Specifically, we aim to use an input sequence $\mathcal{I} = \overline{\sigma_1 \cdots \sigma_N}$ to produce an output sequence $\mathcal{O} = \overline{\sigma'_1 \cdots \sigma'_N}$, where $\overline{\sigma'_{m(\ell+1)+1} \cdots \sigma'_{N-1}}$ is identical to $\overline{\sigma_{m(\ell+1)+2} \cdots \sigma_N}$ and $\sigma'_N = $ '$\$$' (EOS).

We summarize this process with an example. Given an addition problem $57 + 48 + 96 = 201$, the transformed input sequence $\mathcal{I}$ becomes $\overline{\$057 + 048 + 096 = 000 \rightarrow 750 \rightarrow 501 \rightarrow 102}$, with $m = 3$, $n = 2$, $\ell = 3$, and $N = 28$. The goal is to generate an output sequence $\mathcal{O}$ that ends with $\overline{000 \rightarrow 750 \rightarrow 501 \rightarrow 102\$}$.

### D.4 ENCODING FUNCTION

We now define the encoding function, which maps the input sequence $\mathcal{I} \in \mathcal{V}^N$ to the initial embedding matrix $\boldsymbol{X}^{(0)} \in \mathbb{R}^{d \times N}$. In this representation, each column corresponds to the embedding vector of an individual token. The embedding matrix $\boldsymbol{X}^{(0)}$ is constructed by concatenating the token embedding matrix and the position embedding (PE) matrix. We note that this construction can also be interpreted as the element-wise sum of these two different embedding matrices.

An example of the embedding matrix is presented in Table 4. The first 19 rows correspond to the token embedding matrix, while the subsequent $(2P_1 + 2P_2)$ rows represent the PE matrix. We will use this example throughout this appendix to visualize our construction with tables.

#### D.4.1 TOKEN EMBEDDING

The token embedding consists of 19 dimensions, which we will refer to by the following names for clarity:

$$1=\text{NUM}, 2=\text{IS\_BOS}, 3=\text{FULL\_ONES}, 4=\text{PRE\_SUM}, 5=\text{PRE\_CARRY},$$
$$6=\text{PRE\_ARROW}, 7=\text{PRE\_EOS}, \{8,\ldots,17\}=\text{SUM}, 18=\text{ARROW}, \text{and } 19=\text{EOS}.$$

To enhance readability, we will refer to each dimension by its corresponding name rather than by its index.

Initially, the last 16 dimensions are set to zero, while the first three dimensions are filled with their corresponding values. Below, we explain how the values for NUM, IS\_BOS, and FULL\_ONES are determined:

**Dimension 1 (NUM).** If the token is a digit $(0, \ldots, 9)$, we fill the dimension NUM by the token's value. Otherwise, for tokens $+$, $=$, $\rightarrow$, and $\$$, we put zero.

**Dimension 2 (IS\_BOS).** If the token is the BOS token ('$\$$'), we put 1 to the dimension IS\_BOS. Otherwise, we put zero.

**Dimension 3 (FULL\_ONES).** The dimension FULL\_ONES is set to 1 for every token.

#### D.4.2 COUPLED POSITION IDS AND POSITION EMBEDDING

In this section, we explain how the PE vector is determined for each token. As in our experiment, we first assign two position IDs $p_1(k)$ and $p_2(k)$ for each token $\sigma_k$. Then, we map each position ID to a

Table 4: Example initial encoding. We consider $\overline{\$057 + 048 + 096 = 000 \to 750 \to 501 \to 102}$ as an input sequence and the position ID offsets are chosen by 4 (for the first PE module) and 2 (for the second PE module). We denote dimensions for PE vectors as POS_1=$\{20,\ldots,P_1 + 19\}$, POS_1_NEXT=$\{P_1 + 20,\ldots,2P_1 + 19\}$, POS_2=$\{2P_1 + 20,\ldots,2P_1 + P_2 + 19\}$, and POS_2_NEXT=$\{2P_1 + P_2 + 20,\ldots,2P_1 + 2P_2 + 19\}$. The vectors of the form $\boldsymbol{v}_k^D$ are defined in Equation (3). The gray cells will be filled in later.

| $\mathcal{I}$ | $ | 0 | 5 | 7 | + | 0 | 4 | 8 | + | 0 | 9 | 6 | = | $\cdots$ |
|---|---|---|---|---|---|---|---|---|---|---|---|---|---|---|
| $p_1(\cdot)$ | 0 | 7 | 6 | 5 | 4 | 7 | 6 | 5 | 4 | 7 | 6 | 5 | 4 | $\cdots$ |
| $p_2(\cdot)$ | 0 | 2 | 2 | 2 | 2 | 3 | 3 | 3 | 3 | 4 | 4 | 4 | 2 | $\cdots$ |
| 1: NUM | 0 | 0 | 5 | 7 | 0 | 0 | 4 | 8 | 0 | 0 | 9 | 6 | 0 | |
| 2: IS_BOS | 1 | 0 | 0 | 0 | 0 | 0 | 0 | 0 | 0 | 0 | 0 | 0 | 0 | |
| 3: FULL_ONES | 1 | 1 | 1 | 1 | 1 | 1 | 1 | 1 | 1 | 1 | 1 | 1 | 1 | |
| 4: PRE_SUM | 0 | 0 | 0 | 0 | 0 | 0 | 0 | 0 | 0 | 0 | 0 | 0 | 0 | |
| 5: PRE_CARRY | 0 | 0 | 0 | 0 | 0 | 0 | 0 | 0 | 0 | 0 | 0 | 0 | 0 | |
| 6: PRE_ARROW | 0 | 0 | 0 | 0 | 0 | 0 | 0 | 0 | 0 | 0 | 0 | 0 | 0 | |
| 7: PRE_EOS | 0 | 0 | 0 | 0 | 0 | 0 | 0 | 0 | 0 | 0 | 0 | 0 | 0 | |
| 8-17: SUM | $\mathbf{0}_{10}$ | $\mathbf{0}_{10}$ | $\mathbf{0}_{10}$ | $\mathbf{0}_{10}$ | $\mathbf{0}_{10}$ | $\mathbf{0}_{10}$ | $\mathbf{0}_{10}$ | $\mathbf{0}_{10}$ | $\mathbf{0}_{10}$ | $\mathbf{0}_{10}$ | $\mathbf{0}_{10}$ | $\mathbf{0}_{10}$ | $\mathbf{0}_{10}$ | |
| 18: ARROW | 0 | 0 | 0 | 0 | 0 | 0 | 0 | 0 | 0 | 0 | 0 | 0 | 0 | |
| 19: EOS | 0 | 0 | 0 | 0 | 0 | 0 | 0 | 0 | 0 | 0 | 0 | 0 | 0 | |
| POS_1 | $\mathbf{0}_{P_1}$ | $\boldsymbol{v}_7^{P_1}$ | $\boldsymbol{v}_6^{P_1}$ | $\boldsymbol{v}_5^{P_1}$ | $\boldsymbol{v}_4^{P_1}$ | $\boldsymbol{v}_7^{P_1}$ | $\boldsymbol{v}_6^{P_1}$ | $\boldsymbol{v}_5^{P_1}$ | $\boldsymbol{v}_4^{P_1}$ | $\boldsymbol{v}_7^{P_1}$ | $\boldsymbol{v}_6^{P_1}$ | $\boldsymbol{v}_5^{P_1}$ | $\boldsymbol{v}_4^{P_1}$ | |
| POS_1_NEXT | $\mathbf{0}_{P_1}$ | $\boldsymbol{v}_8^{P_1}$ | $\boldsymbol{v}_7^{P_1}$ | $\boldsymbol{v}_6^{P_1}$ | $\boldsymbol{v}_5^{P_1}$ | $\boldsymbol{v}_8^{P_1}$ | $\boldsymbol{v}_7^{P_1}$ | $\boldsymbol{v}_6^{P_1}$ | $\boldsymbol{v}_5^{P_1}$ | $\boldsymbol{v}_8^{P_1}$ | $\boldsymbol{v}_7^{P_1}$ | $\boldsymbol{v}_6^{P_1}$ | $\boldsymbol{v}_5^{P_1}$ | |
| POS_2 | $\mathbf{0}_{P_2}$ | $\boldsymbol{v}_2^{P_2}$ | $\boldsymbol{v}_2^{P_2}$ | $\boldsymbol{v}_2^{P_2}$ | $\boldsymbol{v}_2^{P_2}$ | $\boldsymbol{v}_3^{P_2}$ | $\boldsymbol{v}_3^{P_2}$ | $\boldsymbol{v}_3^{P_2}$ | $\boldsymbol{v}_3^{P_2}$ | $\boldsymbol{v}_4^{P_2}$ | $\boldsymbol{v}_4^{P_2}$ | $\boldsymbol{v}_4^{P_2}$ | $\boldsymbol{v}_2^{P_2}$ | |
| POS_2_NEXT | $\mathbf{0}_{P_2}$ | $\boldsymbol{v}_3^{P_2}$ | $\boldsymbol{v}_3^{P_2}$ | $\boldsymbol{v}_3^{P_2}$ | $\boldsymbol{v}_3^{P_2}$ | $\boldsymbol{v}_4^{P_2}$ | $\boldsymbol{v}_4^{P_2}$ | $\boldsymbol{v}_4^{P_2}$ | $\boldsymbol{v}_4^{P_2}$ | $\boldsymbol{v}_5^{P_2}$ | $\boldsymbol{v}_5^{P_2}$ | $\boldsymbol{v}_5^{P_2}$ | $\boldsymbol{v}_3^{P_2}$ | |

| $\mathcal{I}$ | 0 | 0 | 0 | $\to$ | 7 | 5 | 0 | $\to$ | 5 | 0 | 1 | $\to$ | 1 | 0 | 2 |
|---|---|---|---|---|---|---|---|---|---|---|---|---|---|---|---|
| $p_1(\cdot)$ | 5 | 6 | 7 | 4 | 5 | 6 | 7 | 4 | 5 | 6 | 7 | 4 | 5 | 6 | 7 |
| $p_2(\cdot)$ | 2 | 2 | 2 | 3 | 3 | 3 | 3 | 4 | 4 | 4 | 4 | 5 | 5 | 5 | 5 |
| 1: NUM | 0 | 0 | 0 | 0 | 7 | 5 | 0 | 0 | 5 | 0 | 1 | 0 | 1 | 0 | 2 |
| 2: IS_BOS | 0 | 0 | 0 | 0 | 0 | 0 | 0 | 0 | 0 | 0 | 0 | 0 | 0 | 0 | 0 |
| 3: FULL_ONES | 1 | 1 | 1 | 1 | 1 | 1 | 1 | 1 | 1 | 1 | 1 | 1 | 1 | 1 | 1 |
| 4: PRE_SUM | 0 | 0 | 0 | 0 | 0 | 0 | 0 | 0 | 0 | 0 | 0 | 0 | 0 | 0 | 0 |
| 5: PRE_CARRY | 0 | 0 | 0 | 0 | 0 | 0 | 0 | 0 | 0 | 0 | 0 | 0 | 0 | 0 | 0 |
| 6: PRE_ARROW | 0 | 0 | 0 | 0 | 0 | 0 | 0 | 0 | 0 | 0 | 0 | 0 | 0 | 0 | 0 |
| 7: PRE_EOS | 0 | 0 | 0 | 0 | 0 | 0 | 0 | 0 | 0 | 0 | 0 | 0 | 0 | 0 | 0 |
| 8-17: SUM | $\mathbf{0}_{10}$ | $\mathbf{0}_{10}$ | $\mathbf{0}_{10}$ | $\mathbf{0}_{10}$ | $\mathbf{0}_{10}$ | $\mathbf{0}_{10}$ | $\mathbf{0}_{10}$ | $\mathbf{0}_{10}$ | $\mathbf{0}_{10}$ | $\mathbf{0}_{10}$ | $\mathbf{0}_{10}$ | $\mathbf{0}_{10}$ | $\mathbf{0}_{10}$ | $\mathbf{0}_{10}$ | $\mathbf{0}_{10}$ |
| 18: ARROW | 0 | 0 | 0 | 0 | 0 | 0 | 0 | 0 | 0 | 0 | 0 | 0 | 0 | 0 | 0 |
| 19: EOS | 0 | 0 | 0 | 0 | 0 | 0 | 0 | 0 | 0 | 0 | 0 | 0 | 0 | 0 | 0 |
| POS_1 | $\boldsymbol{v}_5^{P_1}$ | $\boldsymbol{v}_6^{P_1}$ | $\boldsymbol{v}_7^{P_1}$ | $\boldsymbol{v}_4^{P_1}$ | $\boldsymbol{v}_5^{P_1}$ | $\boldsymbol{v}_6^{P_1}$ | $\boldsymbol{v}_7^{P_1}$ | $\boldsymbol{v}_4^{P_1}$ | $\boldsymbol{v}_5^{P_1}$ | $\boldsymbol{v}_6^{P_1}$ | $\boldsymbol{v}_7^{P_1}$ | $\boldsymbol{v}_4^{P_1}$ | $\boldsymbol{v}_5^{P_1}$ | $\boldsymbol{v}_6^{P_1}$ | $\boldsymbol{v}_7^{P_1}$ |
| POS_1_NEXT | $\boldsymbol{v}_6^{P_1}$ | $\boldsymbol{v}_7^{P_1}$ | $\boldsymbol{v}_8^{P_1}$ | $\boldsymbol{v}_5^{P_1}$ | $\boldsymbol{v}_6^{P_1}$ | $\boldsymbol{v}_7^{P_1}$ | $\boldsymbol{v}_8^{P_1}$ | $\boldsymbol{v}_5^{P_1}$ | $\boldsymbol{v}_6^{P_1}$ | $\boldsymbol{v}_7^{P_1}$ | $\boldsymbol{v}_8^{P_1}$ | $\boldsymbol{v}_5^{P_1}$ | $\boldsymbol{v}_6^{P_1}$ | $\boldsymbol{v}_7^{P_1}$ | $\boldsymbol{v}_8^{P_1}$ |
| POS_2 | $\boldsymbol{v}_2^{P_2}$ | $\boldsymbol{v}_2^{P_2}$ | $\boldsymbol{v}_2^{P_2}$ | $\boldsymbol{v}_3^{P_2}$ | $\boldsymbol{v}_3^{P_2}$ | $\boldsymbol{v}_3^{P_2}$ | $\boldsymbol{v}_3^{P_2}$ | $\boldsymbol{v}_4^{P_2}$ | $\boldsymbol{v}_4^{P_2}$ | $\boldsymbol{v}_4^{P_2}$ | $\boldsymbol{v}_4^{P_2}$ | $\boldsymbol{v}_5^{P_2}$ | $\boldsymbol{v}_5^{P_2}$ | $\boldsymbol{v}_5^{P_2}$ | $\boldsymbol{v}_5^{P_2}$ |
| POS_2_NEXT | $\boldsymbol{v}_3^{P_2}$ | $\boldsymbol{v}_3^{P_2}$ | $\boldsymbol{v}_3^{P_2}$ | $\boldsymbol{v}_4^{P_2}$ | $\boldsymbol{v}_4^{P_2}$ | $\boldsymbol{v}_4^{P_2}$ | $\boldsymbol{v}_4^{P_2}$ | $\boldsymbol{v}_5^{P_2}$ | $\boldsymbol{v}_5^{P_2}$ | $\boldsymbol{v}_5^{P_2}$ | $\boldsymbol{v}_5^{P_2}$ | $\boldsymbol{v}_6^{P_2}$ | $\boldsymbol{v}_6^{P_2}$ | $\boldsymbol{v}_6^{P_2}$ | $\boldsymbol{v}_6^{P_2}$ |

PE vector: from $p_i(k)$ we map a $2P_i$-dimensional vector ($i = 1, 2$). We use two different embedding modules to map position IDs and then concatenate them.

We begin by assigning two position IDs to each token. We introduce two hyperparameters, $\texttt{max\_pos}_1(\geq \ell + 1)$ and $\texttt{max\_pos}_2(\geq m + 1)$, which set the maximum possible position ID for the first and the second PE modules, respectively. Then, we select a *position offset* for each module: $s_1 \in [\texttt{max\_pos}_1 - \ell]$ and $s_2 \in [\texttt{max\_pos}_2 - m]$. The position IDs, $p_1(k)$ from the first PE module and $p_2(k)$ from the second PE module, are assigned to each token $\sigma_k$ in the input sequence $\mathcal{I}$ as follows:

$$p_1(k)=\begin{cases} 0, & k=1, \text{ (corresponding to `\$' token)} \\ s_1+\{i(\ell+1)+1\} - k, & k=(i-1)(\ell+1)+2,\ldots,i(\ell+1)+1 \text{ for } i \in [m], \\ s_1-\{(m+i-1)(\ell+1)+1\}+k, & k=(m+i-1)(\ell+1)+1,\ldots,(m+i)(\ell+1) \text{ for } i \in [m+1], \end{cases}$$
(1)

and

$$p_2(k) = \begin{cases} 0, & k = 1, \text{ (corresponding to `\$' token)} \\ s_2 + \left\lfloor \dfrac{k-2}{\ell+1} \right\rfloor, & k = 2, \ldots, m(\ell+1), \\ s_2 + \left\lfloor \dfrac{k - m(\ell+1) - 1}{\ell+1} \right\rfloor, & k = m(\ell+1) + 1, \ldots, (2m+1)(\ell+1). \end{cases} \tag{2}$$

Due to the complexity of the formal expressions, we encourage readers to refer to the dimensions POS_1 and POS_2 in Table 4 for concrete examples.

Next, we explain the design of the PE vector for each position ID. We define $b_i^{(D,k)}$ as the $i$-th (from left) digit of $D$-digit binary representation of $k-1$. The vector $\boldsymbol{v}_k^D$ ($k \in [2^D]$) is defined as:

$$\boldsymbol{v}_k^D = \left[ (-1)^{b_i^{(D,k)}} \right]_{i=1}^D \in \mathbb{R}^D. \tag{3}$$

This can be interpreted as a vertex of $D$-dimensional hypercube $[-1,1]^D$. Importantly, for $k \neq l$,

$$\left\| \boldsymbol{v}_k^D \right\|^2 = D, \quad \left\langle \boldsymbol{v}_k^D, \boldsymbol{v}_l^D \right\rangle \leq D - 2 \tag{4}$$

hold. This property will later be utilized in the construction of the attention layer.

We set a PE vector for the level-1 position ID $p_1(i) = 0$ (indicating a BOS token) as $\boldsymbol{0}_{2P_1}$. For cases where $p_1(i) > 0$, the level-1 PE vector is defined as:

$$\begin{bmatrix} \boldsymbol{v}_{p_1(i)}^{P_1} \\ \boldsymbol{v}_{p_1(i)+1}^{P_1} \end{bmatrix} \in \mathbb{R}^{2P_1}, \tag{5}$$

but we use $\boldsymbol{v}_1^{P_1}$ instead of $\boldsymbol{v}_{p_1(i)+1}^{P_1}$ when $p_1(i) = 2^{P_1}$.

Similarly, the second PE module is defined such that the PE vector is set to $\boldsymbol{0}_{2P_2}$ if $p_2(i) = 0$; otherwise, it is defined as:

$$\begin{bmatrix} \boldsymbol{v}_{p_2(i)}^{P_2} \\ \boldsymbol{v}_{p_2(i)+1}^{P_2} \end{bmatrix} \in \mathbb{R}^{2P_2}, \tag{6}$$

but we use $\boldsymbol{v}_1^{P_2}$ instead of $\boldsymbol{v}_{p_2(i)+1}^{P_2}$ when $p_2(i) = 2^{P_2}$.

Recall that the format $\boldsymbol{v}_k^D$ can represent $2^D$ distinct directions. So, we can set the hyperparameters for maximum position IDs as $\texttt{max\_pos}_1 \leq 2^{P_1}$ and $\texttt{max\_pos}_2 \leq 2^{P_2}$. Also, recall that $\ell + 1 \leq \texttt{max\_pos}_1$, $m + 1 \leq \texttt{max\_pos}_2$, and $\ell = n + 1 + \lfloor \log_{10} m \rfloor$. Thus, it suffices to choose $P_1 \geq \log_2 (n + 1 + \lfloor \log_{10} m \rfloor)$ and $P_2 \geq \log_2(m + 1)$. Since $d = 2(P_1 + P_2) + 19$, a sufficient choice of embedding dimension is

$$\begin{aligned} d &= 2 \left\lceil \log_2 (n + 1 + \lfloor \log_{10} m \rfloor) \right\rceil + 2 \left\lceil \log_2(m+1) \right\rceil + 19 \\ &= \mathcal{O} \left( \log_2(n+1) + \log_2(m+1) \right). \end{aligned}$$

In the next section, we present the construction of the model and demonstrate its capability to solve the multiple operand addition problem when the above conditions are met.

## D.5 TRANSFORMER BLOCK: CAUSAL ATTENTION LAYER

We now introduce the construction of the Transformer block, which utilizes 4 attention heads. The output from these heads are recorded to the dimensions PRE_SUM, PRE_CARRY, PRE_ARROW, and PRE_EOS. The feed-forward layer will then process these 4 dimensions and store the results in dimensions SUM and EOS. Finally, the linear readout at the final layer uses these outputs to predict the next token in the sequence.

### D.5.1 Attention Head 1: Digit-wise Addition without Carries

The goal of the first attention head is to perform digit-wise addition between two single-digit integers and store the result in the dimension PRE_SUM. Recall that $b_j$, defined as $\sum_{i=1}^{j} a_i$, can be computed by adding $a_j$ to $b_{j-1}$. This means that to predict a digit in $b_j$, the model needs to attend to two tokens: the first token (placed in $a_j$) which is in the query sequence, and the second token (placed in $b_{j-1}$) which is in the response sequence. Thus, we aim to design the query weight matrix $Q_1$ and the key weight matrix $K_1$ that generates an attention pattern satisfying this requirement.

Let $P = P_1 + P_2$ and recall that $d = 2P + 19$. Let the dimension of the first attention head be $d_{QK,1} = P + 1$. We define the query matrix $Q_1$ and the key matrix $K_1$ as follows:

$$Q_1 = \begin{pmatrix} \mathbf{0}_{P_1 \times 19} & \mathbf{0}_{P_1 \times P_1} & \sqrt{\alpha_1}I_{P_1} & \mathbf{0}_{P_1 \times P_2} & \mathbf{0}_{P_1 \times P_2} \\ \mathbf{0}_{P_2 \times 19} & \mathbf{0}_{P_2 \times P_1} & \mathbf{0}_{P_2 \times P_1} & \sqrt{\alpha_1}I_{P_2} & \mathbf{0}_{P_2 \times P_2} \\ \sqrt{\alpha_1 P}(e_{\text{FULL\_ONES}}^{19})^\top & \mathbf{0}_{1 \times P_1} & \mathbf{0}_{1 \times P_1} & \mathbf{0}_{1 \times P_2} & \mathbf{0}_{1 \times P_2} \end{pmatrix} \in \mathbb{R}^{d_{QK,1} \times d}, \quad (7)$$

$$K_1 = \begin{pmatrix} \mathbf{0}_{P_1 \times 19} & \sqrt{\alpha_1}I_{P_1} & \mathbf{0}_{P_1 \times P_1} & \mathbf{0}_{P_1 \times P_2} & \mathbf{0}_{P_1 \times P_2} \\ \mathbf{0}_{P_2 \times 19} & \mathbf{0}_{P_2 \times P_1} & \mathbf{0}_{P_2 \times P_1} & \mathbf{0}_{P_2 \times P_2} & \sqrt{\alpha_1}I_{P_2} \\ \sqrt{\alpha_1 P}(e_{\text{IS\_BOS}}^{19})^\top & \mathbf{0}_{1 \times P_1} & \mathbf{0}_{1 \times P_1} & \mathbf{0}_{1 \times P_2} & \mathbf{0}_{1 \times P_2} \end{pmatrix} \in \mathbb{R}^{d_{QK,1} \times d}, \quad (8)$$

where $\alpha_1$ is a scaling factor, which we can choose to be arbitrarily large. Also, let $d_{V,1} = 1$ and define

$$V_1 = 3(e_{\text{NUM}}^d)^\top \in \mathbb{R}^{d_{V,1} \times d}, \quad (9)$$

$$U_1 = e_{\text{PRE\_SUM}}^d \in \mathbb{R}^{d \times d_{V,1}}. \quad (10)$$

For better understanding, we explain the structure of $Q_1 X^{(0)} \in \mathbb{R}^{d_{QK,1} \times N}$. The block $\sqrt{\alpha_1}I_{P_1}$ in the first row of $Q_1$ extracts the dimensions POS_1_NEXT from $X^{(0)}$, and scales them by $\sqrt{\alpha_1}$. Similarly, the block $\sqrt{\alpha_1}I_{P_2}$ in the second row of $Q_1$ selects the dimensions POS_2 from $X^{(0)}$ and scales them by $\sqrt{\alpha_1}$. Lastly, the block $\sqrt{\alpha_1 P}(e_{\text{FULL\_ONES}}^{19})^\top$ in the third row of $Q_1$ takes the dimension FULL_ONES from $X^{(0)}$ and scales it by $\sqrt{\alpha_1 P}$. $Q_1 X^{(0)}$ is a concatenation of these three matrices. The computation of $K_1 X^{(0)}$ follows similar steps, but it operates on the dimensions POS_1, POS_2_NEXT, and IS_BOS. For $U_1 V_1 X^{(0)} \in \mathbb{R}^{d \times N}$, it simply copies the dimension NUM, scales it by 3, and shifts it to the dimension PRE_SUM. An example of $Q_1 X^{(0)}$, $K_1 X^{(0)}$, and $U_1 V_1 X^{(0)}$ is illustrated in Tables 5 to 7.

Table 5: Example of $\frac{1}{\sqrt{\alpha_1}} Q_1 X^{(0)}$, continuing from Table 4.

| $\mathcal{I}$ | \$ | 0 | 5 | 7 | + | 0 | 4 | 8 | + | 0 | 9 | 6 | = | $\cdots$ |
|---|---|---|---|---|---|---|---|---|---|---|---|---|---|---|
| $1$–$P_1$: | $\mathbf{0}_{P_1}$ | $v_8^{P_1}$ | $v_7^{P_1}$ | $v_6^{P_1}$ | $v_5^{P_1}$ | $v_8^{P_1}$ | $v_7^{P_1}$ | $v_6^{P_1}$ | $v_5^{P_1}$ | $v_8^{P_1}$ | $v_7^{P_1}$ | $v_6^{P_1}$ | $v_5^{P_1}$ | |
| $(P_1{+}1)$–$P$: | $\mathbf{0}_{P_2}$ | $v_2^{P_2}$ | $v_2^{P_2}$ | $v_2^{P_2}$ | $v_2^{P_2}$ | $v_3^{P_2}$ | $v_3^{P_2}$ | $v_3^{P_2}$ | $v_3^{P_2}$ | $v_4^{P_2}$ | $v_4^{P_2}$ | $v_4^{P_2}$ | $v_2^{P_2}$ | |
| $P + 1$: | $\sqrt{P}$ | $\sqrt{P}$ | $\sqrt{P}$ | $\sqrt{P}$ | $\sqrt{P}$ | $\sqrt{P}$ | $\sqrt{P}$ | $\sqrt{P}$ | $\sqrt{P}$ | $\sqrt{P}$ | $\sqrt{P}$ | $\sqrt{P}$ | $\sqrt{P}$ | |
| $\mathcal{I}$ | 0 | 0 | 0 | $\rightarrow$ | 7 | 5 | 0 | $\rightarrow$ | 5 | 0 | 1 | $\rightarrow$ | 1 | 0 | 2 |
| $1$–$P_1$: | $v_6^{P_1}$ | $v_7^{P_1}$ | $v_8^{P_1}$ | $v_5^{P_1}$ | $v_6^{P_1}$ | $v_7^{P_1}$ | $v_8^{P_1}$ | $v_5^{P_1}$ | $v_6^{P_1}$ | $v_7^{P_1}$ | $v_8^{P_1}$ | $v_5^{P_1}$ | $v_6^{P_1}$ | $v_7^{P_1}$ | $v_8^{P_1}$ |
| $(P_1{+}1)$–$P$: | $v_2^{P_2}$ | $v_2^{P_2}$ | $v_2^{P_2}$ | $v_3^{P_2}$ | $v_3^{P_2}$ | $v_3^{P_2}$ | $v_3^{P_2}$ | $v_4^{P_2}$ | $v_4^{P_2}$ | $v_4^{P_2}$ | $v_4^{P_2}$ | $v_5^{P_2}$ | $v_5^{P_2}$ | $v_5^{P_2}$ | $v_5^{P_2}$ |
| $P + 1$: | $\sqrt{P}$ | $\sqrt{P}$ | $\sqrt{P}$ | $\sqrt{P}$ | $\sqrt{P}$ | $\sqrt{P}$ | $\sqrt{P}$ | $\sqrt{P}$ | $\sqrt{P}$ | $\sqrt{P}$ | $\sqrt{P}$ | $\sqrt{P}$ | $\sqrt{P}$ | $\sqrt{P}$ | $\sqrt{P}$ |

Table 6: Example of $\frac{1}{\sqrt{\alpha_1}} \boldsymbol{K}_1 \boldsymbol{X}^{(0)}$, continuing from Table 4.

| $\mathcal{I}$ | $ | 0 | 5 | 7 | + | 0 | 4 | 8 | + | 0 | 9 | 6 | = | $\cdots$ |
|---|---|---|---|---|---|---|---|---|---|---|---|---|---|---|
| $1-P_1$: | $\mathbf{0}_{P_1}$ | $v_7^{P_1}$ | $v_6^{P_1}$ | $v_5^{P_1}$ | $v_4^{P_1}$ | $v_7^{P_1}$ | $v_6^{P_1}$ | $v_5^{P_1}$ | $v_4^{P_1}$ | $v_7^{P_1}$ | $v_6^{P_1}$ | $v_5^{P_1}$ | $v_4^{P_1}$ | |
| $(P_1+1)-(P_1+P_2)$: | $\mathbf{0}_{P_2}$ | $v_3^{P_2}$ | $v_3^{P_2}$ | $v_3^{P_2}$ | $v_3^{P_2}$ | $v_4^{P_2}$ | $v_4^{P_2}$ | $v_4^{P_2}$ | $v_4^{P_2}$ | $v_5^{P_2}$ | $v_5^{P_2}$ | $v_5^{P_2}$ | $v_3^{P_2}$ | |
| $P_1 + P_2 + 1$: | $\sqrt{P}$ | 0 | 0 | 0 | 0 | 0 | 0 | 0 | 0 | 0 | 0 | 0 | 0 | |

| $\mathcal{I}$ | 0 | 0 | 0 | $\rightarrow$ | 7 | 5 | 0 | $\rightarrow$ | 5 | 0 | 1 | $\rightarrow$ | 1 | 0 | 2 |
|---|---|---|---|---|---|---|---|---|---|---|---|---|---|---|---|
| $1-P_1$: | $v_5^{P_1}$ | $v_6^{P_1}$ | $v_7^{P_1}$ | $v_4^{P_1}$ | $v_5^{P_1}$ | $v_6^{P_1}$ | $v_7^{P_1}$ | $v_4^{P_1}$ | $v_5^{P_1}$ | $v_6^{P_1}$ | $v_7^{P_1}$ | $v_4^{P_1}$ | $v_5^{P_1}$ | $v_6^{P_1}$ | $v_7^{P_1}$ |
| $(P_1+1)-(P_1+P_2)$: | $v_3^{P_2}$ | $v_3^{P_2}$ | $v_3^{P_2}$ | $v_4^{P_2}$ | $v_4^{P_2}$ | $v_4^{P_2}$ | $v_4^{P_2}$ | $v_5^{P_2}$ | $v_5^{P_2}$ | $v_5^{P_2}$ | $v_5^{P_2}$ | $v_6^{P_2}$ | $v_6^{P_2}$ | $v_6^{P_2}$ | $v_6^{P_2}$ |
| $P_1 + P_2 + 1$: | 0 | 0 | 0 | 0 | 0 | 0 | 0 | 0 | 0 | 0 | 0 | 0 | 0 | 0 | 0 |

Table 7: Example of $\boldsymbol{U}_1 \boldsymbol{V}_1 \boldsymbol{X}^{(0)}$, continuing from Table 4.

| $\mathcal{I}$ | $ | 0 | 5 | 7 | + | 0 | 4 | 8 | + | 0 | 9 | 6 | = | $\cdots$ |
|---|---|---|---|---|---|---|---|---|---|---|---|---|---|---|
| 1: NUM | 0 | 0 | 0 | 0 | 0 | 0 | 0 | 0 | 0 | 0 | 0 | 0 | 0 | |
| 2: IS_BOS | 0 | 0 | 0 | 0 | 0 | 0 | 0 | 0 | 0 | 0 | 0 | 0 | 0 | |
| 3: FULL_ONES | 0 | 0 | 0 | 0 | 0 | 0 | 0 | 0 | 0 | 0 | 0 | 0 | 0 | |
| 4: PRE_SUM | 0 | 0 | 15 | 21 | 0 | 0 | 12 | 24 | 0 | 0 | 27 | 18 | 0 | |
| 5: PRE_CARRY | 0 | 0 | 0 | 0 | 0 | 0 | 0 | 0 | 0 | 0 | 0 | 0 | 0 | |
| 6: PRE_ARROW | 0 | 0 | 0 | 0 | 0 | 0 | 0 | 0 | 0 | 0 | 0 | 0 | 0 | |
| 7: PRE_EOS | 0 | 0 | 0 | 0 | 0 | 0 | 0 | 0 | 0 | 0 | 0 | 0 | 0 | |
| 8-17: SUM | $\mathbf{0}_{10}$ | $\mathbf{0}_{10}$ | $\mathbf{0}_{10}$ | $\mathbf{0}_{10}$ | $\mathbf{0}_{10}$ | $\mathbf{0}_{10}$ | $\mathbf{0}_{10}$ | $\mathbf{0}_{10}$ | $\mathbf{0}_{10}$ | $\mathbf{0}_{10}$ | $\mathbf{0}_{10}$ | $\mathbf{0}_{10}$ | $\mathbf{0}_{10}$ | |
| 18: ARROW | 0 | 0 | 0 | 0 | 0 | 0 | 0 | 0 | 0 | 0 | 0 | 0 | 0 | |
| 19: EOS | 0 | 0 | 0 | 0 | 0 | 0 | 0 | 0 | 0 | 0 | 0 | 0 | 0 | |
| 20–end: | $\mathbf{0}_{2P}$ | $\mathbf{0}_{2P}$ | $\mathbf{0}_{2P}$ | $\mathbf{0}_{2P}$ | $\mathbf{0}_{2P}$ | $\mathbf{0}_{2P}$ | $\mathbf{0}_{2P}$ | $\mathbf{0}_{2P}$ | $\mathbf{0}_{2P}$ | $\mathbf{0}_{2P}$ | $\mathbf{0}_{2P}$ | $\mathbf{0}_{2P}$ | $\mathbf{0}_{2P}$ | |

| $\mathcal{I}$ | 0 | 0 | 0 | $\rightarrow$ | 7 | 5 | 0 | $\rightarrow$ | 5 | 0 | 1 | $\rightarrow$ | 1 | 0 | 2 |
|---|---|---|---|---|---|---|---|---|---|---|---|---|---|---|---|
| 1: NUM | 0 | 0 | 0 | 0 | 0 | 0 | 0 | 0 | 0 | 0 | 0 | 0 | 0 | 0 | 0 |
| 2: IS_BOS | 0 | 0 | 0 | 0 | 0 | 0 | 0 | 0 | 0 | 0 | 0 | 0 | 0 | 0 | 0 |
| 3: FULL_ONES | 0 | 0 | 0 | 0 | 0 | 0 | 0 | 0 | 0 | 0 | 0 | 0 | 0 | 0 | 0 |
| 4: PRE_SUM | 0 | 0 | 0 | 0 | 21 | 15 | 0 | 0 | 15 | 0 | 3 | 0 | 3 | 0 | 6 |
| 5: PRE_CARRY | 0 | 0 | 0 | 0 | 0 | 0 | 0 | 0 | 0 | 0 | 0 | 0 | 0 | 0 | 0 |
| 6: PRE_ARROW | 0 | 0 | 0 | 0 | 0 | 0 | 0 | 0 | 0 | 0 | 0 | 0 | 0 | 0 | 0 |
| 7: PRE_EOS | 0 | 0 | 0 | 0 | 0 | 0 | 0 | 0 | 0 | 0 | 0 | 0 | 0 | 0 | 0 |
| 8-17: SUM | $\mathbf{0}_{10}$ | $\mathbf{0}_{10}$ | $\mathbf{0}_{10}$ | $\mathbf{0}_{10}$ | $\mathbf{0}_{10}$ | $\mathbf{0}_{10}$ | $\mathbf{0}_{10}$ | $\mathbf{0}_{10}$ | $\mathbf{0}_{10}$ | $\mathbf{0}_{10}$ | $\mathbf{0}_{10}$ | $\mathbf{0}_{10}$ | $\mathbf{0}_{10}$ | $\mathbf{0}_{10}$ | $\mathbf{0}_{10}$ |
| 18: ARROW | 0 | 0 | 0 | 0 | 0 | 0 | 0 | 0 | 0 | 0 | 0 | 0 | 0 | 0 | 0 |
| 19: EOS | 0 | 0 | 0 | 0 | 0 | 0 | 0 | 0 | 0 | 0 | 0 | 0 | 0 | 0 | 0 |
| 20–end: | $\mathbf{0}_{2P}$ | $\mathbf{0}_{2P}$ | $\mathbf{0}_{2P}$ | $\mathbf{0}_{2P}$ | $\mathbf{0}_{2P}$ | $\mathbf{0}_{2P}$ | $\mathbf{0}_{2P}$ | $\mathbf{0}_{2P}$ | $\mathbf{0}_{2P}$ | $\mathbf{0}_{2P}$ | $\mathbf{0}_{2P}$ | $\mathbf{0}_{2P}$ | $\mathbf{0}_{2P}$ | $\mathbf{0}_{2P}$ | $\mathbf{0}_{2P}$ |

Now, we consider the attention score matrix $\boldsymbol{C}_1 := (\boldsymbol{K}_1 \boldsymbol{X}^{(0)})^\top \boldsymbol{Q}_1 \boldsymbol{X}^{(0)}$ and the attention matrix $\boldsymbol{A}_1 := \texttt{softmax}(\boldsymbol{C}_1) \in \mathbb{R}^{N \times N}$ (including the causal masking operation inside $\texttt{softmax}$). To understand the structure $\boldsymbol{A}_1$, we first focus on the entry $[\boldsymbol{C}_1]_{ij}$, which can be expressed as following:

$$[\boldsymbol{C}_1]_{ij} = \begin{cases} \alpha_1 P & \text{if } i = 1, \\ \alpha_1 P & \text{else if } \left[\boldsymbol{K}_1 \boldsymbol{X}^{(0)}\right]_{\bullet i} = \left[\boldsymbol{Q}_1 \boldsymbol{X}^{(0)}\right]_{\bullet j}, \\ \leq \alpha_1 (P-2) & \text{otherwise.} \end{cases}$$

In essence, $[\boldsymbol{C}_1]_{ij}$ equals $\alpha_1 P$ if and only if $i = 1$ or the key vector for $\sigma_i$ is identical to the query vector for $\sigma_j$. Otherwise, $[\boldsymbol{C}_1]_{ij}$ is less than $\alpha_1 P$. Let $x_j$ denote the number of indices $i$ ($\leq j$) that satisfy $[\boldsymbol{C}_1]_{ij} = \alpha_1 P$. This allows that, with sufficiently large $\alpha_1$, we have

$$[\boldsymbol{A}_1]_{ij} = \begin{cases} 1/x_j & \text{if } i = 1, \\ 0 & \text{else if } i > j, \\ 1/x_j & \text{else if } \left[\boldsymbol{K}_1 \boldsymbol{X}^{(0)}\right]_{\bullet i} = \left[\boldsymbol{Q}_1 \boldsymbol{X}^{(0)}\right]_{\bullet j}, \\ 0 & \text{otherwise.} \end{cases}$$

The output of the attention layer is computed by multiplying the matrix $U_1 V_1 X^{(0)}$ with the matrix $A_1$. An example of $U_1 V_1 X^{(0)} A_1$ is illustrated in Table 8.

Table 8: Example of $U_1 V_1 X^{(0)} A_1$, continuing from Table 4. For simplicity, we omit the columns corresponding to the tokens preceding the equal sign '=', as they do not influence the next-token prediction in the response.

| $\mathcal{I}$ | = | 0 | 0 | 0 | $\rightarrow$ | 7 | 5 | 0 | $\rightarrow$ | 5 | 0 | 1 | $\rightarrow$ | 1 | 0 | 2 |
|---|---|---|---|---|---|---|---|---|---|---|---|---|---|---|---|---|
| 1: NUM | 0 | 0 | 0 | 0 | 0 | 0 | 0 | 0 | 0 | 0 | 0 | 0 | 0 | 0 | 0 | 0 |
| 2: IS_BOS | 0 | 0 | 0 | 0 | 0 | 0 | 0 | 0 | 0 | 0 | 0 | 0 | 0 | 0 | 0 | 0 |
| 3: FULL_ONES | 0 | 0 | 0 | 0 | 0 | 0 | 0 | 0 | 0 | 0 | 0 | 0 | 0 | 0 | 0 | 0 |
| 4: PRE_SUM | 0 | 0 | 0 | 0 | 7 | 5 | 0 | 0 | 15 | 9 | 0 | 0 | 11 | 9 | 1 | 0 |
| 5: PRE_CARRY | 0 | 0 | 0 | 0 | 0 | 0 | 0 | 0 | 0 | 0 | 0 | 0 | 0 | 0 | 0 | 0 |
| 6: PRE_ARROW | 0 | 0 | 0 | 0 | 0 | 0 | 0 | 0 | 0 | 0 | 0 | 0 | 0 | 0 | 0 | 0 |
| 7: PRE_EOS | 0 | 0 | 0 | 0 | 0 | 0 | 0 | 0 | 0 | 0 | 0 | 0 | 0 | 0 | 0 | 0 |
| 8-17: SUM | $\mathbf{0}_{10}$ | $\mathbf{0}_{10}$ | $\mathbf{0}_{10}$ | $\mathbf{0}_{10}$ | $\mathbf{0}_{10}$ | $\mathbf{0}_{10}$ | $\mathbf{0}_{10}$ | $\mathbf{0}_{10}$ | $\mathbf{0}_{10}$ | $\mathbf{0}_{10}$ | $\mathbf{0}_{10}$ | $\mathbf{0}_{10}$ | $\mathbf{0}_{10}$ | $\mathbf{0}_{10}$ | $\mathbf{0}_{10}$ | $\mathbf{0}_{10}$ |
| 18: ARROW | 0 | 0 | 0 | 0 | 0 | 0 | 0 | 0 | 0 | 0 | 0 | 0 | 0 | 0 | 0 | 0 |
| 19: EOS | 0 | 0 | 0 | 0 | 0 | 0 | 0 | 0 | 0 | 0 | 0 | 0 | 0 | 0 | 0 | 0 |
| 20–end: | $\mathbf{0}_{2P}$ | $\mathbf{0}_{2P}$ | $\mathbf{0}_{2P}$ | $\mathbf{0}_{2P}$ | $\mathbf{0}_{2P}$ | $\mathbf{0}_{2P}$ | $\mathbf{0}_{2P}$ | $\mathbf{0}_{2P}$ | $\mathbf{0}_{2P}$ | $\mathbf{0}_{2P}$ | $\mathbf{0}_{2P}$ | $\mathbf{0}_{2P}$ | $\mathbf{0}_{2P}$ | $\mathbf{0}_{2P}$ | $\mathbf{0}_{2P}$ | $\mathbf{0}_{2P}$ |

To clarify our example, consider the input sequence $\overline{\$057 + 048 + 096 = 000 \rightarrow 750 \rightarrow 501 \rightarrow}$. At this stage, the goal of the model is to predict '1' as the next token, which corresponds to the least significant digit (LSD) of the sum of 6 (the third token of $\overline{096}$) and 5 (the first token of $\overline{501}$). It is important to note that the query vector for the last arrow token is identical to the key vectors for 6 and 5. This can be easily verified by comparing Table 5 and Table 6. As a result, the dimension PRE_SUM of $U_1 V_1 X^{(0)} A_1$ for the last arrow token can be obtained by computing $\frac{1}{3}(0 + 18 + 15) = 11$. Here, the "3" in $\frac{1}{3}$ originates from the number of tokens that the last arrow token attends to: BOS, 6, and 5, and this is the reason why we design the matrix $V_1$ with the scalar 3. The operation of extracting the LSD from 11 will be handled in the subsequent feed-forward layer.

### D.5.2 ATTENTION HEAD 2: CARRY DETECTION

The goal of the second attention head is to fill the dimension PRE_CARRY with the appropriate values. To illustrate this, consider the partial input sequence $\overline{\$057 + 048 + 096 = 000 \rightarrow 750 \rightarrow 501 \rightarrow 1}$. To predict 0 as the next token, the model needs to (1) compute the sum of 9 (the second token of $\overline{096}$) and 0 (the second token of $\overline{501}$), and (2) detect the carry from the previous digit's sum, which was $6 + 5$. The first attention head handles the first part, by making 1 (the last token of our example input sequence) attend to 9 and 0. We aim to design the second attention head to handle the second part. A key observation is that detecting the carry becomes possible when the model attends to 6 and 5, by verifying that $6 + 5 - 1 \in \{9, 10\}$ (see Section E.4.2. of Cho et al. (2024) for more details). Thus, the second attention head's goal is to compute $6 + 5$ and assign a value of 11 to the dimension PRE_CARRY of the token 1.

Recall that $P = P_1 + P_2$ and $d = 2P + 19$. Let the dimension of the second attention head be $d_{QK,2} = P + 1$. We define the query matrix $Q_2$ and the key matrix $K_2$ as follows:

$$Q_2 = \begin{pmatrix} \mathbf{0}_{P_1 \times 19} & \sqrt{\alpha_2} I_{P_1} & \mathbf{0}_{P_1 \times P_1} & \mathbf{0}_{P_1 \times P_2} & \mathbf{0}_{P_1 \times P_2} \\ \mathbf{0}_{P_2 \times 19} & \mathbf{0}_{P_2 \times P_1} & \mathbf{0}_{P_2 \times P_1} & \sqrt{\alpha_2} I_{P_2} & \mathbf{0}_{P_2 \times P_2} \\ \sqrt{\alpha_2} P (e^{19}_{\text{FULL\_ONES}})^\top & \mathbf{0}_{1 \times P_1} & \mathbf{0}_{1 \times P_1} & \mathbf{0}_{1 \times P_2} & \mathbf{0}_{1 \times P_2} \end{pmatrix} \in \mathbb{R}^{d_{QK,2} \times d}, \quad (11)$$

$$K_2 = \begin{pmatrix} \mathbf{0}_{P_1 \times 19} & \sqrt{\alpha_2} I_{P_1} & \mathbf{0}_{P_1 \times P_1} & \mathbf{0}_{P_1 \times P_2} & \mathbf{0}_{P_1 \times P_2} \\ \mathbf{0}_{P_2 \times 19} & \mathbf{0}_{P_2 \times P_1} & \mathbf{0}_{P_2 \times P_1} & \mathbf{0}_{P_2 \times P_2} & \sqrt{\alpha_2} I_{P_2} \\ \sqrt{\alpha_2} P (e^{19}_{\text{IS\_BOS}})^\top & \mathbf{0}_{1 \times P_1} & \mathbf{0}_{1 \times P_1} & \mathbf{0}_{1 \times P_2} & \mathbf{0}_{1 \times P_2} \end{pmatrix} \in \mathbb{R}^{d_{QK,2} \times d}, \quad (12)$$

where $\alpha_2$ is a scaling factor, which we can choose to be arbitrarily large. Also, let $d_{V,2} = 1$ and define

$$V_2 = 3(e^d_{\text{NUM}})^\top \in \mathbb{R}^{d_{V,2} \times d}, \quad (13)$$

$$U_2 = e^d_{\text{PRE\_CARRY}} \in \mathbb{R}^{d \times d_{V,2}}. \quad (14)$$

An example of $Q_2 X^{(0)}$, $K_2 X^{(0)}$, $U_2 V_2 X^{(0)}$, and $U_2 V_2 X^{(0)} A_2$ is illustrated in Tables 9 to 12. Similar to the first attention head, with sufficiently large $\alpha_2$, the $j$-th token $\sigma_j$ will only attend to the BOS token and tokens whose key vectors match the query vector of $\sigma_j$. However, the tokens that $\sigma_j$ attends to differ from those in the first attention as the design of the query matrix $Q_2$ and $K_2$ is slightly modified. By adjusting the placement of $\sqrt{\alpha_2} I_{P_1}$ and $\sqrt{\alpha_2} I_{P_2}$ in the $Q_2$ and $K_2$, we can control which tokens are attended to.

Table 9: Example of $\frac{1}{\sqrt{\alpha_2}} Q_2 X^{(0)}$, continuing from Table 4.

| $\mathcal{I}$ | $\$$ | 0 | 5 | 7 | + | 0 | 4 | 8 | + | 0 | 9 | 6 | = | $\cdots$ |
|---|---|---|---|---|---|---|---|---|---|---|---|---|---|---|
| $1$–$P_1$: | $\mathbf{0}_{P_1}$ | $v_7^{P_1}$ | $v_6^{P_1}$ | $v_5^{P_1}$ | $v_4^{P_1}$ | $v_7^{P_1}$ | $v_6^{P_1}$ | $v_5^{P_1}$ | $v_4^{P_1}$ | $v_7^{P_1}$ | $v_6^{P_1}$ | $v_5^{P_1}$ | $v_4^{P_1}$ | |
| $(P_1+1)$–$P$: | $\mathbf{0}_{P_2}$ | $v_2^{P_2}$ | $v_2^{P_2}$ | $v_2^{P_2}$ | $v_2^{P_2}$ | $v_3^{P_2}$ | $v_3^{P_2}$ | $v_3^{P_2}$ | $v_3^{P_2}$ | $v_4^{P_2}$ | $v_4^{P_2}$ | $v_4^{P_2}$ | $v_2^{P_2}$ | |
| $P+1$: | $\sqrt{P}$ | $\sqrt{P}$ | $\sqrt{P}$ | $\sqrt{P}$ | $\sqrt{P}$ | $\sqrt{P}$ | $\sqrt{P}$ | $\sqrt{P}$ | $\sqrt{P}$ | $\sqrt{P}$ | $\sqrt{P}$ | $\sqrt{P}$ | $\sqrt{P}$ | |

| $\mathcal{I}$ | 0 | 0 | 0 | $\rightarrow$ | 7 | 5 | 0 | $\rightarrow$ | 5 | 0 | 1 | $\rightarrow$ | 1 | 0 | 2 |
|---|---|---|---|---|---|---|---|---|---|---|---|---|---|---|---|
| $1$–$P_1$: | $v_5^{P_1}$ | $v_6^{P_1}$ | $v_7^{P_1}$ | $v_4^{P_1}$ | $v_5^{P_1}$ | $v_6^{P_1}$ | $v_7^{P_1}$ | $v_4^{P_1}$ | $v_5^{P_1}$ | $v_6^{P_1}$ | $v_7^{P_1}$ | $v_4^{P_1}$ | $v_5^{P_1}$ | $v_6^{P_1}$ | $v_7^{P_1}$ |
| $(P_1+1)$–$P$: | $v_2^{P_2}$ | $v_2^{P_2}$ | $v_2^{P_2}$ | $v_3^{P_2}$ | $v_3^{P_2}$ | $v_3^{P_2}$ | $v_3^{P_2}$ | $v_4^{P_2}$ | $v_4^{P_2}$ | $v_4^{P_2}$ | $v_4^{P_2}$ | $v_5^{P_2}$ | $v_5^{P_2}$ | $v_5^{P_2}$ | $v_5^{P_2}$ |
| $P+1$: | $\sqrt{P}$ | $\sqrt{P}$ | $\sqrt{P}$ | $\sqrt{P}$ | $\sqrt{P}$ | $\sqrt{P}$ | $\sqrt{P}$ | $\sqrt{P}$ | $\sqrt{P}$ | $\sqrt{P}$ | $\sqrt{P}$ | $\sqrt{P}$ | $\sqrt{P}$ | $\sqrt{P}$ | $\sqrt{P}$ |

Table 10: Example of $\frac{1}{\sqrt{\alpha_2}} K_2 X^{(0)}$, continuing from Table 4.

| $\mathcal{I}$ | $\$$ | 0 | 5 | 7 | + | 0 | 4 | 8 | + | 0 | 9 | 6 | = | $\cdots$ |
|---|---|---|---|---|---|---|---|---|---|---|---|---|---|---|
| $1$–$P_1$: | $\mathbf{0}_{P_1}$ | $v_7^{P_1}$ | $v_6^{P_1}$ | $v_5^{P_1}$ | $v_4^{P_1}$ | $v_7^{P_1}$ | $v_6^{P_1}$ | $v_5^{P_1}$ | $v_4^{P_1}$ | $v_7^{P_1}$ | $v_6^{P_1}$ | $v_5^{P_1}$ | $v_4^{P_1}$ | |
| $(P_1+1)$–$P$: | $\mathbf{0}_{P_2}$ | $v_3^{P_2}$ | $v_3^{P_2}$ | $v_3^{P_2}$ | $v_3^{P_2}$ | $v_4^{P_2}$ | $v_4^{P_2}$ | $v_4^{P_2}$ | $v_4^{P_2}$ | $v_5^{P_2}$ | $v_5^{P_2}$ | $v_5^{P_2}$ | $v_3^{P_2}$ | |
| $P+1$: | $\sqrt{P}$ | 0 | 0 | 0 | 0 | 0 | 0 | 0 | 0 | 0 | 0 | 0 | 0 | |

| $\mathcal{I}$ | 0 | 0 | 0 | $\rightarrow$ | 7 | 5 | 0 | $\rightarrow$ | 5 | 0 | 1 | $\rightarrow$ | 1 | 0 | 2 |
|---|---|---|---|---|---|---|---|---|---|---|---|---|---|---|---|
| $1$–$P_1$: | $v_5^{P_1}$ | $v_6^{P_1}$ | $v_7^{P_1}$ | $v_4^{P_1}$ | $v_5^{P_1}$ | $v_6^{P_1}$ | $v_7^{P_1}$ | $v_4^{P_1}$ | $v_5^{P_1}$ | $v_6^{P_1}$ | $v_7^{P_1}$ | $v_4^{P_1}$ | $v_5^{P_1}$ | $v_6^{P_1}$ | $v_7^{P_1}$ |
| $(P_1+1)$–$P$: | $v_3^{P_2}$ | $v_3^{P_2}$ | $v_3^{P_2}$ | $v_4^{P_2}$ | $v_4^{P_2}$ | $v_4^{P_2}$ | $v_4^{P_2}$ | $v_5^{P_2}$ | $v_5^{P_2}$ | $v_5^{P_2}$ | $v_5^{P_2}$ | $v_6^{P_2}$ | $v_6^{P_2}$ | $v_6^{P_2}$ | $v_6^{P_2}$ |
| $P+1$: | 0 | 0 | 0 | 0 | 0 | 0 | 0 | 0 | 0 | 0 | 0 | 0 | 0 | 0 | 0 |

Table 11: Example of $\boldsymbol{U}_2\boldsymbol{V}_2\boldsymbol{X}^{(0)}$, continuing from Table 4.

| $\mathcal{I}$ | $ | 0 | 5 | 7 | + | 0 | 4 | 8 | + | 0 | 9 | 6 | = | $\cdots$ |
|---|---|---|---|---|---|---|---|---|---|---|---|---|---|---|
| 1: NUM | 0 | 0 | 0 | 0 | 0 | 0 | 0 | 0 | 0 | 0 | 0 | 0 | 0 | |
| 2: IS_BOS | 0 | 0 | 0 | 0 | 0 | 0 | 0 | 0 | 0 | 0 | 0 | 0 | 0 | |
| 3: FULL_ONES | 0 | 0 | 0 | 0 | 0 | 0 | 0 | 0 | 0 | 0 | 0 | 0 | 0 | |
| 4: PRE_SUM | 0 | 0 | 0 | 0 | 0 | 0 | 0 | 0 | 0 | 0 | 0 | 0 | 0 | |
| 5: PRE_CARRY | 0 | 0 | 15 | 21 | 0 | 0 | 12 | 24 | 0 | 0 | 27 | 18 | 0 | |
| 6: PRE_ARROW | 0 | 0 | 0 | 0 | 0 | 0 | 0 | 0 | 0 | 0 | 0 | 0 | 0 | |
| 7: PRE_EOS | 0 | 0 | 0 | 0 | 0 | 0 | 0 | 0 | 0 | 0 | 0 | 0 | 0 | |
| 8-17: SUM | $\mathbf{0}_{10}$ | $\mathbf{0}_{10}$ | $\mathbf{0}_{10}$ | $\mathbf{0}_{10}$ | $\mathbf{0}_{10}$ | $\mathbf{0}_{10}$ | $\mathbf{0}_{10}$ | $\mathbf{0}_{10}$ | $\mathbf{0}_{10}$ | $\mathbf{0}_{10}$ | $\mathbf{0}_{10}$ | $\mathbf{0}_{10}$ | $\mathbf{0}_{10}$ | |
| 18: ARROW | 0 | 0 | 0 | 0 | 0 | 0 | 0 | 0 | 0 | 0 | 0 | 0 | 0 | |
| 19: EOS | 0 | 0 | 0 | 0 | 0 | 0 | 0 | 0 | 0 | 0 | 0 | 0 | 0 | |
| 20–end: | $\mathbf{0}_{2P}$ | $\mathbf{0}_{2P}$ | $\mathbf{0}_{2P}$ | $\mathbf{0}_{2P}$ | $\mathbf{0}_{2P}$ | $\mathbf{0}_{2P}$ | $\mathbf{0}_{2P}$ | $\mathbf{0}_{2P}$ | $\mathbf{0}_{2P}$ | $\mathbf{0}_{2P}$ | $\mathbf{0}_{2P}$ | $\mathbf{0}_{2P}$ | $\mathbf{0}_{2P}$ | |

| $\mathcal{I}$ | 0 | 0 | 0 | $\rightarrow$ | 7 | 5 | 0 | $\rightarrow$ | 5 | 0 | 1 | $\rightarrow$ | 1 | 0 | 2 |
|---|---|---|---|---|---|---|---|---|---|---|---|---|---|---|---|
| 1: NUM | 0 | 0 | 0 | 0 | 0 | 0 | 0 | 0 | 0 | 0 | 0 | 0 | 0 | 0 | 0 |
| 2: IS_BOS | 0 | 0 | 0 | 0 | 0 | 0 | 0 | 0 | 0 | 0 | 0 | 0 | 0 | 0 | 0 |
| 3: FULL_ONES | 0 | 0 | 0 | 0 | 0 | 0 | 0 | 0 | 0 | 0 | 0 | 0 | 0 | 0 | 0 |
| 4: PRE_SUM | 0 | 0 | 0 | 0 | 0 | 0 | 0 | 0 | 0 | 0 | 0 | 0 | 0 | 0 | 0 |
| 5: PRE_CARRY | 0 | 0 | 0 | 0 | 21 | 15 | 0 | 0 | 15 | 0 | 3 | 0 | 3 | 0 | 6 |
| 6: PRE_ARROW | 0 | 0 | 0 | 0 | 0 | 0 | 0 | 0 | 0 | 0 | 0 | 0 | 0 | 0 | 0 |
| 7: PRE_EOS | 0 | 0 | 0 | 0 | 0 | 0 | 0 | 0 | 0 | 0 | 0 | 0 | 0 | 0 | 0 |
| 8-17: SUM | $\mathbf{0}_{10}$ | $\mathbf{0}_{10}$ | $\mathbf{0}_{10}$ | $\mathbf{0}_{10}$ | $\mathbf{0}_{10}$ | $\mathbf{0}_{10}$ | $\mathbf{0}_{10}$ | $\mathbf{0}_{10}$ | $\mathbf{0}_{10}$ | $\mathbf{0}_{10}$ | $\mathbf{0}_{10}$ | $\mathbf{0}_{10}$ | $\mathbf{0}_{10}$ | $\mathbf{0}_{10}$ | $\mathbf{0}_{10}$ |
| 18: ARROW | 0 | 0 | 0 | 0 | 0 | 0 | 0 | 0 | 0 | 0 | 0 | 0 | 0 | 0 | 0 |
| 19: EOS | 0 | 0 | 0 | 0 | 0 | 0 | 0 | 0 | 0 | 0 | 0 | 0 | 0 | 0 | 0 |
| 20–end: | $\mathbf{0}_{2P}$ | $\mathbf{0}_{2P}$ | $\mathbf{0}_{2P}$ | $\mathbf{0}_{2P}$ | $\mathbf{0}_{2P}$ | $\mathbf{0}_{2P}$ | $\mathbf{0}_{2P}$ | $\mathbf{0}_{2P}$ | $\mathbf{0}_{2P}$ | $\mathbf{0}_{2P}$ | $\mathbf{0}_{2P}$ | $\mathbf{0}_{2P}$ | $\mathbf{0}_{2P}$ | $\mathbf{0}_{2P}$ | $\mathbf{0}_{2P}$ |

Table 12: Example of $\boldsymbol{U}_2\boldsymbol{V}_2\boldsymbol{X}^{(0)}\boldsymbol{A}_2$, continuing from Table 4. For simplicity, we omit the columns corresponding to the tokens preceding the equal sign '=', as they do not influence the next-token prediction in the response.

| $\mathcal{I}$ | = | 0 | 0 | 0 | $\rightarrow$ | 7 | 5 | 0 | $\rightarrow$ | 5 | 0 | 1 | $\rightarrow$ | 1 | 0 | 2 |
|---|---|---|---|---|---|---|---|---|---|---|---|---|---|---|---|---|
| 1: NUM | 0 | 0 | 0 | 0 | 0 | 0 | 0 | 0 | 0 | 0 | 0 | 0 | 0 | 0 | 0 | 0 |
| 2: IS_BOS | 0 | 0 | 0 | 0 | 0 | 0 | 0 | 0 | 0 | 0 | 0 | 0 | 0 | 0 | 0 | 0 |
| 3: FULL_ONES | 0 | 0 | 0 | 0 | 0 | 0 | 0 | 0 | 0 | 0 | 0 | 0 | 0 | 0 | 0 | 0 |
| 4: PRE_SUM | 0 | 0 | 0 | 0 | 0 | 0 | 0 | 0 | 0 | 0 | 0 | 0 | 0 | 0 | 0 | 0 |
| 5: PRE_CARRY | 0 | 0 | 0 | 0 | 0 | 7 | 5 | 0 | 0 | 15 | 9 | 0 | 0 | 11 | 9 | 1 |
| 6: PRE_ARROW | 0 | 0 | 0 | 0 | 0 | 0 | 0 | 0 | 0 | 0 | 0 | 0 | 0 | 0 | 0 | 0 |
| 7: PRE_EOS | 0 | 0 | 0 | 0 | 0 | 0 | 0 | 0 | 0 | 0 | 0 | 0 | 0 | 0 | 0 | 0 |
| 8-17: SUM | $\mathbf{0}_{10}$ | $\mathbf{0}_{10}$ | $\mathbf{0}_{10}$ | $\mathbf{0}_{10}$ | $\mathbf{0}_{10}$ | $\mathbf{0}_{10}$ | $\mathbf{0}_{10}$ | $\mathbf{0}_{10}$ | $\mathbf{0}_{10}$ | $\mathbf{0}_{10}$ | $\mathbf{0}_{10}$ | $\mathbf{0}_{10}$ | $\mathbf{0}_{10}$ | $\mathbf{0}_{10}$ | $\mathbf{0}_{10}$ | $\mathbf{0}_{10}$ |
| 18: ARROW | 0 | 0 | 0 | 0 | 0 | 0 | 0 | 0 | 0 | 0 | 0 | 0 | 0 | 0 | 0 | 0 |
| 19: EOS | 0 | 0 | 0 | 0 | 0 | 0 | 0 | 0 | 0 | 0 | 0 | 0 | 0 | 0 | 0 | 0 |
| 20–end: | $\mathbf{0}_{2P}$ | $\mathbf{0}_{2P}$ | $\mathbf{0}_{2P}$ | $\mathbf{0}_{2P}$ | $\mathbf{0}_{2P}$ | $\mathbf{0}_{2P}$ | $\mathbf{0}_{2P}$ | $\mathbf{0}_{2P}$ | $\mathbf{0}_{2P}$ | $\mathbf{0}_{2P}$ | $\mathbf{0}_{2P}$ | $\mathbf{0}_{2P}$ | $\mathbf{0}_{2P}$ | $\mathbf{0}_{2P}$ | $\mathbf{0}_{2P}$ | $\mathbf{0}_{2P}$ |

### D.5.3 ATTENTION HEAD 3: ARROW DETECTION

The goal of the third attention head is to fill the dimension PRE_ARROW. We aim to put 1 if the next token the model has to predict is the arrow ($\rightarrow$), and otherwise, we will put strictly smaller values (below $1/2$).

Recall that $d = 2P + 19$. Let the dimension of the first attention head be $d_{QK,3} = P_1 + 1$. We define the query matrix $\boldsymbol{Q}_3$ and the key matrix $\boldsymbol{K}_3$ as follows:

$$\boldsymbol{Q}_3 = \begin{pmatrix} \mathbf{0}_{P_1 \times 19} & \mathbf{0}_{P_1 \times P_1} & \sqrt{\alpha_3}\boldsymbol{I}_{P_1} & \mathbf{0}_{P_1 \times P_2} & \mathbf{0}_{P_1 \times P_2} \\ \sqrt{\alpha_3 P_1}(\boldsymbol{e}_{\text{FULL\_ONES}}^{19})^\top & \mathbf{0}_{1 \times P_1} & \mathbf{0}_{1 \times P_1} & \mathbf{0}_{1 \times P_2} & \mathbf{0}_{1 \times P_2} \end{pmatrix} \in \mathbb{R}^{d_{QK,3} \times d}, \quad (15)$$

$$\boldsymbol{K}_3 = \begin{pmatrix} \mathbf{0}_{P_1 \times 19} & \sqrt{\alpha_3}\boldsymbol{I}_{P_1} & \mathbf{0}_{P_1 \times P_1} & \mathbf{0}_{P_1 \times P_2} & \mathbf{0}_{P_1 \times P_2} \\ \sqrt{\alpha_3 P_1}(\boldsymbol{e}_{\text{IS\_BOS}}^{19})^\top & \mathbf{0}_{1 \times P_1} & \mathbf{0}_{1 \times P_1} & \mathbf{0}_{1 \times P_2} & \mathbf{0}_{1 \times P_2} \end{pmatrix} \in \mathbb{R}^{d_{QK,3} \times d}, \quad (16)$$

where $\alpha_3$ is a scaling factor, which we can choose to be arbitrarily large. Also, let $d_{V,3} = 1$ and define

$$\boldsymbol{V}_1 = (\boldsymbol{e}^d_{\text{IS\_BOS}})^\top \in \mathbb{R}^{d_{V,3} \times d}, \tag{17}$$

$$\boldsymbol{U}_1 = \boldsymbol{e}^d_{\text{PRE\_ARROW}} \in \mathbb{R}^{d \times d_{V,3}}. \tag{18}$$

An example of $\boldsymbol{Q}_3 \boldsymbol{X}^{(0)}$, $\boldsymbol{K}_3 \boldsymbol{X}^{(0)}$, $\boldsymbol{U}_3 \boldsymbol{V}_3 \boldsymbol{X}^{(0)}$, and $\boldsymbol{U}_3 \boldsymbol{V}_3 \boldsymbol{X}^{(0)} \boldsymbol{A}_3$ is illustrated in Tables 13 to 16. Similar to the first and the second attention head, with sufficiently large $\alpha_3$, the $j$-token $\sigma_j$ will only attend to the BOS token and tokens whose key vectors match the query vector of $\sigma_j$.

To enhance understanding, consider two input sequences $\mathcal{I}_1 = \overline{\$057 + 048 + 096 = 000 \rightarrow 750}$ and $\mathcal{I}_2 = \overline{\$057 + 048 + 096 = 000 \rightarrow 75}$. We first analyze $\mathcal{I}_1$. By comparing Table 13 and Table 14, we can observe that the token 0 (the last token in $\overline{750}$) only attends to the BOS token. Therefore, the dimension PRE_ARROW of the token 0 will be set to 1 in the matrix $\boldsymbol{U}_3 \boldsymbol{V}_3 \boldsymbol{X}^{(0)} \boldsymbol{A}_3$. Next, for $\mathcal{I}_2$, we can see that the token 5 (the second token in $\overline{750}$) attends to the BOS token and four other tokens (0 from $\overline{057}$, 0 from $\overline{048}$, 0 from $\overline{096}$, 0 from $\overline{000}$). As a result, the dimension PRE_ARROW of the token 5 will be filled by $1/5$, where the denominator 5 comes from the softmax operation.

Consequently, the model can decide to predict the arrow ($\rightarrow$) as the next token if the dimension PRE_ARROW in the matrix $\boldsymbol{U}_3 \boldsymbol{V}_3 \boldsymbol{X}^{(0)} \boldsymbol{A}_3$ of the last token of the given input sequence is equal to 1. Otherwise, for the case where the model should not predict the arrow as the next token, PRE_ARROW is set to the value less than $1/2$.

Table 13: Example of $\frac{1}{\sqrt{\alpha_3}} \boldsymbol{Q}_3 \boldsymbol{X}^{(0)}$, continuing from Table 4.

| $\mathcal{I}$ | \$ | 0 | 5 | 7 | + | 0 | 4 | 8 | + | 0 | 9 | 6 | = | $\cdots$ |
|---|---|---|---|---|---|---|---|---|---|---|---|---|---|---|
| $1-P_1$: | $\boldsymbol{0}_{P_1}$ | $\boldsymbol{v}_8^{P_1}$ | $\boldsymbol{v}_7^{P_1}$ | $\boldsymbol{v}_6^{P_1}$ | $\boldsymbol{v}_5^{P_1}$ | $\boldsymbol{v}_8^{P_1}$ | $\boldsymbol{v}_7^{P_1}$ | $\boldsymbol{v}_6^{P_1}$ | $\boldsymbol{v}_5^{P_1}$ | $\boldsymbol{v}_8^{P_1}$ | $\boldsymbol{v}_7^{P_1}$ | $\boldsymbol{v}_6^{P_1}$ | $\boldsymbol{v}_5^{P_1}$ | |
| $P_1 + 1$: | $\sqrt{P_1}$ | $\sqrt{P_1}$ | $\sqrt{P_1}$ | $\sqrt{P_1}$ | $\sqrt{P_1}$ | $\sqrt{P_1}$ | $\sqrt{P_1}$ | $\sqrt{P_1}$ | $\sqrt{P_1}$ | $\sqrt{P_1}$ | $\sqrt{P_1}$ | $\sqrt{P_1}$ | $\sqrt{P_1}$ | |

| $\mathcal{I}$ | 0 | 0 | 0 | $\rightarrow$ | 7 | 5 | 0 | $\rightarrow$ | 5 | 0 | 1 | $\rightarrow$ | 1 | 0 | 2 |
|---|---|---|---|---|---|---|---|---|---|---|---|---|---|---|---|
| $1-P_1$: | $\boldsymbol{v}_6^{P_1}$ | $\boldsymbol{v}_7^{P_1}$ | $\boldsymbol{v}_8^{P_1}$ | $\boldsymbol{v}_5^{P_1}$ | $\boldsymbol{v}_6^{P_1}$ | $\boldsymbol{v}_7^{P_1}$ | $\boldsymbol{v}_8^{P_1}$ | $\boldsymbol{v}_5^{P_1}$ | $\boldsymbol{v}_6^{P_1}$ | $\boldsymbol{v}_7^{P_1}$ | $\boldsymbol{v}_8^{P_1}$ | $\boldsymbol{v}_5^{P_1}$ | $\boldsymbol{v}_6^{P_1}$ | $\boldsymbol{v}_7^{P_1}$ | $\boldsymbol{v}_8^{P_1}$ |
| $P_1 + 1$: | $\sqrt{P_1}$ | $\sqrt{P_1}$ | $\sqrt{P_1}$ | $\sqrt{P_1}$ | $\sqrt{P_1}$ | $\sqrt{P_1}$ | $\sqrt{P_1}$ | $\sqrt{P_1}$ | $\sqrt{P_1}$ | $\sqrt{P_1}$ | $\sqrt{P_1}$ | $\sqrt{P_1}$ | $\sqrt{P_1}$ | $\sqrt{P_1}$ | $\sqrt{P_1}$ |

Table 14: Example of $\frac{1}{\sqrt{\alpha_3}} \boldsymbol{K}_3 \boldsymbol{X}^{(0)}$, continuing from Table 4.

| $\mathcal{I}$ | \$ | 0 | 5 | 7 | + | 0 | 4 | 8 | + | 0 | 9 | 6 | = | $\cdots$ |
|---|---|---|---|---|---|---|---|---|---|---|---|---|---|---|
| $1-P_1$: | $\boldsymbol{0}_{P_1}$ | $\boldsymbol{v}_7^{P_1}$ | $\boldsymbol{v}_6^{P_1}$ | $\boldsymbol{v}_5^{P_1}$ | $\boldsymbol{v}_4^{P_1}$ | $\boldsymbol{v}_7^{P_1}$ | $\boldsymbol{v}_6^{P_1}$ | $\boldsymbol{v}_5^{P_1}$ | $\boldsymbol{v}_4^{P_1}$ | $\boldsymbol{v}_7^{P_1}$ | $\boldsymbol{v}_6^{P_1}$ | $\boldsymbol{v}_5^{P_1}$ | $\boldsymbol{v}_4^{P_1}$ | |
| $P_1 + 1$: | $\sqrt{P_1}$ | 0 | 0 | 0 | 0 | 0 | 0 | 0 | 0 | 0 | 0 | 0 | 0 | |

| $\mathcal{I}$ | 0 | 0 | 0 | $\rightarrow$ | 7 | 5 | 0 | $\rightarrow$ | 5 | 0 | 1 | $\rightarrow$ | 1 | 0 | 2 |
|---|---|---|---|---|---|---|---|---|---|---|---|---|---|---|---|
| $1-P_1$: | $\boldsymbol{v}_5^{P_1}$ | $\boldsymbol{v}_6^{P_1}$ | $\boldsymbol{v}_7^{P_1}$ | $\boldsymbol{v}_4^{P_1}$ | $\boldsymbol{v}_5^{P_1}$ | $\boldsymbol{v}_6^{P_1}$ | $\boldsymbol{v}_7^{P_1}$ | $\boldsymbol{v}_4^{P_1}$ | $\boldsymbol{v}_5^{P_1}$ | $\boldsymbol{v}_6^{P_1}$ | $\boldsymbol{v}_7^{P_1}$ | $\boldsymbol{v}_4^{P_1}$ | $\boldsymbol{v}_5^{P_1}$ | $\boldsymbol{v}_6^{P_1}$ | $\boldsymbol{v}_7^{P_1}$ |
| $P_1 + 1$: | 0 | 0 | 0 | 0 | 0 | 0 | 0 | 0 | 0 | 0 | 0 | 0 | 0 | 0 | 0 |

Table 15: Example of $\boldsymbol{U}_3\boldsymbol{V}_3\boldsymbol{X}^{(0)}$, continuing from Table 4.

| $\mathcal{I}$ | \$ | 0 | 5 | 7 | + | 0 | 4 | 8 | + | 0 | 9 | 6 | = | $\cdots$ |
|---|---|---|---|---|---|---|---|---|---|---|---|---|---|---|
| 1: NUM | 0 | 0 | 0 | 0 | 0 | 0 | 0 | 0 | 0 | 0 | 0 | 0 | 0 | |
| 2: IS_BOS | 0 | 0 | 0 | 0 | 0 | 0 | 0 | 0 | 0 | 0 | 0 | 0 | 0 | |
| 3: FULL_ONES | 0 | 0 | 0 | 0 | 0 | 0 | 0 | 0 | 0 | 0 | 0 | 0 | 0 | |
| 4: PRE_SUM | 0 | 0 | 0 | 0 | 0 | 0 | 0 | 0 | 0 | 0 | 0 | 0 | 0 | |
| 5: PRE_CARRY | 0 | 0 | 0 | 0 | 0 | 0 | 0 | 0 | 0 | 0 | 0 | 0 | 0 | |
| 6: PRE_ARROW | 1 | 0 | 0 | 0 | 0 | 0 | 0 | 0 | 0 | 0 | 0 | 0 | 0 | |
| 7: PRE_EOS | 0 | 0 | 0 | 0 | 0 | 0 | 0 | 0 | 0 | 0 | 0 | 0 | 0 | |
| 8-17: SUM | $\boldsymbol{0}_{10}$ | $\boldsymbol{0}_{10}$ | $\boldsymbol{0}_{10}$ | $\boldsymbol{0}_{10}$ | $\boldsymbol{0}_{10}$ | $\boldsymbol{0}_{10}$ | $\boldsymbol{0}_{10}$ | $\boldsymbol{0}_{10}$ | $\boldsymbol{0}_{10}$ | $\boldsymbol{0}_{10}$ | $\boldsymbol{0}_{10}$ | $\boldsymbol{0}_{10}$ | $\boldsymbol{0}_{10}$ | |
| 18: ARROW | 0 | 0 | 0 | 0 | 0 | 0 | 0 | 0 | 0 | 0 | 0 | 0 | 0 | |
| 19: EOS | 0 | 0 | 0 | 0 | 0 | 0 | 0 | 0 | 0 | 0 | 0 | 0 | 0 | |
| 20–end: | $\boldsymbol{0}_{2P}$ | $\boldsymbol{0}_{2P}$ | $\boldsymbol{0}_{2P}$ | $\boldsymbol{0}_{2P}$ | $\boldsymbol{0}_{2P}$ | $\boldsymbol{0}_{2P}$ | $\boldsymbol{0}_{2P}$ | $\boldsymbol{0}_{2P}$ | $\boldsymbol{0}_{2P}$ | $\boldsymbol{0}_{2P}$ | $\boldsymbol{0}_{2P}$ | $\boldsymbol{0}_{2P}$ | $\boldsymbol{0}_{2P}$ | |

| $\mathcal{I}$ | 0 | 0 | 0 | $\rightarrow$ | 7 | 5 | 0 | $\rightarrow$ | 5 | 0 | 1 | $\rightarrow$ | 1 | 0 | 2 |
|---|---|---|---|---|---|---|---|---|---|---|---|---|---|---|---|
| 1: NUM | 0 | 0 | 0 | 0 | 0 | 0 | 0 | 0 | 0 | 0 | 0 | 0 | 0 | 0 | 0 |
| 2: IS_BOS | 0 | 0 | 0 | 0 | 0 | 0 | 0 | 0 | 0 | 0 | 0 | 0 | 0 | 0 | 0 |
| 3: FULL_ONES | 0 | 0 | 0 | 0 | 0 | 0 | 0 | 0 | 0 | 0 | 0 | 0 | 0 | 0 | 0 |
| 4: PRE_SUM | 0 | 0 | 0 | 0 | 0 | 0 | 0 | 0 | 0 | 0 | 0 | 0 | 0 | 0 | 0 |
| 5: PRE_CARRY | 0 | 0 | 0 | 0 | 0 | 0 | 0 | 0 | 0 | 0 | 0 | 0 | 0 | 0 | 0 |
| 6: PRE_ARROW | 0 | 0 | 0 | 0 | 0 | 0 | 0 | 0 | 0 | 0 | 0 | 0 | 0 | 0 | 0 |
| 7: PRE_EOS | 0 | 0 | 0 | 0 | 0 | 0 | 0 | 0 | 0 | 0 | 0 | 0 | 0 | 0 | 0 |
| 8-17: SUM | $\boldsymbol{0}_{10}$ | $\boldsymbol{0}_{10}$ | $\boldsymbol{0}_{10}$ | $\boldsymbol{0}_{10}$ | $\boldsymbol{0}_{10}$ | $\boldsymbol{0}_{10}$ | $\boldsymbol{0}_{10}$ | $\boldsymbol{0}_{10}$ | $\boldsymbol{0}_{10}$ | $\boldsymbol{0}_{10}$ | $\boldsymbol{0}_{10}$ | $\boldsymbol{0}_{10}$ | $\boldsymbol{0}_{10}$ | $\boldsymbol{0}_{10}$ | $\boldsymbol{0}_{10}$ |
| 18: ARROW | 0 | 0 | 0 | 0 | 0 | 0 | 0 | 0 | 0 | 0 | 0 | 0 | 0 | 0 | 0 |
| 19: EOS | 0 | 0 | 0 | 0 | 0 | 0 | 0 | 0 | 0 | 0 | 0 | 0 | 0 | 0 | 0 |
| 20–end: | $\boldsymbol{0}_{2P}$ | $\boldsymbol{0}_{2P}$ | $\boldsymbol{0}_{2P}$ | $\boldsymbol{0}_{2P}$ | $\boldsymbol{0}_{2P}$ | $\boldsymbol{0}_{2P}$ | $\boldsymbol{0}_{2P}$ | $\boldsymbol{0}_{2P}$ | $\boldsymbol{0}_{2P}$ | $\boldsymbol{0}_{2P}$ | $\boldsymbol{0}_{2P}$ | $\boldsymbol{0}_{2P}$ | $\boldsymbol{0}_{2P}$ | $\boldsymbol{0}_{2P}$ | $\boldsymbol{0}_{2P}$ |

Table 16: Example of $\boldsymbol{U}_3\boldsymbol{V}_3\boldsymbol{X}^{(0)}\boldsymbol{A}_3$, continuing from Table 4. For simplicity, we omit the columns corresponding to the tokens preceding the equal sign '=', as they do not influence the next-token prediction in the response.

| $\mathcal{I}$ | = | 0 | 0 | 0 | $\rightarrow$ | 7 | 5 | 0 | $\rightarrow$ | 5 | 0 | 1 | $\rightarrow$ | 1 | 0 | 2 |
|---|---|---|---|---|---|---|---|---|---|---|---|---|---|---|---|---|
| 1: NUM | 0 | 0 | 0 | 0 | 0 | 0 | 0 | 0 | 0 | 0 | 0 | 0 | 0 | 0 | 0 | 0 |
| 2: IS_BOS | 0 | 0 | 0 | 0 | 0 | 0 | 0 | 0 | 0 | 0 | 0 | 0 | 0 | 0 | 0 | 0 |
| 3: FULL_ONES | 0 | 0 | 0 | 0 | 0 | 0 | 0 | 0 | 0 | 0 | 0 | 0 | 0 | 0 | 0 | 0 |
| 4: PRE_SUM | 0 | 0 | 0 | 0 | 0 | 0 | 0 | 0 | 0 | 0 | 0 | 0 | 0 | 0 | 0 | 0 |
| 5: PRE_CARRY | 0 | 0 | 0 | 0 | 0 | 0 | 0 | 0 | 0 | 0 | 0 | 0 | 0 | 0 | 0 | 0 |
| 6: PRE_ARROW | 1/4 | 1/4 | 1/4 | 1 | 1/5 | 1/5 | 1/5 | 1 | 1/6 | 1/6 | 1/6 | 1 | 1/7 | 1/7 | 1/7 | 1 |
| 7: PRE_EOS | 0 | 0 | 0 | 0 | 0 | 0 | 0 | 0 | 0 | 0 | 0 | 0 | 0 | 0 | 0 | 0 |
| 8-17: SUM | $\boldsymbol{0}_{10}$ | $\boldsymbol{0}_{10}$ | $\boldsymbol{0}_{10}$ | $\boldsymbol{0}_{10}$ | $\boldsymbol{0}_{10}$ | $\boldsymbol{0}_{10}$ | $\boldsymbol{0}_{10}$ | $\boldsymbol{0}_{10}$ | $\boldsymbol{0}_{10}$ | $\boldsymbol{0}_{10}$ | $\boldsymbol{0}_{10}$ | $\boldsymbol{0}_{10}$ | $\boldsymbol{0}_{10}$ | $\boldsymbol{0}_{10}$ | $\boldsymbol{0}_{10}$ | $\boldsymbol{0}_{10}$ |
| 18: ARROW | 0 | 0 | 0 | 0 | 0 | 0 | 0 | 0 | 0 | 0 | 0 | 0 | 0 | 0 | 0 | 0 |
| 19: EOS | 0 | 0 | 0 | 0 | 0 | 0 | 0 | 0 | 0 | 0 | 0 | 0 | 0 | 0 | 0 | 0 |
| 20–end: | $\boldsymbol{0}_{2P}$ | $\boldsymbol{0}_{2P}$ | $\boldsymbol{0}_{2P}$ | $\boldsymbol{0}_{2P}$ | $\boldsymbol{0}_{2P}$ | $\boldsymbol{0}_{2P}$ | $\boldsymbol{0}_{2P}$ | $\boldsymbol{0}_{2P}$ | $\boldsymbol{0}_{2P}$ | $\boldsymbol{0}_{2P}$ | $\boldsymbol{0}_{2P}$ | $\boldsymbol{0}_{2P}$ | $\boldsymbol{0}_{2P}$ | $\boldsymbol{0}_{2P}$ | $\boldsymbol{0}_{2P}$ | $\boldsymbol{0}_{2P}$ |

### D.5.4 ATTENTION HEAD 4: EOS DETECTION

The goal of the fourth attention head is to fill the dimension PRE_EOS. We aim to put $1/2$ if the next token the model has to predict is the EOS token.

Recall that $P = P_1 + P_2$ and $d = 2P + 19$. Let the dimension of the second attention head be $d_{QK,4} = P + 1$. We define the query matrix $\boldsymbol{Q}_4$ and the key matrix $\boldsymbol{K}_4$ as follows:

$$\boldsymbol{Q}_4 = \begin{pmatrix} \boldsymbol{0}_{P_1 \times 19} & \sqrt{\alpha_4}\boldsymbol{I}_{P_1} & \boldsymbol{0}_{P_1 \times P_1} & \boldsymbol{0}_{P_1 \times P_2} & \boldsymbol{0}_{P_1 \times P_2} \\ \boldsymbol{0}_{P_2 \times 19} & \boldsymbol{0}_{P_2 \times P_1} & \boldsymbol{0}_{P_2 \times P_1} & \sqrt{\alpha_4}\boldsymbol{I}_{P_2} & \boldsymbol{0}_{P_2 \times P_2} \\ \sqrt{\alpha_4 P}(\boldsymbol{e}_{\text{FULL\_ONES}}^{19})^\top & \boldsymbol{0}_{1 \times P_1} & \boldsymbol{0}_{1 \times P_1} & \boldsymbol{0}_{1 \times P_2} & \boldsymbol{0}_{1 \times P_2} \end{pmatrix} \in \mathbb{R}^{d_{QK,4} \times d}, \quad (19)$$

$$\boldsymbol{K}_4 = \begin{pmatrix} \boldsymbol{0}_{P_1 \times 19} & \sqrt{\alpha_4}\boldsymbol{I}_{P_1} & \boldsymbol{0}_{P_1 \times P_1} & \boldsymbol{0}_{P_1 \times P_2} & \boldsymbol{0}_{P_1 \times P_2} \\ \boldsymbol{0}_{P_2 \times 19} & \boldsymbol{0}_{P_2 \times P_1} & \boldsymbol{0}_{P_2 \times P_1} & \sqrt{\alpha_4}\boldsymbol{I}_{P_2} & \boldsymbol{0}_{P_2 \times P_2} \\ \sqrt{\alpha_4 P}(\boldsymbol{e}_{\text{IS\_BOS}}^{19})^\top & \boldsymbol{0}_{1 \times P_1} & \boldsymbol{0}_{1 \times P_1} & \boldsymbol{0}_{1 \times P_2} & \boldsymbol{0}_{1 \times P_2} \end{pmatrix} \in \mathbb{R}^{d_{QK,4} \times d}. \quad (20)$$

where $\alpha_4$ is a scaling factor, which we can choose to be arbitrarily large. Also, let $d_{V,4} = 1$ and define

$$\boldsymbol{V}_4 = (\boldsymbol{e}^d_{\text{IS\_BOS}})^\top \in \mathbb{R}^{d_{V,4} \times d}, \tag{21}$$

$$\boldsymbol{U}_4 = \boldsymbol{e}^d_{\text{PRE\_EOS}} \in \mathbb{R}^{d \times d_{V,4}}. \tag{22}$$

An example of $\boldsymbol{Q}_4 \boldsymbol{X}^{(0)}$, $\boldsymbol{K}_4 \boldsymbol{X}^{(0)}$, $\boldsymbol{U}_4 \boldsymbol{V}_4 \boldsymbol{X}^{(0)}$, and $\boldsymbol{U}_4 \boldsymbol{V}_4 \boldsymbol{X}^{(0)} \boldsymbol{A}_4$ is illustrated in Tables 17 to 20. Like the previous attention heads, with sufficiently large $\alpha_4$, the $j$-token $\sigma_j$ will only attend to the BOS token and tokens whose key vectors match the query vector of $\sigma_j$.

As mentioned earlier, the fourth head aims to fill $1/2$ in the dimension PRE_EOS if the model has to predict the EOS token as the next token. However, it might not be very clear that such a token is not uniquely defined, as multiple tokens in the matrix $\boldsymbol{U}_4 \boldsymbol{V}_4 \boldsymbol{X}^{(0)} \boldsymbol{A}_4$ can have their PRE_EOS entry filled with $1/2$, as illustrated in Table 20. To clarify, **we note that the final decision regarding the EOS token is made by combining the outputs from both the third and the fourth attention head (readers can check this in the subsequent feed-forward layer construction)**. This approach enables the model to correctly determine whether the next token should be the EOS token or not, as the token with PRE_ARROW and PRE_EOS set to $1$ and $1/2$ is uniquely identified. We also note that PRE_EOS can only be either $1/2$ or $1/3$, which will be utilized in the feed-forward layer construction.

Table 17: Example of $\frac{1}{\sqrt{\alpha_4}} \boldsymbol{Q}_4 \boldsymbol{X}^{(0)}$, continuing from Table 4.

| $\mathcal{I}$ | \$ | 0 | 5 | 7 | + | 0 | 4 | 8 | + | 0 | 9 | 6 | = | $\cdots$ |
|---|---|---|---|---|---|---|---|---|---|---|---|---|---|---|
| $1-P_1$: | $\boldsymbol{0}_{P_1}$ | $\boldsymbol{v}_7^{P_1}$ | $\boldsymbol{v}_6^{P_1}$ | $\boldsymbol{v}_5^{P_1}$ | $\boldsymbol{v}_4^{P_1}$ | $\boldsymbol{v}_7^{P_1}$ | $\boldsymbol{v}_6^{P_1}$ | $\boldsymbol{v}_5^{P_1}$ | $\boldsymbol{v}_4^{P_1}$ | $\boldsymbol{v}_7^{P_1}$ | $\boldsymbol{v}_6^{P_1}$ | $\boldsymbol{v}_5^{P_1}$ | $\boldsymbol{v}_4^{P_1}$ | |
| $(P_1+1)-P$: | $\boldsymbol{0}_{P_2}$ | $\boldsymbol{v}_2^{P_2}$ | $\boldsymbol{v}_2^{P_2}$ | $\boldsymbol{v}_2^{P_2}$ | $\boldsymbol{v}_2^{P_2}$ | $\boldsymbol{v}_3^{P_2}$ | $\boldsymbol{v}_3^{P_2}$ | $\boldsymbol{v}_3^{P_2}$ | $\boldsymbol{v}_3^{P_2}$ | $\boldsymbol{v}_4^{P_2}$ | $\boldsymbol{v}_4^{P_2}$ | $\boldsymbol{v}_4^{P_2}$ | $\boldsymbol{v}_2^{P_2}$ | |
| $P+1$: | $\sqrt{P}$ | $\sqrt{P}$ | $\sqrt{P}$ | $\sqrt{P}$ | $\sqrt{P}$ | $\sqrt{P}$ | $\sqrt{P}$ | $\sqrt{P}$ | $\sqrt{P}$ | $\sqrt{P}$ | $\sqrt{P}$ | $\sqrt{P}$ | $\sqrt{P}$ | |

| $\mathcal{I}$ | 0 | 0 | 0 | $\rightarrow$ | 7 | 5 | 0 | $\rightarrow$ | 5 | 0 | 1 | $\rightarrow$ | 1 | 0 | 2 |
|---|---|---|---|---|---|---|---|---|---|---|---|---|---|---|---|
| $1-P_1$: | $\boldsymbol{v}_5^{P_1}$ | $\boldsymbol{v}_6^{P_1}$ | $\boldsymbol{v}_7^{P_1}$ | $\boldsymbol{v}_4^{P_1}$ | $\boldsymbol{v}_5^{P_1}$ | $\boldsymbol{v}_6^{P_1}$ | $\boldsymbol{v}_7^{P_1}$ | $\boldsymbol{v}_4^{P_1}$ | $\boldsymbol{v}_5^{P_1}$ | $\boldsymbol{v}_6^{P_1}$ | $\boldsymbol{v}_7^{P_1}$ | $\boldsymbol{v}_4^{P_1}$ | $\boldsymbol{v}_5^{P_1}$ | $\boldsymbol{v}_6^{P_1}$ | $\boldsymbol{v}_7^{P_1}$ |
| $(P_1+1)-P$: | $\boldsymbol{v}_2^{P_2}$ | $\boldsymbol{v}_2^{P_2}$ | $\boldsymbol{v}_2^{P_2}$ | $\boldsymbol{v}_3^{P_2}$ | $\boldsymbol{v}_3^{P_2}$ | $\boldsymbol{v}_3^{P_2}$ | $\boldsymbol{v}_3^{P_2}$ | $\boldsymbol{v}_4^{P_2}$ | $\boldsymbol{v}_4^{P_2}$ | $\boldsymbol{v}_4^{P_2}$ | $\boldsymbol{v}_4^{P_2}$ | $\boldsymbol{v}_5^{P_2}$ | $\boldsymbol{v}_5^{P_2}$ | $\boldsymbol{v}_5^{P_2}$ | $\boldsymbol{v}_5^{P_2}$ |
| $P+1$: | $\sqrt{P}$ | $\sqrt{P}$ | $\sqrt{P}$ | $\sqrt{P}$ | $\sqrt{P}$ | $\sqrt{P}$ | $\sqrt{P}$ | $\sqrt{P}$ | $\sqrt{P}$ | $\sqrt{P}$ | $\sqrt{P}$ | $\sqrt{P}$ | $\sqrt{P}$ | $\sqrt{P}$ | $\sqrt{P}$ |

Table 18: Example of $\frac{1}{\sqrt{\alpha_4}} \boldsymbol{K}_4 \boldsymbol{X}^{(0)}$, continuing from Table 4.

| $\mathcal{I}$ | \$ | 0 | 5 | 7 | + | 0 | 4 | 8 | + | 0 | 9 | 6 | = | $\cdots$ |
|---|---|---|---|---|---|---|---|---|---|---|---|---|---|---|
| $1-P_1$: | $\boldsymbol{0}_{P_1}$ | $\boldsymbol{v}_7^{P_1}$ | $\boldsymbol{v}_6^{P_1}$ | $\boldsymbol{v}_5^{P_1}$ | $\boldsymbol{v}_4^{P_1}$ | $\boldsymbol{v}_7^{P_1}$ | $\boldsymbol{v}_6^{P_1}$ | $\boldsymbol{v}_5^{P_1}$ | $\boldsymbol{v}_4^{P_1}$ | $\boldsymbol{v}_7^{P_1}$ | $\boldsymbol{v}_6^{P_1}$ | $\boldsymbol{v}_5^{P_1}$ | $\boldsymbol{v}_4^{P_1}$ | |
| $(P_1+1)-P$: | $\boldsymbol{0}_{P_2}$ | $\boldsymbol{v}_2^{P_2}$ | $\boldsymbol{v}_2^{P_2}$ | $\boldsymbol{v}_2^{P_2}$ | $\boldsymbol{v}_2^{P_2}$ | $\boldsymbol{v}_3^{P_2}$ | $\boldsymbol{v}_3^{P_2}$ | $\boldsymbol{v}_3^{P_2}$ | $\boldsymbol{v}_3^{P_2}$ | $\boldsymbol{v}_4^{P_2}$ | $\boldsymbol{v}_4^{P_2}$ | $\boldsymbol{v}_4^{P_2}$ | $\boldsymbol{v}_2^{P_2}$ | |
| $P+1$: | $\sqrt{P}$ | 0 | 0 | 0 | 0 | 0 | 0 | 0 | 0 | 0 | 0 | 0 | 0 | |

| $\mathcal{I}$ | 0 | 0 | 0 | $\rightarrow$ | 7 | 5 | 0 | $\rightarrow$ | 5 | 0 | 1 | $\rightarrow$ | 1 | 0 | 2 |
|---|---|---|---|---|---|---|---|---|---|---|---|---|---|---|---|
| $1-P_1$: | $\boldsymbol{v}_5^{P_1}$ | $\boldsymbol{v}_6^{P_1}$ | $\boldsymbol{v}_7^{P_1}$ | $\boldsymbol{v}_4^{P_1}$ | $\boldsymbol{v}_5^{P_1}$ | $\boldsymbol{v}_6^{P_1}$ | $\boldsymbol{v}_7^{P_1}$ | $\boldsymbol{v}_4^{P_1}$ | $\boldsymbol{v}_5^{P_1}$ | $\boldsymbol{v}_6^{P_1}$ | $\boldsymbol{v}_7^{P_1}$ | $\boldsymbol{v}_4^{P_1}$ | $\boldsymbol{v}_5^{P_1}$ | $\boldsymbol{v}_6^{P_1}$ | $\boldsymbol{v}_7^{P_1}$ |
| $(P_1+1)-P$: | $\boldsymbol{v}_2^{P_2}$ | $\boldsymbol{v}_2^{P_2}$ | $\boldsymbol{v}_2^{P_2}$ | $\boldsymbol{v}_3^{P_2}$ | $\boldsymbol{v}_3^{P_2}$ | $\boldsymbol{v}_3^{P_2}$ | $\boldsymbol{v}_3^{P_2}$ | $\boldsymbol{v}_4^{P_2}$ | $\boldsymbol{v}_4^{P_2}$ | $\boldsymbol{v}_4^{P_2}$ | $\boldsymbol{v}_4^{P_2}$ | $\boldsymbol{v}_5^{P_2}$ | $\boldsymbol{v}_5^{P_2}$ | $\boldsymbol{v}_5^{P_2}$ | $\boldsymbol{v}_5^{P_2}$ |
| $P+1$: | 0 | 0 | 0 | 0 | 0 | 0 | 0 | 0 | 0 | 0 | 0 | 0 | 0 | 0 | 0 |

Table 19: Example of $U_4 V_4 X^{(0)}$, continuing from Table 4. For simplicity, we omit the columns corresponding to the tokens preceding the equal sign '=', as they do not influence the next-token prediction in the response.

| $\mathcal{I}$ | \$ | 0 | 5 | 7 | + | 0 | 4 | 8 | + | 0 | 9 | 6 | = | $\cdots$ |
|---|---|---|---|---|---|---|---|---|---|---|---|---|---|---|
| 1: NUM | 0 | 0 | 0 | 0 | 0 | 0 | 0 | 0 | 0 | 0 | 0 | 0 | 0 | |
| 2: IS_BOS | 0 | 0 | 0 | 0 | 0 | 0 | 0 | 0 | 0 | 0 | 0 | 0 | 0 | |
| 3: FULL_ONES | 0 | 0 | 0 | 0 | 0 | 0 | 0 | 0 | 0 | 0 | 0 | 0 | 0 | |
| 4: PRE_SUM | 0 | 0 | 0 | 0 | 0 | 0 | 0 | 0 | 0 | 0 | 0 | 0 | 0 | |
| 5: PRE_CARRY | 0 | 0 | 0 | 0 | 0 | 0 | 0 | 0 | 0 | 0 | 0 | 0 | 0 | |
| 6: PRE_ARROW | 0 | 0 | 0 | 0 | 0 | 0 | 0 | 0 | 0 | 0 | 0 | 0 | 0 | |
| 7: PRE_EOS | 1 | 0 | 0 | 0 | 0 | 0 | 0 | 0 | 0 | 0 | 0 | 0 | 0 | |
| 8-17: SUM | $\mathbf{0}_{10}$ | $\mathbf{0}_{10}$ | $\mathbf{0}_{10}$ | $\mathbf{0}_{10}$ | $\mathbf{0}_{10}$ | $\mathbf{0}_{10}$ | $\mathbf{0}_{10}$ | $\mathbf{0}_{10}$ | $\mathbf{0}_{10}$ | $\mathbf{0}_{10}$ | $\mathbf{0}_{10}$ | $\mathbf{0}_{10}$ | $\mathbf{0}_{10}$ | |
| 18: ARROW | 0 | 0 | 0 | 0 | 0 | 0 | 0 | 0 | 0 | 0 | 0 | 0 | 0 | |
| 19: EOS | 0 | 0 | 0 | 0 | 0 | 0 | 0 | 0 | 0 | 0 | 0 | 0 | 0 | |
| 20–end: | $\mathbf{0}_{2P}$ | $\mathbf{0}_{2P}$ | $\mathbf{0}_{2P}$ | $\mathbf{0}_{2P}$ | $\mathbf{0}_{2P}$ | $\mathbf{0}_{2P}$ | $\mathbf{0}_{2P}$ | $\mathbf{0}_{2P}$ | $\mathbf{0}_{2P}$ | $\mathbf{0}_{2P}$ | $\mathbf{0}_{2P}$ | $\mathbf{0}_{2P}$ | $\mathbf{0}_{2P}$ | |

| $\mathcal{I}$ | 0 | 0 | 0 | $\rightarrow$ | 7 | 5 | 0 | $\rightarrow$ | 5 | 0 | 1 | $\rightarrow$ | 1 | 0 | 2 |
|---|---|---|---|---|---|---|---|---|---|---|---|---|---|---|---|
| 1: NUM | 0 | 0 | 0 | 0 | 0 | 0 | 0 | 0 | 0 | 0 | 0 | 0 | 0 | 0 | 0 |
| 2: IS_BOS | 0 | 0 | 0 | 0 | 0 | 0 | 0 | 0 | 0 | 0 | 0 | 0 | 0 | 0 | 0 |
| 3: FULL_ONES | 0 | 0 | 0 | 0 | 0 | 0 | 0 | 0 | 0 | 0 | 0 | 0 | 0 | 0 | 0 |
| 4: PRE_SUM | 0 | 0 | 0 | 0 | 0 | 0 | 0 | 0 | 0 | 0 | 0 | 0 | 0 | 0 | 0 |
| 5: PRE_CARRY | 0 | 0 | 0 | 0 | 0 | 0 | 0 | 0 | 0 | 0 | 0 | 0 | 0 | 0 | 0 |
| 6: PRE_ARROW | 0 | 0 | 0 | 0 | 0 | 0 | 0 | 0 | 0 | 0 | 0 | 0 | 0 | 0 | 0 |
| 7: PRE_EOS | 0 | 0 | 0 | 0 | 0 | 0 | 0 | 0 | 0 | 0 | 0 | 0 | 0 | 0 | 0 |
| 8-17: SUM | $\mathbf{0}_{10}$ | $\mathbf{0}_{10}$ | $\mathbf{0}_{10}$ | $\mathbf{0}_{10}$ | $\mathbf{0}_{10}$ | $\mathbf{0}_{10}$ | $\mathbf{0}_{10}$ | $\mathbf{0}_{10}$ | $\mathbf{0}_{10}$ | $\mathbf{0}_{10}$ | $\mathbf{0}_{10}$ | $\mathbf{0}_{10}$ | $\mathbf{0}_{10}$ | $\mathbf{0}_{10}$ | $\mathbf{0}_{10}$ |
| 18: ARROW | 0 | 0 | 0 | 0 | 0 | 0 | 0 | 0 | 0 | 0 | 0 | 0 | 0 | 0 | 0 |
| 19: EOS | 0 | 0 | 0 | 0 | 0 | 0 | 0 | 0 | 0 | 0 | 0 | 0 | 0 | 0 | 0 |
| 20–end: | $\mathbf{0}_{2P}$ | $\mathbf{0}_{2P}$ | $\mathbf{0}_{2P}$ | $\mathbf{0}_{2P}$ | $\mathbf{0}_{2P}$ | $\mathbf{0}_{2P}$ | $\mathbf{0}_{2P}$ | $\mathbf{0}_{2P}$ | $\mathbf{0}_{2P}$ | $\mathbf{0}_{2P}$ | $\mathbf{0}_{2P}$ | $\mathbf{0}_{2P}$ | $\mathbf{0}_{2P}$ | $\mathbf{0}_{2P}$ | $\mathbf{0}_{2P}$ |

Table 20: Example of $U_4 V_4 X^{(0)} A_4$, continuing from Table 4. For simplicity, we omit the columns corresponding to the tokens preceding the equal sign '=', as they do not influence the next-token prediction in the response.

| $\mathcal{I}$ | = | 0 | 0 | 0 | $\rightarrow$ | 7 | 5 | 0 | $\rightarrow$ | 5 | 0 | 1 | $\rightarrow$ | 1 | 0 | 2 |
|---|---|---|---|---|---|---|---|---|---|---|---|---|---|---|---|---|
| 1: NUM | 0 | 0 | 0 | 0 | 0 | 0 | 0 | 0 | 0 | 0 | 0 | 0 | 0 | 0 | 0 | 0 |
| 2: IS_BOS | 0 | 0 | 0 | 0 | 0 | 0 | 0 | 0 | 0 | 0 | 0 | 0 | 0 | 0 | 0 | 0 |
| 3: FULL_ONES | 0 | 0 | 0 | 0 | 0 | 0 | 0 | 0 | 0 | 0 | 0 | 0 | 0 | 0 | 0 | 0 |
| 4: PRE_SUM | 0 | 0 | 0 | 0 | 0 | 0 | 0 | 0 | 0 | 0 | 0 | 0 | 0 | 0 | 0 | 0 |
| 5: PRE_CARRY | 0 | 0 | 0 | 0 | 0 | 0 | 0 | 0 | 0 | 0 | 0 | 0 | 0 | 0 | 0 | 0 |
| 6: PRE_ARROW | 0 | 0 | 0 | 0 | 0 | 0 | 0 | 0 | 0 | 0 | 0 | 0 | 0 | 0 | 0 | 0 |
| 7: PRE_EOS | 1/3 | 1/3 | 1/3 | 1/3 | 1/3 | 1/3 | 1/3 | 1/3 | 1/2 | 1/3 | 1/3 | 1/3 | 1/2 | 1/2 | 1/2 | 1/2 |
| 8-17: SUM | $\mathbf{0}_{10}$ | $\mathbf{0}_{10}$ | $\mathbf{0}_{10}$ | $\mathbf{0}_{10}$ | $\mathbf{0}_{10}$ | $\mathbf{0}_{10}$ | $\mathbf{0}_{10}$ | $\mathbf{0}_{10}$ | $\mathbf{0}_{10}$ | $\mathbf{0}_{10}$ | $\mathbf{0}_{10}$ | $\mathbf{0}_{10}$ | $\mathbf{0}_{10}$ | $\mathbf{0}_{10}$ | $\mathbf{0}_{10}$ | $\mathbf{0}_{10}$ |
| 18: ARROW | 0 | 0 | 0 | 0 | 0 | 0 | 0 | 0 | 0 | 0 | 0 | 0 | 0 | 0 | 0 | 0 |
| 19: EOS | 0 | 0 | 0 | 0 | 0 | 0 | 0 | 0 | 0 | 0 | 0 | 0 | 0 | 0 | 0 | 0 |
| 20–end: | $\mathbf{0}_{2P}$ | $\mathbf{0}_{2P}$ | $\mathbf{0}_{2P}$ | $\mathbf{0}_{2P}$ | $\mathbf{0}_{2P}$ | $\mathbf{0}_{2P}$ | $\mathbf{0}_{2P}$ | $\mathbf{0}_{2P}$ | $\mathbf{0}_{2P}$ | $\mathbf{0}_{2P}$ | $\mathbf{0}_{2P}$ | $\mathbf{0}_{2P}$ | $\mathbf{0}_{2P}$ | $\mathbf{0}_{2P}$ | $\mathbf{0}_{2P}$ | $\mathbf{0}_{2P}$ |

### D.5.5 RESIDUAL CONNECTION

We now consider the residual connection. The output of the attention layer can be expressed as follows:

$$Y^{(1)} = X^{(0)} + \sum_{h \in \{1,2,3,4\}} U_h V_h X^{(0)} A_h \tag{23}$$

One can observe that the dimensions PRE_SUM, PRE_CARRY, PRE_ARROW, and PRE_EOS in $X^{(0)}$ are empty, whereas these same dimensions in $\sum_{h \in \{1,2,3,4\}} U_h V_h X^{(0)} A_h$ contain non-empty values. Thus, the residual connection effectively "fills in the blanks" in the input embedding matrix. An example of the output of residual connection is presented in Table 21. Again, we note that the

error from the `softmax` operation can be made negligible by setting the scalars $\alpha_1$, $\alpha_2$, $\alpha_3$, and $\alpha_4$ sufficiently large.

Table 21: Example output of residual connection, $\boldsymbol{Y}^{(1)}$, continuing from Tables 4, 8, 12, 16 and 20. The orange rows correspond to the values computed through the attention layer. We omit the columns corresponding to the tokens preceding the equal sign '=', as they do not influence the next-token prediction in the response. The gray rows will be filled during the subsequent feed-forward layer.

| $\mathcal{I}$ | = | 0 | 0 | 0 | → | 7 | 5 | 0 | → | 5 | 0 | 1 | → | 1 | 0 | 2 |
|---|---|---|---|---|---|---|---|---|---|---|---|---|---|---|---|---|
| 1: NUM | 0 | 0 | 0 | 0 | 0 | 7 | 5 | 0 | 0 | 5 | 0 | 1 | 0 | 1 | 0 | 2 |
| 2: IS_BOS | 0 | 0 | 0 | 0 | 0 | 0 | 0 | 0 | 0 | 0 | 0 | 0 | 0 | 0 | 0 | 0 |
| 3: FULL_ONES | 1 | 1 | 1 | 1 | 1 | 1 | 1 | 1 | 1 | 1 | 1 | 1 | 1 | 1 | 1 | 1 |
| 4: PRE_SUM | 0 | 0 | 0 | 0 | 7 | 5 | 0 | 0 | 15 | 9 | 0 | 0 | 11 | 9 | 1 | 0 |
| 5: PRE_CARRY | 0 | 0 | 0 | 0 | 0 | 7 | 5 | 0 | 0 | 15 | 9 | 0 | 0 | 11 | 9 | 1 |
| 6: PRE_ARROW | 1/4 | 1/4 | 1/4 | 1 | 1/5 | 1/5 | 1/5 | 1 | 1/6 | 1/6 | 1/6 | 1 | 1/7 | 1/7 | 1/7 | 1 |
| 7: PRE_EOS | 1/3 | 1/3 | 1/3 | 1/3 | 1/3 | 1/3 | 1/3 | 1/3 | 1/2 | 1/3 | 1/3 | 1/3 | 1/2 | 1/2 | 1/2 | 1/2 |
| 8-17: SUM | $\mathbf{0}_{10}$ | $\mathbf{0}_{10}$ | $\mathbf{0}_{10}$ | $\mathbf{0}_{10}$ | $\mathbf{0}_{10}$ | $\mathbf{0}_{10}$ | $\mathbf{0}_{10}$ | $\mathbf{0}_{10}$ | $\mathbf{0}_{10}$ | $\mathbf{0}_{10}$ | $\mathbf{0}_{10}$ | $\mathbf{0}_{10}$ | $\mathbf{0}_{10}$ | $\mathbf{0}_{10}$ | $\mathbf{0}_{10}$ | $\mathbf{0}_{10}$ |
| 18: ARROW | 0 | 0 | 0 | 0 | 0 | 0 | 0 | 0 | 0 | 0 | 0 | 0 | 0 | 0 | 0 | 0 |
| 19: EOS | 0 | 0 | 0 | 0 | 0 | 0 | 0 | 0 | 0 | 0 | 0 | 0 | 0 | 0 | 0 | 0 |
| POS_1 | $v_4^{P_1}$ | $v_5^{P_1}$ | $v_6^{P_1}$ | $v_7^{P_1}$ | $v_4^{P_1}$ | $v_5^{P_1}$ | $v_6^{P_1}$ | $v_7^{P_1}$ | $v_4^{P_1}$ | $v_5^{P_1}$ | $v_6^{P_1}$ | $v_7^{P_1}$ | $v_4^{P_1}$ | $v_5^{P_1}$ | $v_6^{P_1}$ | $v_7^{P_1}$ |
| POS_1_NEXT | $v_5^{P_1}$ | $v_6^{P_1}$ | $v_7^{P_1}$ | $v_8^{P_1}$ | $v_5^{P_1}$ | $v_6^{P_1}$ | $v_7^{P_1}$ | $v_8^{P_1}$ | $v_5^{P_1}$ | $v_6^{P_1}$ | $v_7^{P_1}$ | $v_8^{P_1}$ | $v_5^{P_1}$ | $v_6^{P_1}$ | $v_7^{P_1}$ | $v_8^{P_1}$ |
| POS_2 | $v_2^{P_2}$ | $v_2^{P_2}$ | $v_2^{P_2}$ | $v_2^{P_2}$ | $v_3^{P_2}$ | $v_3^{P_2}$ | $v_3^{P_2}$ | $v_3^{P_2}$ | $v_4^{P_2}$ | $v_4^{P_2}$ | $v_4^{P_2}$ | $v_4^{P_2}$ | $v_5^{P_2}$ | $v_5^{P_2}$ | $v_5^{P_2}$ | $v_5^{P_2}$ |
| POS_2_NEXT | $v_3^{P_2}$ | $v_3^{P_2}$ | $v_3^{P_2}$ | $v_3^{P_2}$ | $v_4^{P_2}$ | $v_4^{P_2}$ | $v_4^{P_2}$ | $v_4^{P_2}$ | $v_5^{P_2}$ | $v_5^{P_2}$ | $v_5^{P_2}$ | $v_5^{P_2}$ | $v_6^{P_2}$ | $v_6^{P_2}$ | $v_6^{P_2}$ | $v_6^{P_2}$ |

## D.6 TRANSFORMER BLOCK: TOKEN-WISE FEED-FORWARD LAYER

The goal of the feed-forward layer is to fill the dimensions SUM, ARROW, and EOS. While the attention layer enables interactions between different tokens, the feed-forward layer operates solely on the dimensions within each token. After processing, SUM specifies the number the model will predict as the next token, while ARROW and EOS serve as flags of whether the next token should be an arrow ($\rightarrow$) or the EOS token ($\$$), respectively. Based on the output of the feed-forward layer, the next-token prediction is carried out in a subsequent linear readout process.

We will construct 3 one-hidden-layer ReLU networks ($\mathrm{FF}_1^1$, $\mathrm{FF}_1^2$, $\mathrm{FF}_1^3$) that take inputs from dimensions 1 to 7. Each network will handle a specific output: SUM (8–17), ARROW (18), and EOS (19), respectively. The goals of each network are as follows:

- $\mathrm{FF}_1^1$: If the model has to predict a digit (let's say $k \in \{0, 1, \ldots, 9\}$), it assigns the value of 1 to the dimension $8 + k$, which is the $(k + 1)$-th dimension of SUM, while setting all other dimensions in SUM to 0.

- $\mathrm{FF}_1^2$: If the model has to predict the arrow ($\rightarrow$) as the next token, set ARROW by 1.

- $\mathrm{FF}_1^3$: If the model has to predict the EOS token as the next token, set EOS by 1.

By combining these three sub-networks, we can construct a single one-hidden-layer ReLU network $\mathrm{FF}_1$, that takes inputs from dimensions 1 to 7 and outputs the proper values at dimensions 8 to 19. We provide our example in Table 22.

### D.6.1 SUBNETWORK 1: CONSTRUCTION FOR SUM (DIMENSION 8–17)

For $\mathrm{FF}_1^1$, we can apply the construction provided in Section E.5.1 of Cho et al. (2024). Below, we provide an overview of their approach, and refer readers to Cho et al. (2024) for the underlying principles.

Since the feed-forward layer processes tokens individually, we denote each column vector of $\boldsymbol{Y}^{(1)}$ as $\boldsymbol{x} \in \mathbb{R}^d$ in a unified manner. We first define a linear function $g : \mathbb{R}^d \to \mathbb{R}$ as:

$$g(\boldsymbol{x}) := \boldsymbol{x}_{\text{PRE\_SUM}} + \frac{\boldsymbol{x}_{\text{PRE\_CARRY}} - \boldsymbol{x}_{\text{NUM}}}{10} + 0.21.$$

Using $g$, we construct a one-hidden-layer ReLU network $f_k : \mathbb{R} \to \mathbb{R}$ ($k = 0, 1, \ldots, 9$) defined as

$$f_k(x) = 2\Big[\phi(x - (k - 0.5)) - \phi(x - k) - \phi(x - (k + 0.5)) + \phi(x - (k + 1))$$
$$+ \phi(x - (k + 9.5)) - \phi(x - (k + 10)) - \phi(x - (k + 10.5)) + \phi(x - (k + 11))\Big].$$

Now, we construct $\mathtt{FF}_1^1$ as

$$\big[\mathtt{FF}_1^1(\boldsymbol{x})\big]_{\text{SUM}} = [f_0\left(g\left(\boldsymbol{x}\right)\right) \quad \cdots \quad f_9\left(g\left(\boldsymbol{x}\right)\right)]^\top .$$

We note that the construction of $g$ slightly differs from the original formulation in Cho et al. (2024), due to the difference in how value is stored in the PRE_CARRY. Specifically, we store the sum of digits at the (previous) same significance level, whereas Cho et al. (2024) stores the result of additional summation of the value in NUM to the sum of digits.

Intuitively, $\mathtt{FF}_1^1$ is designed to output a one-hot vector that acts as a flag indicating the digit which the model has to predict as the next token. The difference between the value in the PRE_CARRY and NUM takes a role of detecting whether a carry occurs from the previous digit, while $f_k$ identifies whether $x$ corresponds to $k$ or $k + 10$. Certain constants, such as $0.21$ and $0.5$, are chosen to manage numerical errors.

### D.6.2 SUBNETWORK 2: CONSTRUCTION FOR ARROW (DIMENSION 18)

We will construct a subnetwork $\mathtt{FF}_1^2 : \mathbb{R}^d \to \mathbb{R}^d$ that outputs 1 in the dimension ARROW if the dimension PRE_ARROW is set to 1; otherwise, outputs 0. As PRE_ARROW can have a value less than $1/2$ if it is not 1, this can be easily achieved by constructing $\mathtt{FF}_1^2$ as

$$[\mathtt{FF}_1^2(\boldsymbol{x})]_{\text{ARROW}} = 2\phi(\boldsymbol{x}_{\text{PRE\_ARROW}} - 1/2),$$

where $\boldsymbol{x}$ represents each column vector of $\boldsymbol{Y}^{(1)}$.

### D.6.3 SUBNETWORK 3: CONSTRUCTION FOR EOS (DIMENSION 19)

We will construct a subnetwork $\mathtt{FF}_1^3 : \mathbb{R}^d \to \mathbb{R}^d$ that outputs 1 in the dimension EOS if the dimension PRE_ARROW is set to 1 and PRE_EOS is set to $1/2$. We construct $\mathtt{FF}_1^3$ as

$$[\mathtt{FF}_1^3(\boldsymbol{x})]_{\text{EOS}} := 2\phi(\boldsymbol{x}_{\text{PRE\_ARROW}} - 1/2) + 6\phi(\boldsymbol{x}_{\text{PRE\_EOS}} - 1/3) - 1,$$

where $\boldsymbol{x}$ represents each column vector of $\boldsymbol{Y}^{(1)}$. Note that PRE_ARROW can take values of either 1 or less than $1/2$, and PRE_EOS can be either $1/2$ or $1/3$. Therefore, the output of $\mathtt{FF}_1^3$ equals 1 only when the dimension PRE_ARROW is set to 1 and PRE_EOS is set to $1/2$. Importantly, this condition uniquely identifies the token as the most significant digit of $b_m$.

Table 22: Example of the output of the feed-forward layer. The yellow rows represent the values that are generated during the feed-forward operation. We omit the columns corresponding to the tokens preceding the equal token '=', as they do not influence the next-token prediction.

| $\mathcal{I}$ | = | 0 | 0 | 0 | $\to$ | 7 | 5 | 0 | $\to$ | 5 | 0 | 1 | $\to$ | 1 | 0 | 2 |
|---|---|---|---|---|---|---|---|---|---|---|---|---|---|---|---|---|
| 1: NUM | 0 | 0 | 0 | 0 | 0 | 0 | 0 | 0 | 0 | 0 | 0 | 0 | 0 | 0 | 0 | 0 |
| 2: IS_BOS | 0 | 0 | 0 | 0 | 0 | 0 | 0 | 0 | 0 | 0 | 0 | 0 | 0 | 0 | 0 | 0 |
| 3: FULL_ONES | 0 | 0 | 0 | 0 | 0 | 0 | 0 | 0 | 0 | 0 | 0 | 0 | 0 | 0 | 0 | 0 |
| 4: PRE_SUM | 0 | 0 | 0 | 0 | 0 | 0 | 0 | 0 | 0 | 0 | 0 | 0 | 0 | 0 | 0 | 0 |
| 5: PRE_CARRY | 0 | 0 | 0 | 0 | 0 | 0 | 0 | 0 | 0 | 0 | 0 | 0 | 0 | 0 | 0 | 0 |
| 6: PRE_ARROW | 0 | 0 | 0 | 0 | 0 | 0 | 0 | 0 | 0 | 0 | 0 | 0 | 0 | 0 | 0 | 0 |
| 7: PRE_EOS | 0 | 0 | 0 | 0 | 0 | 0 | 0 | 0 | 0 | 0 | 0 | 0 | 0 | 0 | 0 | 0 |
| 8-17: SUM | $\boldsymbol{e}_1^{10}$ | $\boldsymbol{e}_1^{10}$ | $\boldsymbol{e}_1^{10}$ | $\boldsymbol{e}_1^{10}$ | $\boldsymbol{e}_8^{10}$ | $\boldsymbol{e}_6^{10}$ | $\boldsymbol{e}_1^{10}$ | $\boldsymbol{e}_1^{10}$ | $\boldsymbol{e}_6^{10}$ | $\boldsymbol{e}_1^{10}$ | $\boldsymbol{e}_2^{10}$ | $\boldsymbol{e}_1^{10}$ | $\boldsymbol{e}_2^{10}$ | $\boldsymbol{e}_1^{10}$ | $\boldsymbol{e}_3^{10}$ | $\boldsymbol{e}_1^{10}$ |
| 18: ARROW | 0 | 0 | 0 | 1 | 0 | 0 | 0 | 1 | 0 | 0 | 0 | 1 | 0 | 0 | 0 | 1 |
| 19: EOS | 0 | -1 | -1 | 0 | -1 | -1 | -1 | 0 | -1 | -1 | -1 | 0 | 0 | 0 | 0 | 1 |
| 20–end: | $\boldsymbol{0}_{2P}$ | $\boldsymbol{0}_{2P}$ | $\boldsymbol{0}_{2P}$ | $\boldsymbol{0}_{2P}$ | $\boldsymbol{0}_{2P}$ | $\boldsymbol{0}_{2P}$ | $\boldsymbol{0}_{2P}$ | $\boldsymbol{0}_{2P}$ | $\boldsymbol{0}_{2P}$ | $\boldsymbol{0}_{2P}$ | $\boldsymbol{0}_{2P}$ | $\boldsymbol{0}_{2P}$ | $\boldsymbol{0}_{2P}$ | $\boldsymbol{0}_{2P}$ | $\boldsymbol{0}_{2P}$ | $\boldsymbol{0}_{2P}$ |

#### D.6.4 RESIDUAL CONNECTION

Similar to the residual connection applied after the attention layer, the residual connection following the feed-forward layer "fills in the blanks" of the matrix $\boldsymbol{Y}^{(1)}$ with the output of each subnetwork as

$$\boldsymbol{X}^{(1)} = \boldsymbol{Y}^{(1)} + \mathtt{FF}_1(\boldsymbol{Y}^{(1)}).$$

The example of $\boldsymbol{X}^{(1)}$ is illustrated in Table 23.

Table 23: Example of after applying the residual connection, $\boldsymbol{X}^{(1)}$, continuing from Table 22.

| $\mathcal{I}$ | = | 0 | 0 | 0 | $\rightarrow$ | 7 | 5 | 0 | $\rightarrow$ | 5 | 0 | 1 | $\rightarrow$ | 1 | 0 | 2 |
|---|---|---|---|---|---|---|---|---|---|---|---|---|---|---|---|---|
| 1: NUM | 0 | 0 | 0 | 0 | 0 | 7 | 5 | 0 | 0 | 5 | 0 | 1 | 0 | 1 | 0 | 2 |
| 2: IS_BOS | 0 | 0 | 0 | 0 | 0 | 0 | 0 | 0 | 0 | 0 | 0 | 0 | 0 | 0 | 0 | 0 |
| 3: FULL_ONES | 1 | 1 | 1 | 1 | 1 | 1 | 1 | 1 | 1 | 1 | 1 | 1 | 1 | 1 | 1 | 1 |
| 4: PRE_SUM | 0 | 0 | 0 | 0 | 7 | 5 | 0 | 0 | 15 | 9 | 0 | 0 | 11 | 9 | 1 | 0 |
| 5: PRE_CARRY | 0 | 0 | 0 | 0 | 0 | 7 | 5 | 0 | 0 | 15 | 9 | 0 | 0 | 11 | 9 | 1 |
| 6: PRE_ARROW | 1/4 | 1/4 | 1/4 | 1 | 1/5 | 1/5 | 1/5 | 1 | 1/6 | 1/6 | 1/6 | 1 | 1/7 | 1/7 | 1/7 | 1 |
| 7: PRE_EOS | 1/2 | 1/3 | 1/3 | 1/3 | 1/3 | 1/3 | 1/3 | 1/3 | 1/3 | 1/3 | 1/3 | 1/3 | 1/2 | 1/2 | 1/2 | 1/2 |
| 8-17: SUM | $e_1^{10}$ | $e_1^{10}$ | $e_1^{10}$ | $e_1^{10}$ | $e_8^{10}$ | $e_6^{10}$ | $e_1^{10}$ | $e_1^{10}$ | $e_6^{10}$ | $e_1^{10}$ | $e_2^{10}$ | $e_1^{10}$ | $e_2^{10}$ | $e_1^{10}$ | $e_3^{10}$ | $e_1^{10}$ |
| 18: ARROW | 0 | 0 | 0 | 1 | 0 | 0 | 0 | 1 | 0 | 0 | 0 | 1 | 0 | 0 | 0 | 1 |
| 19: EOS | 0 | -1 | -1 | 0 | -1 | -1 | -1 | 0 | -1 | -1 | -1 | 0 | 0 | 0 | 0 | 1 |
| POS_1 | $v_4^{P_1}$ | $v_5^{P_1}$ | $v_6^{P_1}$ | $v_7^{P_1}$ | $v_4^{P_1}$ | $v_5^{P_1}$ | $v_6^{P_1}$ | $v_7^{P_1}$ | $v_4^{P_1}$ | $v_5^{P_1}$ | $v_6^{P_1}$ | $v_7^{P_1}$ | $v_4^{P_1}$ | $v_5^{P_1}$ | $v_6^{P_1}$ | $v_7^{P_1}$ |
| POS_1_NEXT | $v_5^{P_1}$ | $v_6^{P_1}$ | $v_7^{P_1}$ | $v_8^{P_1}$ | $v_5^{P_1}$ | $v_6^{P_1}$ | $v_7^{P_1}$ | $v_8^{P_1}$ | $v_5^{P_1}$ | $v_6^{P_1}$ | $v_7^{P_1}$ | $v_8^{P_1}$ | $v_5^{P_1}$ | $v_6^{P_1}$ | $v_7^{P_1}$ | $v_8^{P_1}$ |
| POS_2 | $v_2^{P_2}$ | $v_2^{P_2}$ | $v_2^{P_2}$ | $v_2^{P_2}$ | $v_3^{P_2}$ | $v_3^{P_2}$ | $v_3^{P_2}$ | $v_3^{P_2}$ | $v_4^{P_2}$ | $v_4^{P_2}$ | $v_4^{P_2}$ | $v_4^{P_2}$ | $v_5^{P_2}$ | $v_5^{P_2}$ | $v_5^{P_2}$ | $v_5^{P_2}$ |
| POS_2_NEXT | $v_3^{P_2}$ | $v_3^{P_2}$ | $v_3^{P_2}$ | $v_3^{P_2}$ | $v_4^{P_2}$ | $v_4^{P_2}$ | $v_4^{P_2}$ | $v_4^{P_2}$ | $v_5^{P_2}$ | $v_5^{P_2}$ | $v_5^{P_2}$ | $v_5^{P_2}$ | $v_6^{P_2}$ | $v_6^{P_2}$ | $v_6^{P_2}$ | $v_6^{P_2}$ |

### D.7 DECODING FUNCTION

The final step is decoding: the model decides which token to predict as the next token based on the embedding matrix. Specifically, with a weight matrix $\boldsymbol{W}_{\text{out}} \in \mathbb{R}^{|\mathcal{V}| \times d}$, the model first compute the multiplication between $\boldsymbol{W}_{\text{out}} \in \mathbb{R}^{|\mathcal{V}| \times d}$ and $\boldsymbol{X}^{(1)}$. Then, the model takes a (token-wise) arg-max operation for greedy decoding. Mathematically, the next-prediction at $i$-th token $\sigma_i$ can be written as follows:

$$k_i := \arg\max_{k \in [|\mathcal{V}|]} \left\{ o_k : \boldsymbol{W}_{\text{out}} \boldsymbol{X}^{(1)}_{\bullet i} = \begin{bmatrix} o_1 & \cdots & o_{|\mathcal{V}|} \end{bmatrix}^\top \right\}. \tag{24}$$

The design of the weight matrix $\boldsymbol{W}_{\text{out}} \in \mathbb{R}^{|\mathcal{V}| \times d}$ is illustrated in Table 24, and the example of the matrix $\boldsymbol{W}_{\text{out}} \boldsymbol{X}^{(1)}_1$ and the output sequence is presented in Tables 25 and 26, respectively.

Table 24: The transposed weight matrix $\boldsymbol{W}_{\text{out}}^\top$ of the linear readout in decoding function.

| $\mathcal{V}$ | 0 | 1 | 2 | 3 | 4 | 5 | 6 | 7 | 8 | 9 | + | = | $\rightarrow$ | \$ |
|---|---|---|---|---|---|---|---|---|---|---|---|---|---|---|
| 1–7: NUM-PRE_EOS | $\mathbf{0}_7$ | $\mathbf{0}_7$ | $\mathbf{0}_7$ | $\mathbf{0}_7$ | $\mathbf{0}_7$ | $\mathbf{0}_7$ | $\mathbf{0}_7$ | $\mathbf{0}_7$ | $\mathbf{0}_7$ | $\mathbf{0}_7$ | $\mathbf{0}_7$ | $\mathbf{0}_7$ | $\mathbf{0}_7$ | $\mathbf{0}_7$ |
| 8: SUM$_1$ | 1 | 0 | 0 | 0 | 0 | 0 | 0 | 0 | 0 | 0 | 0 | 0 | 0 | 0 |
| 9: SUM$_2$ | 0 | 1 | 0 | 0 | 0 | 0 | 0 | 0 | 0 | 0 | 0 | 0 | 0 | 0 |
| 10: SUM$_3$ | 0 | 0 | 1 | 0 | 0 | 0 | 0 | 0 | 0 | 0 | 0 | 0 | 0 | 0 |
| 11: SUM$_4$ | 0 | 0 | 0 | 1 | 0 | 0 | 0 | 0 | 0 | 0 | 0 | 0 | 0 | 0 |
| 12: SUM$_5$ | 0 | 0 | 0 | 0 | 1 | 0 | 0 | 0 | 0 | 0 | 0 | 0 | 0 | 0 |
| 13: SUM$_6$ | 0 | 0 | 0 | 0 | 0 | 1 | 0 | 0 | 0 | 0 | 0 | 0 | 0 | 0 |
| 14: SUM$_7$ | 0 | 0 | 0 | 0 | 0 | 0 | 1 | 0 | 0 | 0 | 0 | 0 | 0 | 0 |
| 15: SUM$_8$ | 0 | 0 | 0 | 0 | 0 | 0 | 0 | 1 | 0 | 0 | 0 | 0 | 0 | 0 |
| 16: SUM$_9$ | 0 | 0 | 0 | 0 | 0 | 0 | 0 | 0 | 1 | 0 | 0 | 0 | 0 | 0 |
| 17: SUM$_{10}$ | 0 | 0 | 0 | 0 | 0 | 0 | 0 | 0 | 0 | 1 | 0 | 0 | 0 | 0 |
| 18: ARROW | 0 | 0 | 0 | 0 | 0 | 0 | 0 | 0 | 0 | 0 | 0 | 0 | 10 | 0 |
| 19: EOS | 0 | 0 | 0 | 0 | 0 | 0 | 0 | 0 | 0 | 0 | 0 | 0 | 0 | 100 |
| 20–end: POSITIONS | $\mathbf{0}_{2P}$ | $\mathbf{0}_{2P}$ | $\mathbf{0}_{2P}$ | $\mathbf{0}_{2P}$ | $\mathbf{0}_{2P}$ | $\mathbf{0}_{2P}$ | $\mathbf{0}_{2P}$ | $\mathbf{0}_{2P}$ | $\mathbf{0}_{2P}$ | $\mathbf{0}_{2P}$ | $\mathbf{0}_{2P}$ | $\mathbf{0}_{2P}$ | $\mathbf{0}_{2P}$ | $\mathbf{0}_{2P}$ |

Table 25: Example output of linear readout, $\boldsymbol{W}_{\text{out}}\boldsymbol{X}^{(1)}$, continuing from Tables 23 and 24. The yellow cells represent the maximum value of each column.

| $\mathcal{I}$ | = | 0 | 0 | 0 | → | 7 | 5 | 0 | → | 5 | 0 | 1 | → | 1 | 0 | 2 |
|---|---|---|---|---|---|---|---|---|---|---|---|---|---|---|---|---|
| 0 | 1 | 1 | 1 | 1 | 0 | 0 | 1 | 1 | 0 | 1 | 0 | 1 | 0 | 1 | 0 | 1 |
| 1 | 0 | 0 | 0 | 0 | 0 | 0 | 0 | 0 | 0 | 0 | 1 | 0 | 1 | 0 | 0 | 0 |
| 2 | 0 | 0 | 0 | 0 | 0 | 0 | 0 | 0 | 0 | 0 | 0 | 0 | 0 | 0 | 1 | 0 |
| 3 | 0 | 0 | 0 | 0 | 0 | 0 | 0 | 0 | 0 | 0 | 0 | 0 | 0 | 0 | 0 | 0 |
| 4 | 0 | 0 | 0 | 0 | 0 | 0 | 0 | 0 | 0 | 0 | 0 | 0 | 0 | 0 | 0 | 0 |
| 5 | 0 | 0 | 0 | 0 | 0 | 1 | 0 | 0 | 1 | 0 | 0 | 0 | 0 | 0 | 0 | 0 |
| 6 | 0 | 0 | 0 | 0 | 0 | 0 | 0 | 0 | 0 | 0 | 0 | 0 | 0 | 0 | 0 | 0 |
| 7 | 0 | 0 | 0 | 0 | 1 | 0 | 0 | 0 | 0 | 0 | 0 | 0 | 0 | 0 | 0 | 0 |
| 8 | 0 | 0 | 0 | 0 | 0 | 0 | 0 | 0 | 0 | 0 | 0 | 0 | 0 | 0 | 0 | 0 |
| 9 | 0 | 0 | 0 | 0 | 0 | 0 | 0 | 0 | 0 | 0 | 0 | 0 | 0 | 0 | 0 | 0 |
| + | 0 | 0 | 0 | 0 | 0 | 0 | 0 | 0 | 0 | 0 | 0 | 0 | 0 | 0 | 0 | 0 |
| = | 0 | 0 | 0 | 0 | 0 | 0 | 0 | 0 | 0 | 0 | 0 | 0 | 0 | 0 | 0 | 0 |
| → | 0 | 0 | 0 | 10 | 0 | 0 | 0 | 10 | 0 | 0 | 0 | 10 | 0 | 0 | 0 | 10 |
| $ | 0 | -100 | -100 | 0 | -100 | -100 | -100 | 0 | -100 | -100 | -100 | 0 | 0 | 0 | 0 | 100 |

Table 26: Example output sequence $\mathcal{O}$, continuing from Table 25.

| $\mathcal{I}$ | = | 0 | 0 | 0 | → | 7 | 5 | 0 | → | 5 | 0 | 1 | → | 1 | 0 | 2 |
|---|---|---|---|---|---|---|---|---|---|---|---|---|---|---|---|---|
| $\mathcal{O}$ | 0 | 0 | 0 | → | 7 | 5 | 0 | → | 5 | 0 | 1 | → | 1 | 0 | 2 | $ |

# E MORE ATTENTION PATTERNS OF TRAINED TRANSFORMERS

Continuing from the discussion in Section 4.4 on the attention patterns due to the (non-)existence of scratchpad, we showcase more examples of the attention matrices $\texttt{softmax}(QK^\top + \Lambda)$ of actually trained Transformers (where $\Lambda$ is a causal mask).

## E.1 ATTENTION PATTERNS **WITH** SCRATCHPAD

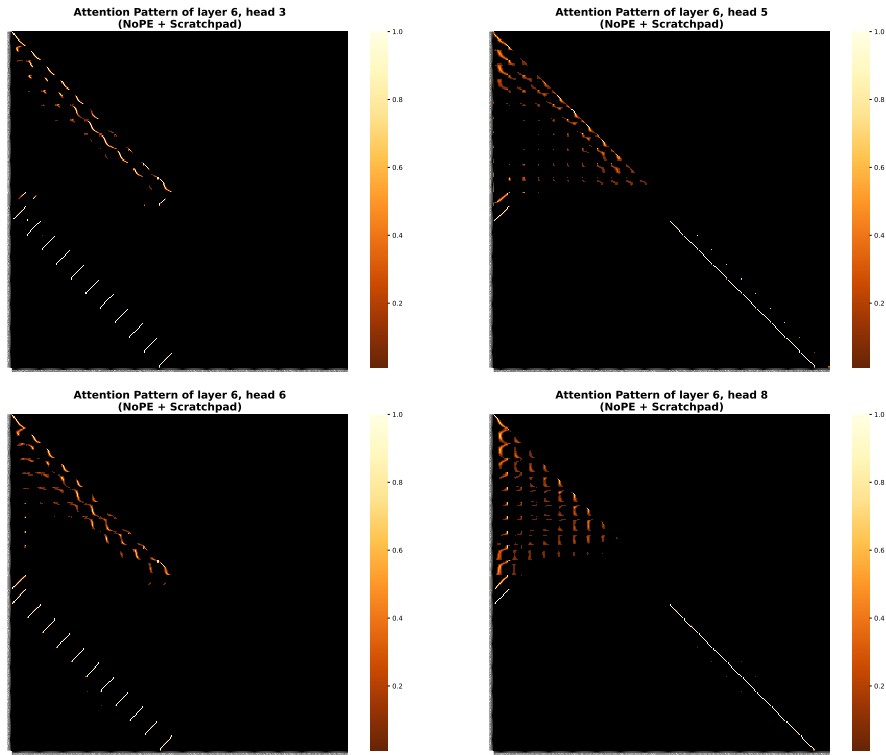

Figure 15: Some attention patterns of 6-layer 8-head decoder-only Transformers trained with NoPE and using the scratchpad.

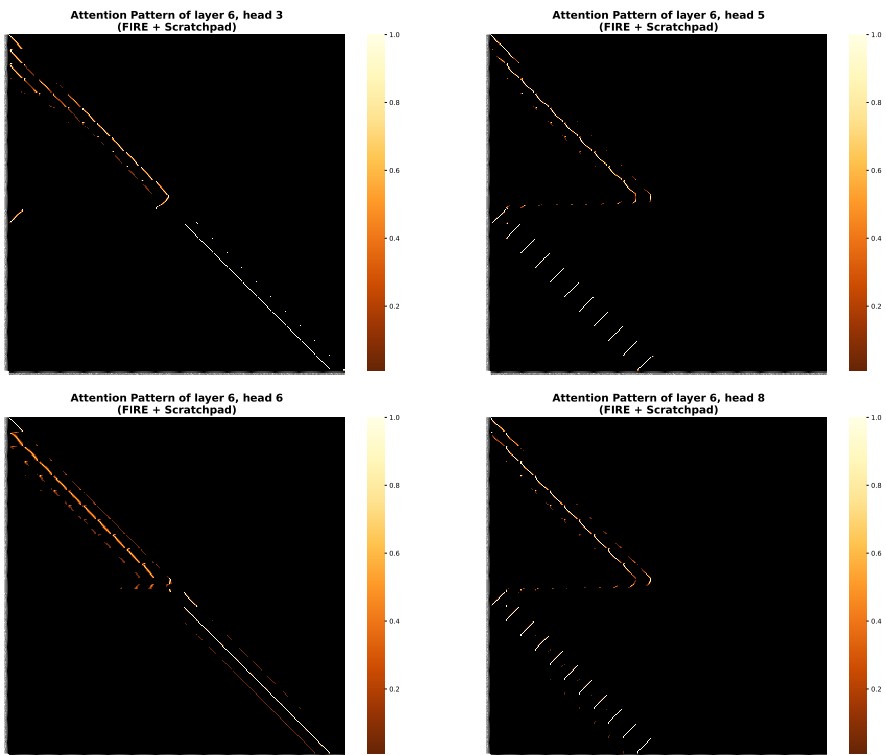

Figure 16: Some attention patterns of 6-layer 8-head decoder-only Transformers trained with FIRE and using the scratchpad.

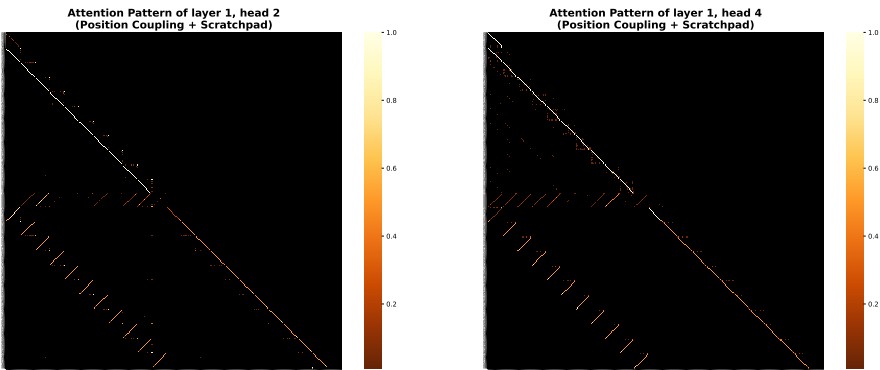

Figure 17: Some attention patterns of 6-layer 8-head decoder-only Transformers trained with bi-level Position Coupling and using the scratchpad.

## E.2 ATTENTION PATTERNS **WITHOUT** SCRATCHPAD

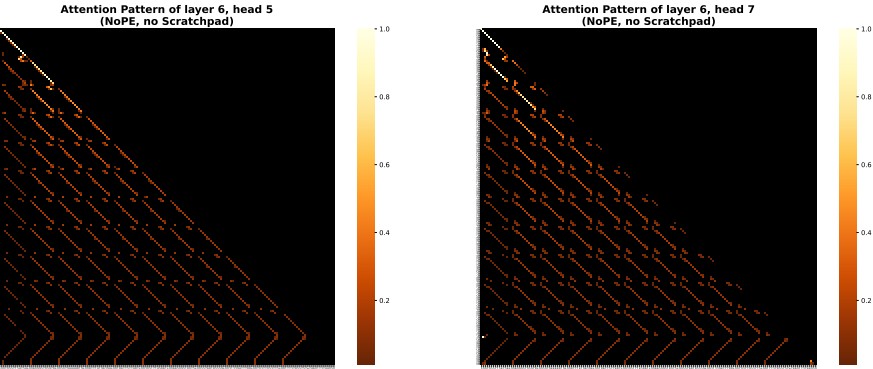

Figure 18: Some attention patterns of 6-layer 8-head decoder-only Transformers trained with NoPE but not using the scratchpad.

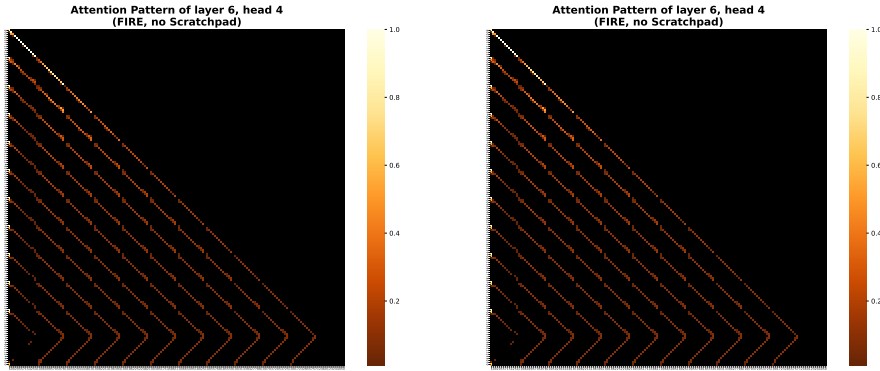

Figure 19: Some attention patterns of 6-layer 8-head decoder-only Transformers trained with FIRE but not using the scratchpad.

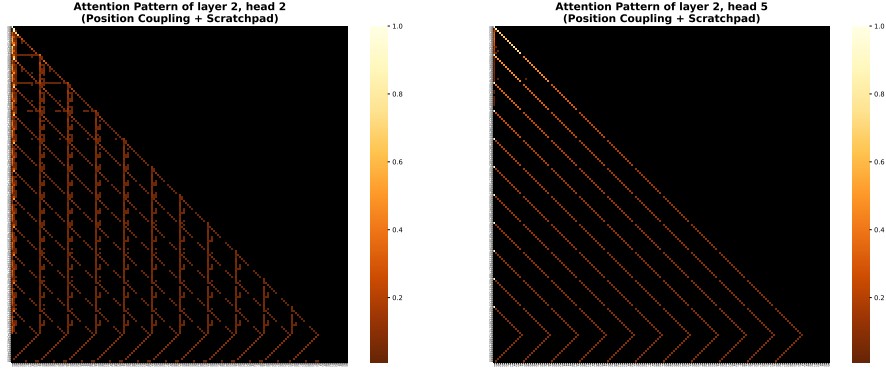

Figure 20: Some attention patterns of 6-layer 8-head decoder-only Transformers trained with (single-level) Position Coupling but not using the scratchpad.

