# OpenReview forum: "Arithmetic Transformers Can Length-Generalize in Both Operand Length and Count"
_ICLR.cc/2025/Conference — ICLR 2025 Poster_

### Official Review · Reviewer_3hhr · 2024-10-30

**Soundness:** 3
**Presentation:** 4
**Contribution:** 2
**Rating:** 5
**Confidence:** 4

**Summary:**

In this paper, arithmetic transformers refer to transformers trained on arithmetic tasks.  The core idea of the paper is to combine position coupling and scratchpad, and the authors demonstrate that this allows for length extrapolation in arithmetic tasks.

**Strengths:**

They show that by using a specific position embedding encoding (tri-level position coupling), they are able to also perform well on multiplication. This is an interesting observation for multiplication that when baked into the ‘structure’ of the transformer can give generalisation on arithmetic.

**Weaknesses:**

My first slightly minor nitpick is that the scratchpad requires scratchpad data generated in that particular format, for the multiplication specific task. In a general pretraining or fine-tuning setting, this could probably be added in as additional data to shape the model’s behaviour.
However, the specificity of position coupling (for all variants of position coupling) means that it can’t be a general modification to standard Transformers in a pretraining setting, and have it improve arithmetic capabilities. To their credit, the authors have already addressed this in the Limitations section of the paper.

**Questions:**

- Is there a way for the position coupling to be done “automatically” that you know of? I am unfamiliar with this specific line of work.
- For PARITY, at least, I have seen a prompt construction that asks the model to number the bit that the model has generated as it generates the internal state (odd or even) given the prefix.
  For example, given the sequence 1101.
  ```
    1. 1 odd
    2. 1 even
    3. 0 even
    4. 1 odd

    Final answer: odd.
  ````
  The intuition here being that the numbering allows the attention mechanism to more accurately pin-point the required informaton, which has a similar flavour to your approach.
  I was wondering if you have attempted to prompt a model so that it generates the same positioning scheme (just in human readable form), and I wonder if that will improve multiplication generalisation.

**Details Of Ethics Concerns:**

No concerns

---

> ### Author Response · Authors · 2024-11-21
>
> We thank the reviewer for the insightful comments. Before we address your comments and questions, please allow us to again clarify our main contributions here.
>
> - Novelty in problem settings: Our paper is the first work that successfully achieves length generalizations on multi-operand integer addition and general integer multiplication. While 2-operand integer addition and multiplication with a fixed length of multiplier (or multiplicand) have been widely studied, our tasks are more general and combinatorially complex, subsuming the previously studied tasks as well.
> - Novelty in the method: To achieve a significant length generalization in these tasks, we show that our proposed combination of scratchpad & multi-level Position Coupling is crucial and effective. Neither the scratchpad nor Position Coupling alone can achieve a nice length generalization for our tasks.
>
> We would also like to note that, perhaps surprisingly, these simple-looking arithmetic tasks remain as a hurdle even for recent state-of-the-art LLMs. Even recent versions of GPT struggle with simple integer multiplication: https://x.com/SmokeAwayyy/status/1836157777628250583.
>
> Below, we provide our response to the weaknesses and questions raised by the reviewer.
>
> **W1. The specificity of position coupling (for all variants of position coupling) means that it can’t be a general modification to standard Transformers in a pretraining setting.**
>
> - Thank you for your insightful comment. We agree that our current approach is not well-suited for the pretraining setting in general LLMs, as Position Coupling is only applied to input sequences of specific formats. However, we would like to note that the goal of this study is to develop a method that enables models trained from scratch to achieve strong length generalization for arithmetic problems that are considered difficult in the literature. To achieve this goal, we focus on designing a method that provides a strong inductive bias to the model, with less emphasis on its universality or general applicability.
> - Additionally, a recent work on length generalization [1] proposes a framework to train additional attention heads in LLMs tailored for arithmetics, aiming to integrate their approach into general models. We believe that our approach could be compatible with this framework. Given that even state-of-the-art LLMs show surprisingly bad performance in simple arithmetic tasks, implanting our method as a subroutine in LLMs can potentially improve their arithmetic capabilities and may provide valuable insights for broader applications.
>
> **Q1. Is there a way for the position coupling to be done “automatically” that you know of?**
>
> - As you pointed out, the next promising step for this work is extending its application to the case where the task structure is implicit or even entirely known, which would require an “automatic” position coupling. While we have been considering potential methods to address this challenge, we are not currently aware of any prior or related work in this area.
>
> **Q2. I am wondering if you have attempted to prompt a model so that it generates the same positioning scheme.**
>
> - Thank you for your thoughtful question. The approach you describe, involving prompting a model to generate its own positioning scheme, is indeed an interesting direction. However, we would like to note that this question lies outside the scope of our work, as our study focuses on training models from scratch rather than prompting on a trained LLM.
> - (Continue in the subsequent comment)

---

> ### Author Response · Authors · 2024-11-21
>
> - Having said that, we tried prompting ChatGPT (based on GPT-4o) to see if LLMs can few-shot learn how to length generalize in integer addition. The results are quite interesting, but it still leaves something to be desired; we will discuss the takeaway after presenting the results.
> - For length generalization in terms of operand length, this is how our conversation went:
>
>         (our prompt) 23420630423568239234+66423939693030435634=
>
>         (ChatGPT) The sum of 23420630423568239234 and 66423939693030435634 is: 23420630423568239234 + 66423939693030435634 = 89844570116698674868.
>             (Comment by authors: This is wrong because the correct answer is 89844570116598674868.)
>
>         (our prompt) You got it wrong. Let me give you some hint on how to do integer addition through an example.
>
>         When you need to two long integers, you can do this step by step, where at step j you calculate the sum of j-th digits and the carry from the previous step j-1. For example:
>
>         12395+69405
>         1. 0+5+5=10. sum: 0, carry: 1
>         2. 1+9+0=10. sum: 0, carry: 1
>         3. 1+3+4=8. sum: 8, carry: 0
>         4. 0+2+9=11. sum: 1, carry: 1
>         5. 1+1+6=8. sum:8, carry: 0
>         Hence, the final answer is: 12395+69405=81800.
>
>         Now, using this approach, please try to solve the following problem in a step-by-step manner: 23420630423568239234+66423939693030435634=
>
>         (ChatGPT) Using the step-by-step approach: 23420630423568239234+66423939693030435634=89844570116598674868 This matches the process correctly. Thank you for the clarification!
>             (Comment by authors: Instead of creating scratchpads, ChatGPT analyzed our instructions and generated a python code that solves the problem. Impressive!)
>
> - We next tried length generalization in terms of operand count while fixing the maximum operand length to 10. This is how our conversation went:
>
>         (our prompt) 1534915666+5345912534+1125242414+5461276086+236362345+126+73655129+7093058144+69819285+652=
>
>         (ChatGPT) The result of the sum 1534915666+5345912534+1125242414+5461276086+236362345+126+73655129+7093058144+69819285+652 is: 21,530,244,381
>             (Comment by authors: This is wrong, the correct answer is 20,940,242,381.)
>
>         (our prompt) You got it wrong. Let me give you some hint on how to do multi-operand addition through an example.
>
>         When you need to add N numbers, you can do this step by step, where at step j you calculate the sum of the first j numbers by adding the sum of the first j-1 numbers you calculated in the previous step and the j-th operand. For example:
>
>         12351+69405+79595+126+98471
>         1. 0+12351=12351
>         2. 12351+69405=81756
>         3. 81756+79595=161351
>         4. 161351+126=161477
>         5. 161477+98471=259948
>         Hence, the final answer is: 12351+69405+79595+126+98471=259948.
>
>         Now, using this approach, please try to solve the following problem in a step-by-step manner: 1534915666+5345912534+1125242414+5461276086+236362345+126+73655129+7093058144+69819285+652=
>
>         (ChatGPT) Let's solve 1534915666 + 5345912534 + 1125242414 + 5461276086 + 236362345 + 126 + 73655129 + 7093058144 + 69819285 + 652 step by step, using the method you provided:
>
>         Step 1: 0+1534915666=1534915666
>         Step 2: 1534915666+5345912534=6880828200
>         Step 3: 6880828200+1125242414=8006070614
>         Step 4: 8006070614+5461276086=13467346700
>         Step 5: 13467346700+236362345=13703709045
>         Step 6: 13703709045+126=13703709171
>         Step 7: 13703709171+73655129=13777364200
>         Step 8: 13777364200+7093058144=20870422344
>         Step 9: 20870422344+69819285=20940241629
>         Step 10: 20940241629+652=20940242281
>
>         Final Answer:
>         1534915666 + 5345912534 + 1125242414 + 5461276086 + 236362345 + 126 + 73655129 + 7093058144 + 69819285 + 652 = 20,940,242,281
>             (Comment by authors: The result was more accurate, but still off by one digit. It made a mistake in the hundreds digit of Step 7.)
>
> - In summary, our preliminary experiments suggest that prompting holds promise as an effective approach to teach LLMs to perform better on arithmetic tasks. However, to fully realize the potential of prompting, it is essential to first improve the model's baseline performance (i.e., without sophisticated prompting) on fundamental problems such as 10-digit, 2-operand addition. We believe our proposed method can make a meaningful contribution in this regard by enhancing LLMs’ arithmetic capabilities with standard, unprompted inputs.

---

> ### Author Response · Authors · 2024-11-21
>
> We hope our responses have sufficiently addressed your concerns. If you have any remaining questions or comments, we would be grateful to hear them. Your initial assessment was slightly negative (weak reject), and we are curious if there are any remaining reservations preventing you from recommending acceptance. If so, we would be happy to address them further. Otherwise, we kindly ask you to reconsider your evaluation of our work.
>
> ---
>
> **References**
>
> [1] Sabbaghi et al., Explicitly Encoding Structural Symmetry is Key to Length Generalization in Arithmetic Tasks. arXiv preprint. 2024.

---

> ### Author Response · Authors · 2024-11-28
>
> Dear reviewer 3hhr
>
> Thank you for your important questions and insightful comments in the first review. We uploaded our developed manuscript that improves clarity and includes additional ablation studies. As the discussion period is almost over, please check out our revised paper and our response if you haven’t done so. We hope that our response resolves your initial concerns. If you feel the same, we would appreciate if this could be reflected through an updated score. Also, please do not hesitate to let us know if you have any remaining questions. We appreciate your contribution to the reviewing process.
>
>
> Sincerely,
>
> Authors

---

### Official Review · Reviewer_uejG · 2024-11-02

**Soundness:** 3
**Presentation:** 3
**Contribution:** 3
**Rating:** 8
**Confidence:** 4

**Summary:**

The paper describes methods allowing decoder-only transformers to generalize to longer operands when performing integer addition or multiplication, and to generalize to a larger number of operands when performing addition.

The authors mix two existing techniques, scratchpad, which forces the model to output a sequence of intermediary steps before producing the result, and position coupling, problem-specific hierarchical positional encodings. They experiment with three tasks.

For parity calculation over bit strings, they use a scratchpad which computes parity integration, ie force the transformer to compute the parities of successive prefixes, and use the same (absolute) positional encoding in the input and output sequences.

For addition of $n$ positive integers, they use a scratch pad of partial sums (0, first operand, sum of the first two...) and two positional encodings: one for the position of digits in each operand and output, and one for the position of each operand and output in the operation and scratchpad.

For multiplication of 2 position integers, they use a scratchpad of single digit products and partial sums, and three positional encodings, which reflect the different subtasks in multi-digit multiplication.

Experimental results indicate that such techniques allow for generalization to longer lengths on all three tasks, and a larger number of operands for addition.

**Strengths:**

The paper studies an important limitation of arithmetic transformers, their inability to scale to longer sequences that those seen at training. The authors propose a solution to this problem.

The paper is clearly written, the methods is clearly described and the experimental results are compelling. Some ablation results are provided.

**Weaknesses:**

The authors seem to only focus on positive integers. This greatly reduces the practical interest of the findings, since integer arithmetic usually involve positive and negative numbers. Could the results be extended to relative numbers, either by adding a sign in the number representation, or changing the tokenization scheme to a number system that encode negative numbers as well (e.g. encoding them in base -10 instead of base 10, or using balanced base 10, with digits -4, -3, -2, -1, 0, 1, 2, 3, 4, 5)? Since the only change is the tokenizer, this could work out of the box. (besides, arithmetic might be easier to learn in balanced base, because the addition and multiplication tables feature less carries).

The paper makes heavy use of techniques introduced in prior work, this is not a problem, but it would be fitting to have a proper "related work" section in the main, describing these methods and acknowledging anteriority. At present, these descriptions are scattered in sections 1 and 2. This could be done at the expense of the preliminaries section (sec. 2, which serves little purpose), and perhaps some of the parity results, which are underwhelming, compared to the rest of the experiments.

Both the scratchpad structure and the position encodings proposed in this paper are strongly dependent on the task. This suggests that the model can be trained to perform one operation (addition OR multiplication), but not calculations featuring both. This is a strong limitation. Have you experimented with such tasks (add and multiply)? Can they be learned zero or few shots? A discussion of these questions would be useful.

**Questions:**

* At test time, for a model prediction to be correct, you insist on both the output and the scratchpad to be correctly predicted. Prior work suggests that models sometimes learn despite incorrect scratchpads. Have you tested models by judging them on the final result only? (this could have an impact ablation results, and comparisons with other approaches)
* Is reversing the order of digits in the scratchpad and output sequence really necessary? Can you provide ablations for this?
* What would be the impact of reversing the digits in the input sequence?
* Your architecture ablations suggest that shallow models can learn and length generalize. Unfortunately, they all use a large embedding dimension (512), which translates into a large number of parameters. Have you tried models with smaller dimensions (128, 64 even)?
* In figure 10, you report better performance of 2 layer models than 4 or 6 layer models with 2 and 4 heads, and better performance of 4 layer models than 6 layer models with 8 heads. Could this be due to the size of your training sample (500k may be on the low side for the larger architectures), or the number of steps you allow?
* An ablation on the training set size would be useful.
* For addition, what is the point of adding 0 and a copy of the first operand in the scratchpad?
* The main text claims the model has 63M parameters, the appendix claims 25M, which is right?
* Would this approach benefit from an encoder-decoder model? (intuitively, bidirectional attention over the input sequence should help, all the more as the digits in input sequences are not reversed)

---

> ### Author Response · Authors · 2024-11-21
>
> We deeply appreciate the reviewer's valuable questions and rich comments, and we hope our response relevantly addresses all points raised in the review.
>
> **W1. Could the proposed method be extended to handle negative integers by modifying the tokenization scheme or adopting alternative number systems?**
>
> - We believe that as long as the fundamental structure of the operations remains unchanged—for example, for addition problems, where digits in the same position are added and carries are propagated to the next position—the approach could be applicable across various number systems. Therefore, extending the method to handle signed numbers or even a balanced number system seems feasible.
> - Additionally, we note that most prior works addressing length generalization for addition problems have focused exclusively on positive integers and do not consider subtraction or negative numbers. While extending to negative numbers is an interesting future direction, our work follows the standard approach in the current literature.
>
> **W2. Introduce the techniques in the “related work” section.**
>
> - Thanks for the suggestion. We agree that presenting the explanation of the main techniques in a dedicated "Related Work" section (Section 2.1) would enhance the clarity of the paper. We have revised the paper accordingly and would appreciate it if you could review the updated version.
>
> **W3. This suggests the model can be trained to perform one operation (addition OR multiplication), but not calculations featuring both.**
>
> - Thank you for the insightful comment. We would like to note that, to the best of our knowledge, no prior work has explored the length generalization capabilities of models designed to handle multiple operations, such as addition and multiplication, within a single model.
> - We agree that the reviewer’s idea is interesting. However, due to limited computational resources, we were not able to conduct extra experiments on this idea. We will make an effort to run this experiment and report the results in a future update.
>
> **Q1. Have you tested models by judging them on the final result only?**
>
> - Originally, we counted a prediction as correct only if both the output and the scratchpad are correctly predicted. Following the reviewer’s suggestion, we re-evaluated the models by judging the final result alone, ignoring the scratchpad. The results are shown in Figure 15 of the revised paper, which is a reassessment of Figure 14 (about input format ablation).
> - This evaluation revealed a general increase in the accuracy numbers across various experimental setups, but the overall comparison between approaches remains unchanged.
>
> **Q2. Is reversing the order of digits in the scratchpad and output sequence really necessary?**
>
> - Reversing the output sequence has been widely adopted in the literature examining length generalization for arithmetic tasks in decoder-only Transformers [1-3,5-6]. This approach makes the target function simpler, as illustrated by the following example: consider the addition problem 9999+3=10002. Without the reversed output format, the model must attend to every digit token in the query to predict 1, as the carry propagation starts from the least significant digit. In contrast, with the reversed output format, the model now predicts digits starting from the least significant digit, reducing the need to detect long carry chains.
> - We provide the experimental results of employing the plain output format in Figure 14 of the revised paper (ZeroPad+NoReverse). We see that there is a significant degradation of length generalization in terms of **operand length**, which is consistent with existing observations in the literature. However, an intriguing observation is that the model still is capable of length generalization in terms of **operand count.** We believe that this highlights the power of scratchpad, in the following sense. In existing works studying 2-operand addition, it is observed that even without output reversing, the model is capable of in-distribution generalization (i.e., if the model is trained on 10-digit numbers, it can generalize in 10-digit addition). For our case, the scratchpad reduces the multi-operand addition problem into a sequence of 2-operand addition problems. Hence, when the model is trained with scratchpad and without output reversing, the model can learn to length-generalize in operand count even though it does not extrapolate in terms of operand length.

---

> ### Author Response · Authors · 2024-11-21
>
> **Q3. What would be the impact of reversing the digits in the input sequence?**
>
> - We present the performance with the reversed query in Figure 14 of the revised paper (ZeroPad+ReverseAll). With reversed input (as well as the response), we do observe some degradation, especially in terms of the number of operands. Notably, zero-padded and reversed input sequences show a significant failure rate when the number of operands is exactly 20! We did not have a chance to dive deeper into the underlying reasons yet, but we observe that the model fails to predict the EOS token when it has to.
> - Although we observe some performance degradation, we still believe that there is potential for improvement, as the hyperparameters have not been tuned for the reversed query setup. Moreover, according to our theory which addresses the expressive power/capacity of Transformers, the construction remains valid for the reversed query format (provided that the position IDs are properly assigned). Also, recent literature offers no consensus on whether reversing the query helps the length generalization, as both plain [3,4] and reversed [5,6] query formats have been adopted in prior studies.
>
> **Q4. Have you tried models with smaller dimensions (128, 64 even)?**
>
> - First, we would like to clarify that the embedding dimension used in Fig 10 is 1024, not 512 (Refer to Tables 2 and 3).
> - To address your question, we trained 1L4H models, each with embedding dimensions of 64, 128, 256, and 512 (with dimensions per head of 16, 32, 64, and 128, respectively). We observe that there is a significant performance gap between these configurations. While models with small embedding dimensions have sufficient expressive capacity for solving the task, we believe that large embedding dimensions are crucial for effective optimization. The results can be found in Figure 12 of our revised paper.
>
> **Q5. Why (2 layers > 4, 8 layers for 2, 4 heads models) and (4 layers > 8 layers for 8 heads models)?**
>
> - Currently, we do not have a clear explanation for the inferior performance of larger models, as there could be various factors. These include the training set size, total number of iterations, learning rate, randomness (e.g., initialization, dataset, training process), and different implicit biases. In light of our Theorem 4.1 which shows that even a 1-layer 4-head model is expressive enough to perfectly solve the multi-operand addition task, we hypothesize that using a too large model may implicitly bias the model toward some “shortcut solutions” that can achieve in-distribution generalization but hurt out-of-distribution performance.
> - However, as we discuss in the next question, we observe that controlling the training set size does not significantly impact the model’s performance. Our next step is to investigate the effect of randomness, and we plan to conduct experiments with additional seeds once our ongoing experiments are completed.
>
> **Q6. Ablation on the training set size.**
>
> - To evaluate the effect of the training set size, we trained 6L8H models on training data sizes of 20K, 100K, 500K (our initial setting), 2M, and 10M, each with 8 seeds. The results are illustrated in Figure 13 of the revised version. We observe that the training set size does not significantly impact the model's performance.
>
> **Q7. For addition, what is the point of adding 0 and a copy of the first operand in the scratchpad?**
>
> - Including 0s and a copy of the first operand in the scratchpad is primarily a design choice that aligns with our theoretical framework (Theorem 4.1) that employs this format. Our theoretical construction becomes much simpler with this design choice, and we decided to maintain consistency between our theory and experiments. Other than that, it does not contain any significant meaning.
> - Furthermore, in earlier experiments, we tested the model without placeholder 0’s in the scratchpad and observed similar performance. We expect that removing the copy of the first operand would yield similar results as well.

---

> ### Author Response · Authors · 2024-11-21
>
> **Q8. The main text claims the model has 63M parameters, but the appendix claims 25M, which one is correct?**
>
> - Thank you for your careful reading. Both are correct because we used different embedding dimensions depending on the task.
> - For the parity task, we used $d=512$, which results in a 6-layer 8-head model with 25M parameters. This is correctly written in our Table 1.
> - For the other tasks like the multi-operand addition and multiplication, we used $d=1024$, resulting in a 6-layer 8-head model with 63M parameters. This is exactly written in Tables 2 and 3.
>
> **Q9. Would this approach benefit from an encoder-decoder model?**
>
> - As the reviewer suggested, we also believe that the encoder-decoder model could effectively learn the function and achieve length generalization without requiring the response to be reversed, as the bidirectional attention allows the model to leverage more information compared to a decoder-only model. However, as the scope of our work focuses on decoder-only models, we hope you understand that we do not provide experimental results to verify this intuition.
>
> ---
>
> **References**
>
> [1] Lee, Nayoung, et al. "Teaching arithmetic to small transformers." ICLR 2024.
>
> [2] Zhou, Hattie, et al. "What algorithms can transformers learn? a study in length generalization." ICLR 2024.
>
> [3] Cho, Hanseul, et al. "Position Coupling: Improving Length Generalization of Arithmetic Transformers Using Task Structure." NeurIPS 2024.
>
> [4] Fan, Ying, et al. "Looped Transformers for Length Generalization." *arXiv preprint arXiv:2409.15647* (2024).
>
> [5] Zhou, Yongchao, et al. "Transformers can achieve length generalization but not robustly." ICLR 2024 Workshop on Understanding of Foundation Models (ME-FoMo).
>
> [6] McLeish, Sean, et al. "Transformers Can Do Arithmetic with the Right Embeddings." NeurIPS 2024.

---

> > ### Comment · Reviewer_uejG · 2024-11-25
> >
> > Thank you very much for your replies, that clarified my understanding of the paper. I have increased my score.

---

> > > ### Author Response · Authors · 2024-11-25
> > > **Thank you very much**
> > >
> > > Thank you very much for your response and a positive reassessment of our work! If you come up with more questions or comments, please do not hesitate to leave them; we will happily answer them.

---

### Official Review · Reviewer_ois9 · 2024-11-04

**Soundness:** 3
**Presentation:** 3
**Contribution:** 2
**Rating:** 5
**Confidence:** 5

**Summary:**

The paper extends the idea of position coupling to get better generalization for the tasks of addition and multiplication. Position coupling is one of the several methods for leveraging the symmetries of a task, and in this paper, it is used in tandem with scratch-padding, padding the integers, and reversing. The paper instantiates generalization for up to 30 digits for addition with the mentioned techniques, and around 20 digits for multiplication, which is impressive.

**Strengths:**

1- The paper is well written, with their ideas well articulated in all the sections. By comparing their methods to some of the previous work they have made the reader convinced that their methods are necessary in the given framework (leveraging the structure grounded on APE) to achieve length generalization in arithmetic.

2- The idea of position coupling is applied to parity to achieve perfect length generalization.

3- Tri-level position coupling facilitates length generalization on both the first and second operands in multiplication.

**Weaknesses:**

1- In line 18, the paper claims to offer the first length generalization results for arithmetic tasks. To my knowledge, there are several other works that have argued the same. For instance, [1] has leveraged the same symmetries of addition (In a different framework, and using RPE)  to offer generalization up to 50 digits (for both operands) when trained on samples up to 5 digits. Can you explain the advantages of your work? Authors should have done a better comparison with previous work and stayed truthful about the results.

2- Scratch-padding has been explored before for the problem of parity as also mentioned in line 179. I'm not convinced that position coupling is the core solution for this problem, since it's been only compared with a NoPE  + scratchpad approach. To me it is clear why such an approach cannot work in a next-word-prediction setting as the generated string in the output relies on the order of the the bits in the input, and  scratch-padding makes the final answer reliant on the previous bits in the output. Therefore, even though the final answer of parity is intrinsically independent of the string order, with this solution it is in fact regarding the order. Position coupling had to be compared with other methods that utilizes positional encoding as well.

3- The contribution of the paper is pretty limited considering that position coupling is introduced in previous work, and bi-level and tri-level versions are two extensions of that method. Besides, the effectiveness of padding and reversing is already well-known and widely studied.

4- The scope of the experiments are limited to solely having numbers and in the absence of any text. How does this apply to a broader range when numbers are amongst text? For instance, [1] has tried to address this with training some additional attention heads tailored for arithmetic along with rest of the architecture. Besides, is it true that with your method multiplication and addition require different sets of positional encoding? Please correct me if I'm wrong.


[1] Sabbaghi et al., Explicitly Encoding Structural Symmetry is Key to Length Generalization in Arithmetic Tasks.

**Questions:**

Please address the concerns expressed in the weaknesses.

---

> ### Author Response · Authors · 2024-11-21
>
> We thank the reviewer for their time and effort in reviewing our paper. However, our guess is that the reviewer may have misunderstood certain aspects of our paper. Before addressing their concerns, we want to highlight our key contributions.
>
> - First and foremost, the tasks addressed in our work (multi-operand integer addition and integer multiplication) are strictly more general than those considered in the recent literature (2-operand integer addition and integer multiplication with one operand of fixed length). These challenging tasks have not been successfully solved in the literature on length generalization of arithmetic Transformers.
> - Moreover, we propose a novel combination of methods to achieve a significant length generalization in these tasks: multi-level Position Coupling and (possibly multi-stage) scratchpad. By an ablation study (e.g., Figure 1), we demonstrated that neither Position Coupling nor scratchpad alone can achieve such performance, showing that the combination is indeed crucial.
>
> Below, we address the concerns and questions raised by the reviewer.
>
> **W1. What is the advantage of your work above other works on length generalization for arithmetic tasks?**
>
> - We first kindly clarify that the term “first” in our abstract specifically refers to multi-operand addition and general multiplication, rather than 2-operand addition and multiplication with fixed-length multiplier (or multiplicand). The paper [1] you mentioned also addresses the 2-operand addition problem. In our paper, we truthfully acknowledge that numerous researchers have already explored the latter tasks [1-9]. For example, in the second paragraph of Section 1, we state that “as manageable and intriguing test beds, arithmetic and algorithmic tasks are commonly used to study the capabilities (including length generalization) of Transformers,” citing a line of several related works. By the way, we became aware of the paper [1] after the submission deadline; thank you for the pointer, and we have now cited it in our revised manuscript.
> - For addition tasks, our work targets to achieve length generalization for **both** operand length and count. This requires extrapolating the model’s performance simultaneously to unseen operand lengths and unseen operand counts. Multi-operand addition is strictly more challenging than 2-operand addition task because it requires one additional axis of length generalization: the number of operands.
> - Similarly, for multiplication tasks, our work is the first to demonstrate generalization over the lengths of **both** operands. In contrast, the prior studies fix th*e* length of either the multiplicand or the multiplier (usually 1, 2, or 3 digits) and aim to length-generalize in terms of the length of the other operand [1,8].
>
> **W2. Position Coupling had to be compared with other methods that utilize positional encoding for the Parity task.**
>
> - To address the reviewer’s concern, we conducted experiments for the Parity task using RoPE [10] and FIRE [11], both with and without the scratchpad. Please refer to the revised Figure 3. The results indicate that even with the scratchpad, neither RoPE nor FIRE achieves comparable length generalization to that of Position Coupling with scratchpad.
> - As the reviewer pointed out, Position Coupling may not be the only solution for achieving length generalization for the Parity task. However, the main purpose of introducing Section 3 was to illustrate the synergy created by combining scratchpad with Position Coupling, using the Parity task as an introductory example. Our aim was not to claim that Position Coupling is the optimal or only way to achieve length generalization for the Parity task. Indeed, we did not highlight the success of our approach to solving the Parity task in the Contribution section. For this reason, we did not include detailed comparisons with other positional encoding approaches (e.g., RPE, RoPE, FIRE) in the experimental figure of our initial submission.
>
> **W3. Regarding the contribution of the paper.**
>
> - First of all, we do not claim that the input format (padding and reversing) is our primary contribution. As we noted in Lines 83-97, the main contribution of the paper lies in how we **adapt** and **combine** existing techniques—**scratchpad** and **Position Coupling**—to address challenging arithmetic problems (multi-operand addition, general multiplication) that have not been solved in the literature. Our work demonstrates that integrating these methods creates the synergy effect that enables the model to learn the true structure of the task and achieve strong length generalization.

---

> ### Author Response · Authors · 2024-11-21
>
> **W4. What if numbers are given with text? / Multiplication and addition require different encoding.**
>
> - After checking [1], which proposes a framework for handling Text + Numbers by training additional attention heads tailored for arithmetics, we recognize that our approach could be compatible with their framework. This is because our approach and the method proposed in [1] share a similar spirit, as both methods are designed to explicitly encode the structural symmetry of the task. Unfortunately, due to limited computational resources and time, we were not able to test our approach within this framework. Nevertheless, we believe extending this framework is an interesting direction for future research. Thanks for the insightful comment.
> - For the second part, the reviewer is correct that different position encodings are applied to the addition and multiplication tasks. Specifically, the number of Position Coupling modules (bi- vs. tri-) and the Position ID assignment rules vary between the two tasks.
>
> ---
>
> **References**
>
> [1] Sabbaghi et al., Explicitly Encoding Structural Symmetry is Key to Length Generalization in Arithmetic Tasks. arXiv preprint. 2024.
>
> [2] Kim et al., Have You Seen That Number? Investigating Extrapolation in Question Answering Models. ACL 2021.
>
> [3] Nogueira et al., Investigating the Limitations of Transformers with Simple Arithmetic Tasks. ICLR 2021 Workshop on Mathematical Reasoning in General Artificial Intelligence.
>
> [4] Qian et al., Limitations of Language Models in Arithmetic and Symbolic Induction. ACL 2023.
>
> [5] Kajemnejad et al., The impact of positional encoding on length generalization in transformers. NeurIPS 2023.
>
> [6] Zhou et al., What algorithms can transformers learn? a study in length generalization. ICLR 2024.
>
> [7] Zhou et al., Transformers Can Achieve Length Generalization But Not Robustly. ICLR 2024 Workshop ME-FoMo.
>
> [8] Cho et al., Position coupling: Improving length generalization of arithmetic transformers using taskstructure. NeurIPS 2024.
>
> [9] McLeish et al., Transformers can do arithmetic with the right embeddings. NeurIPS 2024.
>
> [10] Su et al., RoFormer: Enhanced transformer with Rotary Position Embedding. Neurocomputing 2024.
>
> [11] Li et al., Functional Interpolation for Relative Positions improves Long Context Transformers. ICLR 2024.

---

> > ### Comment · Reviewer_ois9 · 2024-11-25
> > **Revising the score**
> >
> > Hi,
> > I appreciate the effort you’ve made in clarifying and improving the paper. As it's been pointed out by other reviewers as well, the main weakness of the paper lies in its similarity to previous work [Cho et al]. Nevertheless, I acknowledge the paper's contribution in the length generalization of the task of addition when there are more than two integers, and for the task of multiplication over both operands. Therefore I will increase the score for their contribution to "fair" and the overall to 5. However, the paper still looks like an extension of prior work, and I would encourage the authors to incorporate some of the proposed future directions directly into this paper to strengthen its completeness

---

> > > ### Author Response · Authors · 2024-11-25
> > > **Thank you for raising the score! + Significance of our work**
> > >
> > > Thank you very much for your response and raising the score. We are glad that you recognize some of our contributions.
> > >
> > > However, we would like to emphasize that the contribution of our work is significant enough over Cho et al. (2024), starting with a recent observation on LLM's arithmetic capabilities. Note that the content below is almost identical to the additional response we have already provided to Reviewer joP4.
> > >
> > > ### **Significance of our work**
> > >
> > > - Recall that Cho et al. (2024) have already solved “one-sided” length generalization on the multiplication task where an operand’s length is fixed by a small number (e.g., 1 or 2). At this point, we highlight a recent observation about the LLMs’ performances on integer multiplication tasks (https://x.com/SmokeAwayyy/status/1836157777628250583). You can notice that the “one-sided-long” multiplications are relatively easy instances of arithmetic tasks, especially for o1-mini. However, even o1-mini struggles to solve multiplications between two integers with more than 10 digits: handling both long operands is known to be extremely difficult.
> > > - In fact, it has been already noticed by Cho et al. (2024). In their conclusion section, the “two-sided” length generalization in the multiplication task, considering both operands’ lengths to be long, is mentioned as a challenging problem (as well as the multi-operand addition task) and leaves them as open questions for future work.
> > > - We first identify (in plain text) why these tasks are essentially difficult. In these tasks, including our warm-up example “parity” task, **the number of tokens that are important to entirely solve them linearly increases as the overall sequence length gets longer.** This is a clear difference from the tasks tackled by Cho et al. (2024). Moreover, this is why the plain Position Coupling is insufficient and inapplicable to the parity task, the multi-operand addition task, and the multiplication task, even after we apply some basic input formatting (e.g. reversing, zero-padding).
> > > - To tackle the tasks with an increasing number of important tokens per answer token, we incorporated a scratchpad as a remedy. We emphasize that **our designs of scratchpads for the multi-operand addition task and the general multiplication task are already novel but not obvious!** Importantly, they enable a model to learn the tasks by attending at most a constant number of tokens (as visualized in Figure 7), and we find this particular attention mechanism implementable by carefully adapting Position Coupling in a multi-level sense. As a result, with this synergetic combination (**which is NOT an obvious attachment of two separate methods**), we achieve a significant length generalization in the tasks of our interest. We also empirically verified that neither the scratchpad nor the Position Coupling can solely achieve such a length generalization performance, by conducting extensive ablation studies.
> > > - If you want to visually compare the contributions of this work and Cho et al. (2024), you can simply look at the performance heatmaps of side lengths ~30. The length generalization capability achieved by Cho et al. (2024) can be represented as the leftmost few columns of each heatmap. This area is relatively and considerably narrower than our contribution: **we cover at least about two-thirds of (and up to the whole) heatmap with blue (i.e., high EM accuracies).**
> > > - To summarize: the tasks we aim to solve are still challenging for the latest versions of LLMs. Unfortunately, the naive Position Coupling cannot be an immediate solution due to the increasing number of important tokens for each inference of an answer token. Our remedy is a synergetic amalgamation of two components: a novel design of scratchpads and a careful adaptation of multi-level Position Coupling. By this non-trivial combination, we achieve a remarkable length generalization in two challenging arithmetic length generalization tasks.
> > > - We hope these additional contexts and explanations clarify the noteworthiness of our work.
> > >
> > > Thank you again and feel free to leave further comments or questions regarding our revised paper (marking the modifications in green) and this additional comment if you have any.

---

### Official Review · Reviewer_joP4 · 2024-11-05

**Soundness:** 4
**Presentation:** 4
**Contribution:** 3
**Rating:** 8
**Confidence:** 4

**Summary:**

This paper studies the problem of length generalization in transformer models over arithmetic operations, particularly over both number of operands and length of operands. They achieve 2-3$\times$ length generalization on both addition and multiplication using task-specific scratchpads and multi-level versions of Position Coupling (Cho et al., McLeish et al.). They also provide a theoretical result for a 1-layer transformer showing that it can solve multi-operand addition up to lengths and counts exponential in the embedding dimension.

**Strengths:**

1. The paper is well written with well described experiments and intuition.
2. Length generalization is known to be challenging on arithmetic problems with Transformers. The results in this paper are extremely impressive in that they not only achieve length generalization in operand lenght, but also operand count.
3. The authors perform extensive ablations on th effect of zero-padding, trained-lengths and also connect some of their experimental results with their theoretical construction.

**Weaknesses:**

1. While the problem of length generalization is important, it is mainly relevant in how it applies to more general tasks. However, position  coupling seems to be highly task specific and it is not clear to me how to generalize this idea to non-arithmetic tasks other than very artificial problems.
2. The size of the test set seems to be too small for 30 operands of length up to 30. I would like to see if the test accuracy remains roughly the same even when scaling up the test set sizes.
3. The model assumes a singly digit tokenizer. It would be interesting to see if the results extend to more general tokenizers at all.
4. The zero-padding to match lengths seems to make the input size explode. If you have n inputs with max length n, then the input length will be $O(n^2)$ even if only one operand is of max length.
5. Do you have any intuition on why 1L8H has particularly bad performance? Shouldn't it be strictly more expressive than the 1L4H construction?
6. In Theorem 4.1, does "solve" mean solve perfectly? Are there any constants in max digits or operands? $2^d$ is a very large number, so this is an extremely impressive result!

**Questions:**

1. Is the main contribution of the paper over Cho et al., proving length generalization over number of operands and the introduction of multi-level position coupling? I would appreciate this being made more clear.
2. For the comparison in Figure 3, why use NoPE? What about FIRE or Rotary position embeddings? With NoPE, is the model trained for longer durations?
3. Is there a comparison with He et al., and Zhang et al. on multi-level relative PEs?
4. In the data sampling, what is the purpose of the second chunk? Is it so that you ensure that you see every position?

---

> ### Author Response · Authors · 2024-11-21
>
> We thank the reviewer for the positive review and valuable comments. Below, we address the reviewer’s concern.
>
>
> **W1. It is not clear how to generalize this method to more general non-arithmetic tasks.**
>
> - We agree that the applicability of our approach is currently limited to artificial problems. However, we would like to emphasize that the tasks discussed in this work represent a significant contribution in their own right. Achieving length generalization of Transformers has been recognized as a challenging problem even for simple arithmetic tasks [1-4]. In fact, it was only this year that methods for achieving significant length generalization for addition with two operands and multiplication with a fixed length of multiplicand (or multiplier) were proposed [5, 6, 7]. Thus, we believe that achieving length generalization for multi-operand addition and general multiplication, which are more significantly more challenging tasks that subsume previously considered ones, is a meaningful step forward in the literature.
> - Naturally, the next step of our work is to extend the application of our method to more complex, non-arithmetic tasks. While our method may not be directly utilized, we believe that our key observation—combining the scratchpad with position coupling induces a synergy effect that helps length generalization—can contribute to future research in this direction.
>
>
> **W2. The test set size is too small.**
>
> - To address the reviewer’s concern, we present a test accuracy plot evaluated on 10000 samples for each combination of operand count and length. It turns out that the evaluation of all entries in the plot for a single model requires more than a whole day on our GPU. Due to limited computational resources and time, we were only able to run this evaluation for our baseline setup. We update the result in Figure 1 of the revised paper, where only the last plot (Bi-level Position Coupling+Scratchpad) is evaluated on 10K samples. We observe that there is no degradation in the numbers evaluated with a larger test dataset; in fact, the worst performance among the 970 grids is improved. We will include results for other setups in our final revision.
>
>
> **W3. More general tokenizers?**
>
> - Besides the single-digit tokenizer, a popular choice is the “chunks of n-digits” tokenizer, where “123456” is divided into “123” and “456” when n is 3. Our theoretical construction can be extended to cases where this “chunks of n-digits” tokenizer is used, provided that each number is properly tokenized starting from the least significant digit. While we have not conducted any experiments beyond the single-digit tokenizer, we believe that our approach could still be effective for a “chunks of n-digits” tokenizer, as long as the fundamental structure of addition—tokens within the same position across different operands are summed together and the carry is propagated to the next position—remains unchanged.
> - Additionally, we note that most prior works in the literature have only considered single-digit tokenizers. Even recent LLMs such as Llama 1 and 2 employ single-digit tokenization.
>
>
> **W4. Zero-padding makes the input size explode.**
>
> - As the reviewer pointed out, zero-padding can result in very long query sequences. However, our approach does not strictly rely on zero-padding and can be applied without it. In fact, we already addressed this issue in Lines 324-332 of our paper, where we demonstrate that achieving length generalization without zero-padding is indeed possible.
> - During the rebuttal period, we conducted additional experiments with broader input formats on more seeds.  The results are illustrated in Figure 14 of the revised paper, with the ablation on zero-padding presented in the plot titled NoPad+ReverseResponse.
>
> **W5. Why is 1L8H model’s performance bad?**
>
> - We believe the inferior performance of the 1L8H model is due to the **reduced dimension per head**, which diminishes the expressive power of each head. To validate this claim, we conducted additional experiments by controlling the total head dimension. The results show that the 1L8H model with a 2048 head dimension (256 dim per head) exhibits significantly better performance than the 1L8H model with a 1024 head dimension (128 dim per head). Similarly, we observe the performance degradation of the 1L4H model when we decrease the total head dimension from 1024 to 512 (256 to 128 dim per head). The results can be found in Figure 11 of the revised version.
> - Additionally, it appears that the multi-layer model can mitigate the degradation due to small head dimension, as 8H models with 2 or more layers demonstrate strong performance without increasing the total head dimension.

---

> ### Author Response · Authors · 2024-11-21
>
> **W6. In Thm 4.1, does “solve” mean solve perfectly?**
>
> - We are glad that the reviewer has recognized the significance of this result. Yes, the constructed Transformer model can generate the exact correct response (including the scratchpad) for every possible addition problem, with the number of operands and the maximum digit length up to approximately $2^{(d-19)/4}$. There is no additional constant factor in the exponential term.
> - The key technique lies in assigning the positional embedding vectors such that each position ID corresponds to a vertex of a $\Theta(d)$-dimensional hypercube. Additionally, introducing a moderately high temperature in the softmax operation during the attention layer allows the model to effectively distinguish between position IDs.
>
>
> **Q1. The main contribution of the paper over Cho et al. (2024) should be made more clear.**
>
> - Our main contribution can be summarized as follows:
> - Novelty in problem settings: Our paper is the first work that successfully achieves length generalizations on multi-integer addition and general integer multiplication (that does not fix the length of the 2nd operand). These arithmetic tasks include previously studied tasks as special cases, and are considered difficult in the literature.
> - Novelty in the method: We emphasize that our design of scratchpads also plays a crucial role in our approach. To achieve significant length generalization in our tasks, we show that our proposed **combination** of scratchpad & multi-level Position Coupling is both crucial and effective. Neither the scratchpad nor Position Coupling alone can achieve comparable length generalization for these tasks.
>
>
> **Q2. Regarding comparisons on Parity (Figure 3)**
>
> - The purpose of Section 3 is to motivate the combination of the scratchpad with Position Coupling, and the Parity task is used as an introductory example due to its simplicity. For this reason, our initial submission only compared our approach with NoPE, trained under the same condition, including the total number of iterations.
> - However, another reviewer also commented on this as a weakness.  In response, we have added experimental results for RoPE and FIRE. Please refer to Figure 3 in our revised manuscript for the updated results. The results show that even with scratchpads, neither RoPE nor FIRE achieves length generalization performance comparable to that of Position Coupling with the scratchpad.
> - Additionally, we note that NoPE is a reasonable baseline in the literature on length generalization for algorithmic/arithmetic tasks, as Kazemnejad et al. (2023) have argued that decoder-only Transformers with NoPE exhibit a certain amount of length generalization, comparable to or better than existing PE methods such as T5’s Relative PE, Absolute PE, and Rotary PE.
>
> **Q3. Comparison with multi-level relative PEs**
>
> - Bilevel Positional Encoding (BiPE) (He et al., 2024) is a positional encoding scheme designed to improve length extrapolation for natural language processing tasks. In this scheme, each token is assigned two position IDs: one representing the intra-segment position and the other representing the inter-segment position.
> - Similarly, Hierarchical Rotary Position Embedding (HiRoPE) (Zhang et al., 2024) utilizes two position IDs for each token. This method aims to improve length extrapolation for code models, and designs the first position ID to represent token-level information and the second position ID to represent the functional level or statement level in source code.
> - In summary, all three methods—Position Coupling, BiPE, and HiRoPE—share a common spirit of leveraging multi-level position ID to reflect the structural characteristics of a given task. However, we note that the specific structures they aim to represent differ. For example, Position Coupling encodes (1) the digit place within each operand and (2) the operand index; BiPE encodes (1) the token place within each paragraph and (2) the paragraph index; and HiRoPE encodes (1) the token place within each function and (2) the function index. Also, the specific implementation details vary: Position Coupling is based on APE while BiPE combines APE and RPE, and HiRoPE employs RoPE. We believe that implementing Position Coupling with RPE or RoPE would be an interesting research direction, but due to limited computational resources and time, we were not able to test this idea during the rebuttal period.

---

> ### Author Response · Authors · 2024-11-21
>
> **Q4. What is the second chunk for (in data sampling of multi-operand addition)?**
>
> - Using only the first chunk, the most significant digit (MSD) of the answer is often small due to the zero-paddings, because the lengths of the operands are mostly different from each other in training time. To mitigate the sampling bias in terms of the answer’s MSD, we manually match the operand lengths for a fixed portion of the training data. This is why the second chunk is introduced.
>
> ---
>
> **References**
>
> [1] Kim et al., Have You Seen That Number? Investigating Extrapolation in Question Answering Models. ACL 2021.
>
> [2] Nogueira et al., Investigating the Limitations of Transformers with Simple Arithmetic Tasks. ICLR 2021 Workshop on Mathematical Reasoning in General Artificial Intelligence.
>
> [3] Kajemnejad et al., The impact of positional encoding on length generalization in transformers. NeurIPS 2023.
>
> [4] Zhou et al., What algorithms can transformers learn? a study in length generalization. ICLR 2024.
>
> [5] Zhou et al., Transformers Can Achieve Length Generalization But Not Robustly. ICLR 2024 Workshop ME-FoMo.
>
> [6] Cho et al., Position coupling: Improving length generalization of arithmetic transformers using taskstructure. NeurIPS 2024.
>
> [7] McLeish et al., Transformers can do arithmetic with the right embeddings. NeurIPS 2024.

---

> > ### Comment · Reviewer_joP4 · 2024-11-24
> > **Thank you for the detailed/exhaustive responses**
> >
> > I thank the authors for their detailed and exhaustive responses. They have addressed each of my concerns thoroughly. In fact, if possible, I would recommend they add some of the additional explanations they have provided me here into the paper if they haven't done so already.
> >
> > I still retain one concern however. While I find the paper to be good, I am still not entirely sure about the significance of both the problem and the improvement over Cho et al. It is definitely an improvement, but I don't find it particularly significant. Therefore, I would retain my score. I would still like to commend the authors for their comprehensive experimental work.

---

> > > ### Author Response · Authors · 2024-11-25
> > > **Thank you for your response! + Significance of our work**
> > >
> > > Thank you for your response! We are happy to hear that our rebuttal addressed your concerns. Also, we revised our paper, marking the changes in green. Please take a look at our revised manuscript if you haven't done so, and feel free to leave further comments or questions if you have any.
> > >
> > > ## **Significance of our work**
> > >
> > > - Let us emphasize the significance of our work over Cho et al. (2024). Recall that they already solved “one-sided” length generalization on the multiplication task where an operand’s length is fixed by a small number (e.g., 1 or 2). At this point, we highlight a recent observation about the LLMs’ performances on integer multiplication tasks (https://x.com/SmokeAwayyy/status/1836157777628250583). You can notice that the “one-sided-long” multiplications are relatively easy instances of arithmetic tasks, especially for o1-mini. However, even o1-mini struggles to solve multiplications between two integers with more than 10 digits: handling both long operands is known to be extremely difficult.
> > > - In fact, it has been already noticed by Cho et al. (2024). In their conclusion section, the “two-sided” length generalization in the multiplication task, considering both operands’ lengths to be long, is mentioned as a challenging problem (as well as the multi-operand addition task) and leaves them as open questions for future work.
> > > - We first identify (in plain text) why these tasks are essentially difficult. In these tasks, including our warm-up example “parity” task, the number of tokens that are important to entirely solve them linearly increases as the overall sequence length gets longer. This is a clear difference from the tasks originally tackled by Cho et al. (2024): in the two-operand addition task, for example, only at most a constant number of tokens is important to infer every answer token. Moreover, this is why the plain Position Coupling is inapplicable to the parity task, the multi-operand addition task, and the multiplication task, even after we apply some basic input formatting (e.g. reversing, zero-padding).
> > > - In order to tackle the tasks with an increasing number of important tokens per answer token, we incorporated scratchpad as a remedy. We emphasize that our designs of scratchpads for the multi-operand addition task and the general multiplication task are already novel but not obvious! Importantly, they enable a model to learn the tasks by attending at most a constant number of tokens (as visualized in Figure 7), and we find this particular attention mechanism implementable by carefully adapting Position Coupling in a multi-level sense. As a result, with this synergetic combination (**which is NOT an obvious attachment of two separate methods**), we achieve a significant length generalization in the tasks of our interest. We also empirically verified that neither the scratchpad nor the Position Coupling can solely achieve such a length generalization performance, by conducting extensive ablation studies.
> > > - If you want to *visually* compare the contributions of this work and Cho et al. (2024), you can simply look at the performance heatmaps of side lengths ~30. The length generalization capability achieved by Cho et al. (2024) can be represented as the leftmost (one or two) column(s) of each heatmap. This area is relatively and considerably narrower than our contribution: **we cover at least about two-thirds of (and up to the whole) heatmap with blue (i.e., high EM accuracies).**
> > > - To summarize: the tasks we aim to solve are still challenging for the latest versions of LLMs. Unfortunately, the naive Position Coupling cannot be an immediate solution due to the increasing number of important tokens for each inference of an answer token. Our remedy is a synergetic amalgamation of two components: a novel design of scratchpads and a careful adaptation of multi-level Position Coupling. By this non-trivial combination, we achieve a remarkable length generalization in two challenging arithmetic length generalization tasks.
> > > - We hope these additional contexts and explanations clarify the noteworthiness of our work.

---

> ### Comment · Reviewer_joP4 · 2024-11-26
> **Thank you for the edits and clarification**
>
> I thank the authors for explaining in detail, their contribution over Cho et al. and also pointing out the significance and hardness of their results. After reading their response and going through their edited manuscript, I am convinced and am raising my score to 8.

---

> > ### Author Response · Authors · 2024-11-27
> > **Thank you for your response and raising the score!**
> >
> > Thank you very much for recognizing the significance of our results and the difficulty of the tasks we solved, and raising the score accordingly! We will make the significance of our work clearer in our final manuscript.

---

### Meta-Review · Area_Chair_LHGd · 2024-12-22

**Metareview:**

This paper combines scratchpad learning with position coupling to enable Transformer models to generalize to longer sequences in arithmetic tasks. The approach uses task-specific scratchpads and multi-level position coupling to help models focus on relevant tokens.

Reviewers appreciated the clear technical presentation and strong empirical results.
Key concerns raised during review included: (1) limited contribution given the paper's reliance on previously established techniques like position coupling, and (2) the specificity of the approach to pure arithmetic tasks without considering more general contexts like numbers embedded in text.

The authors provided responses addressing these concerns, particularly regarding the technical challenges of their target tasks. Given the solid technical execution and clear results on challenging arithmetic tasks, I recommend accepting this pape. The authors are strongly encouraged to incorporate reviewer feedback in their camera-ready.

**Additional Comments On Reviewer Discussion:**

See above

---

### Decision · Program_Chairs · 2025-01-22

Accept (Poster)